# Learning Mean Field Games on Sparse Graphs: A Hybrid Graphex Approach

**Christian Fabian, Kai Cui & Heinz Koeppl**
Dept. of Electrical Engineering and Information Technology, Technische Universität Darmstadt
`{christian.fabian, heinz.koeppl}@tu-darmstadt.de`

## Abstract

Learning the behavior of large agent populations is an important task for numerous research areas. Although the field of multi-agent reinforcement learning (MARL) has made significant progress towards solving these systems, solutions for many agents often remain computationally infeasible and lack theoretical guarantees. Mean Field Games (MFGs) address both of these issues and can be extended to Graphon MFGs (GMFGs) to include network structures between agents. Despite their merits, the real world applicability of GMFGs is limited by the fact that graphons only capture dense graphs. Since most empirically observed networks show some degree of sparsity, such as power law graphs, the GMFG framework is insufficient for capturing these network topologies. Thus, we introduce the novel concept of Graphex MFGs (GXMFGs) which builds on the graph theoretical concept of graphexes. Graphexes are the limiting objects to sparse graph sequences that also have other desirable features such as the small world property. Learning equilibria in these games is challenging due to the rich and sparse structure of the underlying graphs. To tackle these challenges, we design a new learning algorithm tailored to the GXMFG setup. This hybrid graphex learning approach leverages that the system mainly consists of a highly connected core and a sparse periphery. After defining the system and providing a theoretical analysis, we state our learning approach and demonstrate its learning capabilities on both synthetic graphs and real-world networks. This comparison shows that our GXMFG learning algorithm successfully extends MFGs to a highly relevant class of hard, realistic learning problems that are not accurately addressed by current MARL and MFG methods.

## 1 Introduction

In various research fields, scientists are confronted with situations where many agents or particles interact with each other. Examples range from energy optimization (González-Briones et al., 2018) or financial loan markets (Corbae & D'Erasmo, 2021) to the dynamics of public opinions (You et al., 2022). To predict the behavior in these setups of great practical interest, one can employ numerous algorithms from the field of multi-agent reinforcement learning (MARL), see Zhang et al. (2021) or Gronauer & Diepold (2022) for overviews. Although there is a rapidly growing interest in MARL methods, many approaches are hardly scalable in the number of agents and often lack a rigorous theoretical foundation. One way to address these issues is the concept of mean field games (MFGs) introduced by Huang et al. (2006) and Lasry & Lions (2007). The basic assumption of MFGs and their corresponding learning methods is that there is a large number of indistinguishable agents where each individual has an almost negligible influence on the entire system. Then, it is possible to focus on a representative agent and infer the group behavior from the representative individual. Approaches to learning MFGs include entropy regularization (Cui & Koeppl, 2021a; Guo et al., 2022; Anahtarci et al., 2023), fictitious play (Cardaliaguet & Hadikhanloo, 2017; Perrin et al., 2020; Elie et al., 2020; Xie et al., 2021; Min & Hu, 2021), normalizing flows (Perrin et al., 2021), and online mirror descent (OMD) (Pérolat et al., 2022; Laurière et al., 2022b), see Laurière et al. (2022a) for a survey.

Despite their benefits, standard MFGs also have some strong limitations. Especially, basic MFGs assume that each agent interacts with all of the other agents. For real-world applications such as financial markets, pandemics, or social networks it appears to be more realistic that agents are connected by some sort of network and only interact with their network neighbors. To incorporate

such network structures into MFGs, Graphon MFGs (GMFGs) have been proposed and learned in the literature (Caines & Huang, 2019; Gao et al., 2021; Cui & Koeppl, 2021b; Vasal et al., 2021). The graph theoretical concept of graphons gives a limiting object for growing sequences of random dense graphs, see Lovász (2012). However, their applicability to real-world networks is limited by the rather strong assumptions on the networks necessary to use graphon theory.

GMFG algorithms can only learn equilibria for dense graph sequences where each agent is connected to an infinite number of neighbors in the limit. This excludes crucial sparse networks such as power laws graphs observed in many real world networks (Newman, 2018). Even somewhat sparse extensions of GMFGs, namely LPGMFGs (Fabian et al., 2023), can only model networks with low power law coefficients between zero and one where the number of neighbors per agent still diverges to infinity. Besides the degree distribution, researchers are frequently interested in other important graph features such as the so-called small world property. This property states that if persons A and B are friends and B and C are friends, A and C are likely to be friends as well (*the friend of my friend is my friend*) and is a characteristic for many real world networks (Watts, 1999; Amaral et al., 2000).

To obtain an expressive model and corresponding learning algorithm, we employ a different, both more flexible and more involved concept, called a graphex (Caron & Fox, 2017; Veitch & Roy, 2015; Borgs et al., 2018b) which generalizes the graphon idea and builds on the representation theorem by Kallenberg (1990). While we point to later sections for details, a graphex intuitively enables the modelling of graph sequences with features such as the small world property and sparse graph sequences where almost all nodes have a finite degree in the limit. Furthermore, their flexibility enables the combination of different graph features, e.g., having a block structure with a power law degree distribution, see Caron et al. (2022). Building on graphexes, we propose graphex mean field games (GXMFGs) which bring the above mentioned network features into an MFG setup and thereby closer to reality. This expressiveness adds substantial new challenges for the design of a learning algorithm as well as the respective theory which we address with a novel hybrid graphex approach.

The aim of this paper is to design a learning algorithm for this challenging and realistic setup and provide a theoretical analysis of the model. Thus, we propose the hybrid graphex learning approach, which consists of two subsystems, a highly connected core and a sparsely connected periphery. This allows us to depict the large fraction of finite degree agents, who would be insufficiently modelled by standard MFG techniques. Our method contains a novel learning scheme specifically designed for the hybrid character of the system. Finally, we demonstrate the predictive power of our learning algorithm on both synthetic and real world networks. These empirical observations show that GXMFGs can formalize agent interactions in many complex real world networks and provide a learning mechanism to determine equilibria in these systems. Our contributions can be summarized as follows:

1. We define the novel concept of graphex mean field games to extend MFGs to an important class of problems;

2. We provide theoretical guarantees to show that GXMFGs are an increasingly accurate approximation of the finite system;

3. We develop a learning algorithm tailored to the challenging class of GXMFGs, where we exploit the hybrid structure caused by the sparse nature of the underlying graphs;

4. We demonstrate the accuracy of our GXMFG approximation on different examples on both synthetic and empirical networks.

## 2 THE GRAPHEX CONCEPT

This section provides a brief introduction to graphex theory. It is based on the existing literature to which we refer to for more details, e.g. Caron et al. (2022) and Janson (2022). Intuitively, the crucial benefit of graphexes is that they can capture the structure of many real world networks where both graphons and Lp graphons provide insufficient results, see Figure 1 for an illustrative example. In our context, a graphex $W : [0, \infty)^2 \to [0, 1]$ is a symmetric, measurable function with $0 < \int_{\mathbb{R}^2_+} W(\alpha, \beta) \, d\alpha d\beta < \infty$ and $\int_{\mathbb{R}_+} W(\alpha, \alpha) \, d\alpha < \infty$. For technical reasons, we also assume that $\lim_{\alpha \to \infty} W(\alpha, \alpha)$ and $\lim_{\alpha \to 0} W(\alpha, \alpha)$ both exist. For convenience, we often write

$$\xi_W(\alpha) \coloneqq \int_{\mathbb{R}_+} W(\alpha, \beta) \, d\beta \quad \text{and} \quad \bar{\xi}_W \coloneqq \int_{\mathbb{R}^2_+} W(\alpha, \beta) \, d\alpha d\beta \,.$$

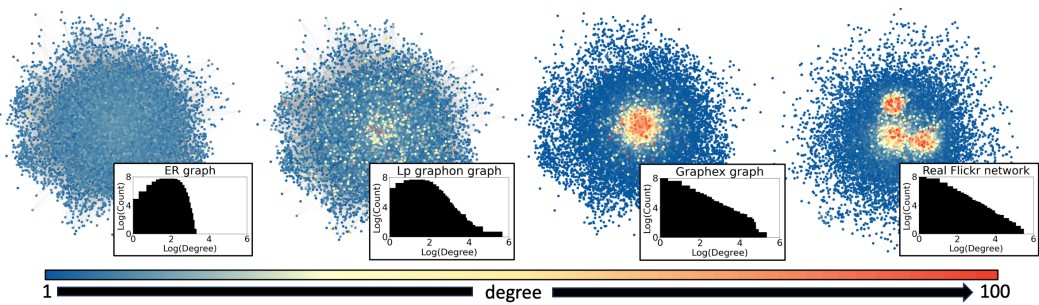

Figure 1: Four networks, each with around 15k nodes and 50k edges. The Erdős-Rényi graph (left) generated by a standard graphon does not have any high degree nodes. The Lp graphon graph (middle left) has a few high degree nodes expressed by the flat degree distribution tail. The graphex (middle right) and real subsampled Flickr network (right) (data from Mislove et al. (2007); Kunegis (2013)) show a very similar degree distribution with rather broad tails and a core-periphery like structure.

Now, we can associate each finite graph $G$ with a graphex $W^G$. Therefore, assume that $G$ is some finite graph with vertex set $V(G)$ and edge set $E(G)$ and that without loss of generality the $N$ vertices are ordered in some way $v_1, \ldots, v_N$. One can associate $G$ with a graphex $W^G$ as follows. First, partition the unit interval into $N$ subintervals $I_1, \ldots, I_N$ of equal length with $I_j := \left(\frac{j-1}{N}, \frac{j}{N}\right]$ for all $1 \leq j \leq N$. Then, $W^G(\alpha, \beta) = 1$ for $\alpha \in I_j$ and $\beta \in I_k$ if and only if $(v_j, v_k) \in E(G)$ and $W^G(\alpha, \beta) = 0$ otherwise. To obtain expressive associated graphexes for sparse graphs, we stretch them by defining the stretched canonical graphon as $W^{G,s}(\alpha, \beta) := W^G\left(\|W^G\|_1^{1/2}\alpha, \|W^G\|_1^{1/2}\beta\right)$ if $0 \leq \alpha, \beta \leq \|W^G\|_1^{-1/2}$ with $\|W^G\|_1 = 2E(G)/N^2$ and zero otherwise.

Conversely, to sample almost surely finite graphs from a graphex, we start with a unit-rate Poisson process $(\theta_i, \vartheta_i)_{i \in \mathbb{N}}$ on $\mathbb{R}_+^2$ where each tuple $(\theta_i, \vartheta_i)$ is a potential node for the sampled graph. Then, any pair of candidate nodes $(\theta_i, \vartheta_i), (\theta_j, \vartheta_j)$ is connected by an edge independent of all other edges with probability $W(\vartheta_i, \vartheta_j)$. To obtain an almost surely finite sampled graph, we stop the process at an arbitrary but fixed time $\nu > 0$ by only keeping candidate nodes with $\theta_i \leq \nu$ and edges $(v_i, v_j)$ with $\theta_i, \theta_j \leq \nu$. In the remaining graph, we discard all isolated vertices with degree 0 which yields an almost surely finite graph $G_\nu = (V_\nu, E_\nu)$ (Veitch & Roy, 2015, Theorem 4.9). A random sequence of such sampled graphs $(G_\nu)_{\nu>0}$ converges almost surely to the generating graphex in the stretched cut metric (Borgs et al., 2018b, Theorem 28). This motivates our first assumption.

**Assumption 1.** *a) The sequence of stretched empirical graphexes $(\widehat{W}_\nu)_{\nu>0}$ converges to the limiting graphex $W$ in the cut norm, i.e.*

$$\|W - \widehat{W}_\nu\|_\square := \sup_{U,V} \left| \int_{U \times V} W(\alpha, \beta) - \widehat{W}_\nu(\alpha, \beta) \, \mathrm{d}\alpha\mathrm{d}\beta \right| \to 0 \quad as \quad \nu \to \infty$$

*where the supremum is over measurable subsets $U, V \subset \mathbb{R}_+$. Also, assume that*

*b) $\xi_W$ is non-increasing with $\xi_W^{-1}(\alpha) := \inf\{\beta > 0 : \xi_W(\beta) \leq \alpha\} = (1 + o(1))\ell(1/\alpha)\alpha^{-\sigma}$ as $\alpha \to 0$ with $\sigma \in [0, 1]$ and $\ell$ a slowly varying function ($\forall c > 0 : \lim_{x \to \infty} \ell(cx)/\ell(x) = 1$);*

*c) and there exist $C, a > 0$ and $\alpha_0 \geq 0$ with $\xi_W(\alpha_0) > 0$ such that $\int_0^\infty W(\alpha, z)W(\beta, z)\mathrm{d}z \leq C\xi_W(\alpha)^a\xi_W(\beta)^a$ for all $\alpha, \beta \geq \alpha_0$ with $a > \max\{\frac{1}{2}, \sigma\}$ if $\sigma \in [0, 1)$ and $a = 1$ if $\sigma = 1$.*

Parts b) and c) of Assumption 1 are standard, see Caron et al. (2022) for details. Assumption 1 b) focuses on the behavior of $\xi_W$ at infinity and states for $\sigma \in (0, 1)$ that $\xi_W(\alpha)$ roughly follows a power function $\alpha^{-\sigma}$ for large $\alpha$. Assumption 1 c) is of technical nature and especially holds for the rich class of separable graphexes which satisfy $W(\alpha, \beta) = \xi_W(\alpha)\xi_W(\beta)/\bar{\xi}_W$. Combining these insights, Assumption 1 is especially fulfilled by the separable power-law graphex used in our learning algorithm. We choose the separable power-law graphex for its conceptual simplicity and ability to capture the topology of many real world networks (Naulet et al., 2021) and leave the use of different graphexes to future work. Graphexes are the limit of very different sparse graph sequences than those depicted by Lp graphons (Borgs et al., 2018a; 2019), see Borgs et al. (2018b, Proposition 20).

## 3 GRAPHEX MEAN FIELD GAMES

In this section we introduce the finite and limiting game. The proofs corresponding to the theoretical results can be found in the appendix. Throughout the paper, we assume the state space $\mathcal{X}$ and action space $\mathcal{U}$ to be finite and work with a discrete, finite time horizon $\mathcal{T} := \{0, \ldots, T-1\}$. Let $\mathcal{P}(A)$ be the set of probability distributions on an arbitrary finite set $A$. We start with the finite game.

### 3.1 THE FINITE GAME

Consider a finite set $V_\nu$ of agents who are connected by a graph $G_\nu = (V_\nu, E_\nu)$ sampled from a graphex by stopping at $\nu$. For an individual $i \in V_\nu$ define the neighborhood state distribution by

$$\mathbb{G}_{i,t}^\nu := \frac{1}{\deg_{V_\nu}(i)} \sum_{j \in V_\nu} \mathbf{1}_{\{ij \in E_\nu\}} \delta_{X_t^j}.$$

Each agent $i$ chooses a policy $\pi_i \in \Pi := \mathcal{P}(\mathcal{U})^{\mathcal{T} \times \mathcal{X}}$ to competitively maximize the objective

$$J_i^\nu(\pi_1, \ldots, \pi_{|V_\nu|}) = \mathbb{E}\left[\sum_{t \in \mathcal{T}} r\left(X_{i,t}, U_{i,t}, \mathbb{G}_{i,t}^\nu\right)\right]$$

where $r : \mathcal{X} \times \mathcal{U} \times \mathcal{P}(\mathcal{X}) \to \mathbb{R}$ is some arbitrary reward function. The dynamics of the finite system are for all $i \in V_\nu$ given by the initialization $X_{i,0} \sim \mu_0$ and for all $t \in \mathcal{T}$ by

$$U_{i,t} \sim \pi_{i,t}\left(\cdot \mid X_{i,t}\right) \quad \text{and} \quad X_{i,t+1} \sim P\left(\cdot \mid X_{i,t}, U_{i,t}, \mathbb{G}_{i,t}^\nu\right).$$

Here, $P : \mathcal{X} \times \mathcal{U} \times \mathcal{P}(\mathcal{X}) \to \mathcal{P}(\mathcal{X})$ is an arbitrary transition kernel that formalizes the transition probabilities from the current state to the next one given the action and neighborhood. To complete the model, we adopt a suitable equilibrium concept from the literature (Carmona, 2004; Elie et al., 2020) that allows for the occurrence of small local deviations from the limiting graphex structure.

**Definition 1.** *An $(\varepsilon, p)$-Markov-Nash equilibrium (MNE) with $\varepsilon, p > 0$ is a tuple of policies $(\pi_1, \ldots, \pi_{|V_\nu|}) \in \Pi^{|V_\nu|}$ such that there exists some set $V_\nu' \subset V_\nu$ with $|V_\nu'| \geq (1-p)|V_\nu|$ and*

$$J_i^\nu(\pi_1, \ldots, \pi_{|V_\nu|}) \geq \sup_{\bar{\pi}_i \in \Pi} J_i^\nu(\pi_1, \ldots, \bar{\pi}_i, \ldots, \pi_{|V_\nu|}) - \varepsilon \quad \text{for all} \quad i \in V_\nu'.$$

### 3.2 THE LIMITING GXMFG SYSTEM

Due to the underlying graphex structure, our limiting system with infinitely many agents differs significantly from existing MFG models which are based on graphons, for example. It consists of two subsystems we call the high degree core and the low degree periphery which both have very different characteristics. While the core consists of relatively few agents with a high number of connections between themselves, low degree individuals in the periphery almost exclusively connect to agents in the core. Analyzing this novel hybrid system is challenging and allows us to develop a new learning algorithm for approximating equilibria in these sparse and often rather realistic systems, see e.g. Figure 1. To obtain meaningful theoretical results, we make a standard Lipschitz assumption.

**Assumption 2.** *$P, W$, and $r$ are Lipschitz.*

**The high degree core.** Nodes are part of the core if their latent parameter $\alpha$ fulfills $0 \leq \alpha \leq \alpha^*$. Here, $0 < \alpha^* < \infty$ is an arbitrary but fixed cutoff parameter that marks the border of the core. With an increasing stopping time $\nu$ the expected degrees in the core also increase and become infinite in the limit. Therefore, highly connected core agents are characterized by their low parameters $\alpha \leq \alpha^*$. The neighborhood distribution $\mathbb{G}_{\alpha,t}^\infty \in \mathcal{P}(\mathcal{X})$ for every $0 \leq \alpha \leq \alpha^*$ and $t \in \mathcal{T}$ is defined by

$$\mathbb{G}_{\alpha,t}^\infty(\boldsymbol{\mu}) := \frac{1}{\xi_{W,\alpha^*}(\alpha)} \int_0^{\alpha^*} W(\alpha, \beta)\mu_{\beta,t} \, \mathrm{d}\beta$$

with $\xi_{W,\alpha^*}(\alpha) := \int_0^{\alpha^*} W(\alpha, \beta)\mathrm{d}\beta$. For notational convenience, we often drop the dependence on $\boldsymbol{\mu}$ and just write $\mathbb{G}_{\alpha,t}^\infty$ when $\boldsymbol{\mu}$ is clear from the context. Then, for all core agents $\alpha \in [0, \alpha^*]$ the model dynamics are

$$U_{\alpha,t} \sim \pi_{\alpha,t}^\infty\left(\cdot \mid X_{\alpha,t}\right) \quad \text{and} \quad X_{\alpha,t+1} \sim P\left(\cdot \mid X_{\alpha,t}, U_{\alpha,t}, \mathbb{G}_{\alpha,t}^\infty\right)$$

for all $t \in \mathcal{T}$ and initial $X_{\alpha,0} \sim \mu_0$. Each agent $\alpha$ chooses a policy $\pi_\alpha$ to maximize $J_\alpha^{\boldsymbol{\mu}} := \mathbb{E}\left[\sum_{t \in \mathcal{T}} r\left(X_{\alpha,t}, U_{\alpha,t}, \mathbb{G}_{\alpha,t}^\infty\right)\right]$. The corresponding core MF forward equation is given by

$$\mu_{\alpha,t+1}^\infty := \mu_{\alpha,t}^\infty P_{t,\boldsymbol{\mu}',W}^{\pi,\infty} := \sum_{x \in \mathcal{X}} \mu_{\alpha,t}^\infty(x) \sum_{u \in \mathcal{U}} \pi_{\alpha,t}^\infty(u \mid x) \cdot P\left(\cdot \mid x, u, \mathbb{G}_{\alpha,t}^\infty(\boldsymbol{\mu}')\right) \tag{1}$$

with $\mu_{\alpha,0} = \mu_0$. Furthermore, define the space of measurable core mean field ensembles $\boldsymbol{\mathcal{M}}^\infty := \mathcal{P}(\mathcal{X})^{\mathcal{T} \times [0,\alpha^*]}$ with $\alpha \mapsto \mu_{\alpha,t}$ being measurable for all $(\boldsymbol{\mu}, t, x) \in \boldsymbol{\mathcal{M}}^\infty \times \mathcal{T} \times \mathcal{X}$. Analogously, the space $\boldsymbol{\Pi}^\infty \subseteq \Pi^{[0,\alpha^*]}$ contains all measurable policy ensembles with $\alpha \mapsto \pi_{\alpha,t}(u|x)$ being measurable for all $(\boldsymbol{\pi}, t, x, u) \in \boldsymbol{\Pi}^\infty \times \mathcal{T} \times \mathcal{X} \times \mathcal{U}$. We write $\boldsymbol{\mu} = \Psi(\boldsymbol{\pi})$ for a tuple $(\boldsymbol{\mu}, \boldsymbol{\pi}) \in \boldsymbol{\mathcal{M}}^\infty \times \boldsymbol{\Pi}^\infty$ if $\boldsymbol{\mu}$ is generated by $\boldsymbol{\pi}$ according to the above MF forward equation. Conversely, define $\Phi(\boldsymbol{\mu})$ as the set of optimal policy ensembles $\boldsymbol{\pi}$ with $\pi_\alpha \in \arg\max_{\pi \in \Pi} J_\alpha^{\boldsymbol{\mu}}(\pi)$ for every $\alpha \in [0, \alpha^*]$ under the given MF $\boldsymbol{\mu}$. For the limiting system we define a mean field core equilibrium (MFCE).

**Definition 2.** *A MFCE is a tuple $(\boldsymbol{\mu}, \boldsymbol{\pi}) \in \boldsymbol{\mathcal{M}}^\infty \times \boldsymbol{\Pi}^\infty$ with $\boldsymbol{\mu} = \Psi(\boldsymbol{\pi})$ and $\boldsymbol{\pi} \in \Phi(\boldsymbol{\mu})$.*

Under a standard Lipschitz assumption, a mean field core equilibrium is guaranteed to exist.

**Lemma 1.** *Under Assumption 2 there exists a mean field core equilibrium $(\boldsymbol{\mu}, \boldsymbol{\pi}) \in \boldsymbol{\mathcal{M}}^\infty \times \boldsymbol{\Pi}^\infty$.*

For measurable, bounded functions $f \colon \mathcal{X} \times \mathcal{I} \to \mathbb{R}$ we define a mean field core operator by

$$\boldsymbol{\mu}_t^\infty(f, \alpha^*) := \frac{1}{\alpha^*} \int_0^{\alpha^*} \sum_{x \in \mathcal{X}} f(x, \alpha) \mu_{\alpha,t}(x) \, \mathrm{d}\alpha \,.$$

**The low degree periphery.** Let $\boldsymbol{\mathcal{G}}^k := \{G \in \mathcal{P}(\mathcal{X})^k : k \cdot G \in \mathbb{N}_0^k\}$ be the set of possible neighborhoods for agents with a finite number $k \in \mathbb{N}$ of neighbors. For agents in the periphery with latent parameter $\alpha$ and finite degree $k$ the model dynamics are

$$\mathbb{G}_{\alpha,t}^k \sim P_{\boldsymbol{\pi}}\left(\mathbb{G}_{\alpha,t}^k = \cdot \mid X_{\alpha,t}\right), \quad U_{\alpha,t} \sim \pi_{\alpha,t}^k(\cdot \mid X_{\alpha,t}), \quad X_{\alpha,t+1} \sim P\left(\cdot \mid X_{\alpha,t}, U_{\alpha,t}, \mathbb{G}_{\alpha,t}^k\right)$$

for all $t \in \mathcal{T}$ and $X_{\alpha,0} \sim \mu_0$. The probability distribution $P_{\boldsymbol{\pi}}\left(\mathbb{G}_{\alpha,t}^k = \cdot \mid X_{\alpha,t}\right)$ can be calculated analytically, see Appendix C for details. A key difference to the core dynamics with deterministic neighborhoods in the limit is that in the periphery, the neighborhoods remain stochastic. This is caused by the finite number of neighbors of a periphery agent, which is finite even in the limit. An extension to degree specific transition kernels ($P^k$ for each $k$) is straightforward but neglected for expositional simplicity. Each agent $\alpha$ chooses a policy $\pi_\alpha$ to maximize $J_\alpha^{\boldsymbol{\mu}}(\pi_\alpha) := \mathbb{E}\left[\sum_{t \in \mathcal{T}} r\left(X_{\alpha,t}, U_{\alpha,t}, \mathbb{G}_{\alpha,t}^k\right)\right]$. The corresponding MF evolves according to

$$\mu_{\alpha,t+1}^k := \mu_{\alpha,t}^k P_{t,\boldsymbol{\mu}',W}^{\pi,k} := \sum_{x \in \mathcal{X}} \mu_{\alpha,t}^k(x) \sum_{G \in \boldsymbol{\mathcal{G}}^k} P_{\boldsymbol{\pi}}\left(\mathbb{G}_{\alpha,t}^k(\boldsymbol{\mu}') = G \mid x_{\alpha,t} = x\right)$$
$$\cdot \sum_{u \in \mathcal{U}} \pi_{\alpha,t}^k(u \mid x) \cdot P\left(\cdot \mid x, u, G\right) \,.$$

The periphery empirical MF operator for finite degree $k$ and measurable, bounded $f \colon \mathcal{X} \times \mathcal{I} \to \mathbb{R}$ is

$$\hat{\boldsymbol{\mu}}_t^{\nu,k}(f) := \frac{\sqrt{2|E_\nu|}}{|V_{\nu,k}|} \sum_{i \in V_\nu} \mathbf{1}_{\{\deg(v_i)=k\}} \int_{\frac{i-1}{\sqrt{2|E_\nu|}}}^{\frac{i}{\sqrt{2|E_\nu|}}} \sum_{x \in \mathcal{X}} f(x, \alpha) \hat{\mu}_{\alpha,t}^\nu(x) \, \mathrm{d}\alpha$$

which intuitively is the average over evaluating the function $f$ over all agents with degree $k$. For the $k$-degree mean field in the limiting system stopped at $\nu$ we analogously define the MF operator

$$\boldsymbol{\mu}_t^k(f, \nu) := \frac{\sqrt{\bar{\xi}_W}}{\int_0^\infty \mathrm{Poi}_{\nu,\alpha}^W(k)\mathrm{d}\alpha} \int_0^\infty \mathrm{Poi}_{\nu,\alpha}^W(k) \sum_{x \in \mathcal{X}} f(x, \alpha) \mu_{\alpha,t}^k(x) \, \mathrm{d}\alpha$$

where $\mathrm{Poi}_{\nu,\alpha}^W(k)$ is the probability for $k$ in a Poisson distribution with parameter $\nu\xi_W(\alpha)$. Stopping at $\nu$ is necessary to ensure the existence of the integral via the factor $\mathrm{Poi}_{\nu,\alpha}^W(k)$. As shown in the next subsection, these two operators are closely related for sufficiently large $\alpha^*$ and $\nu$. For our learning algorithm, we choose $k_{max} < \infty$ as the maximal degree contained in the periphery.

**Remark 1.** *The union of the core and periphery does not contain some intermediate degree nodes. These intermediate nodes are negligible for the limiting system and our learning algorithm for two reasons. Due to the integrability of the graphex $W$, intermediate nodes have a vanishing influence on the neighborhoods of all other agents for sufficiently large $\alpha^*$. Similarly, as the maximal degree $k_{\max}$ for nodes contained in the periphery increases, the fraction of intermediate nodes becomes arbitrarily small (Caron et al., 2022, Corollary 5) and makes them negligible for the MF of the whole system.*

### 3.3 GXMFG Approximation for Finite Systems

To compare the finite and limiting GXMFG system, we give some definitions that connect both systems. For an empirical graph $G_\nu$ sampled up to time $\nu$, the corresponding empirical MF is $\hat{\mu}^\nu_{\alpha,t} :=$ $\sum_{i \in V_\nu} \mathbf{1}_{\alpha \in (\frac{i-1}{\sqrt{2|E_\nu|}}, \frac{i}{\sqrt{2|E_\nu|}}]} \delta_{X^i_t}$ and the empirical policies are $\hat{\pi}^\nu_{\alpha,t} := \sum_{i \in V_\nu} \mathbf{1}_{\alpha \in (\frac{i-1}{\sqrt{2|E_\nu|}}, \frac{i}{\sqrt{2|E_\nu|}}]} \pi^i_t$. Furthermore, define the map $\Gamma_\nu(\boldsymbol{\pi}) := (\pi_1, \ldots, \pi_{|V_\nu|}) \in \Pi^{|V_\nu|}$ with $\pi_i = \pi^{\deg(v_i)}_{\alpha(i)}$ if $\deg(v_i) \leq k_{\max}$ and $\pi_i = \pi^\infty_{\alpha(i)}$ otherwise, where $\alpha(i) = i/\sqrt{2|E_\nu|}$ for notational simplicity. If one agents deviates by playing policy $\bar{\pi}_i$ instead of $\pi_i$, denote this by $\Gamma_\nu(\boldsymbol{\pi}, \bar{\pi}_i) := (\pi_1, \ldots, \bar{\pi}_i, \ldots \pi_{|V_\nu|})$. First, we provide the MF convergence result for the high degree core.

**Theorem 1** (Core MF convergence). *Let $\boldsymbol{\pi} \in \boldsymbol{\Pi}$ be Lipschitz up to a finite number of discontinuities with $\boldsymbol{\mu} = \Psi(\boldsymbol{\pi})$. Under Assumptions 1, 2, and the policy $\Gamma_\nu(\boldsymbol{\pi}, \bar{\pi}_i) \in \Pi^{|V_\nu|}$ with $\bar{\pi}_i \in \Pi$, $t \in \mathcal{T}$, for all measurable functions $f \colon \mathcal{X} \times \mathcal{I} \to \mathbb{R}$ uniformly bounded by some $M_f > 0$ and for each $\varepsilon > 0$ there exist some $\nu'(\alpha'), \alpha' > 0$ such that*

$$\mathbb{E}\left[|\hat{\boldsymbol{\mu}}^\nu_t(f, \alpha^*) - \boldsymbol{\mu}^\infty_t(f, \alpha^*)|\right] \leq \varepsilon$$

*holds for all $\nu > \nu'(\alpha')$ and $\alpha^* > \alpha'$ uniformly over all possible deviations $\bar{\pi}_i \in \Pi, i \in V_\nu$.*

A similar result holds for the low degree periphery and connects the two operators defined previously.

**Theorem 2** (Periphery MF convergence). *In the situation as in Theorem 1, for each $\varepsilon > 0$ and finite $k \in \mathbb{N}$ there exist $\alpha', \nu'(\alpha') > 0$ such that*

$$\mathbb{E}\left[\left|\hat{\boldsymbol{\mu}}^{\nu,k}_t(f) - \boldsymbol{\mu}^k_t(f, \nu)\right|\right] \leq \varepsilon$$

*holds for all $\alpha^* > \alpha'$ and $\nu \geq \nu'(\alpha')$ uniformly over all possible deviations $\bar{\pi}_i \in \Pi, i \in V_\nu$.*

Finally, we establish approximate optimality of the GXMFG policies both in the core and periphery.

**Theorem 3** (Core approximate optimality). *Consider a MFCE $(\boldsymbol{\mu}, \boldsymbol{\pi})$ under Assumptions 1 and 2 with $\boldsymbol{\pi}$ Lipschitz up to a finite number of discontinuities. Then, for any $\varepsilon, p > 0$ there exist $\alpha', \nu'(\alpha') > 0$ such that for all $\alpha^* > \alpha', \nu \geq \nu'(\alpha')$ the policy $\Gamma_\nu(\boldsymbol{\pi})$ is an $(\varepsilon, p)$-MNE for the core.*

**Theorem 4** (Periphery approximate optimality). *Under Assumptions 1 and 2, consider an optimal policy ensemble $\boldsymbol{\pi}$ Lipschitz up to a finite number of discontinuities for a MF ensemble $\boldsymbol{\mu}$, i.e. $\pi_\alpha \in \arg\max J^{\boldsymbol{\mu}}_\alpha$ for all $\alpha \in \mathbb{R}_+$. Then, for any $\varepsilon, p > 0$ there exist $\alpha', \nu'(\alpha') > 0$ such that for all $\alpha^* > \alpha', \nu \geq \nu'(\alpha')$ the policy $\Gamma_\nu(\boldsymbol{\pi})$ is a periphery $(\varepsilon, p)$-MNE.*

## 4 Learning Algorithm

The hybrid graphex approach and the corresponding learning algorithm, see Algorithm 1, consist of two parts. Intuitively, the first step of the algorithm is to learn an approximate equilibrium for the high degree core agents. In a second step, this approximate core equilibrium is leveraged to also learn the optimal behavior of the many periphery agents. For approximating the core equilibrium, we utilize the well-established online mirror descent (OMD) algorithm (Pérolat et al., 2022) and emphasize that the theoretical algorithmic guarantees provided by Pérolat et al. (2022) also hold in our case. Therefore, we partition the core parameter interval $[0, \alpha^*]$ into $M$ subintervals of equal length and approximate all agents in each group by the agent at the centre of the respective subinterval. Thus, we obtain a game with $M$ equivalence classes of agents for which an equilibrium is learned via OMD. Here, we choose OMD for its state-of-the-art performance but other learning methods could be incorporated into our framework just as well. In Algorithm 1, the $Q$ function is

$$Q^{\pi,\mu}_{i,t}(x, u) := r(x, u, \mathbb{G}_{i,t}) + \sum_{x' \in \mathcal{X}} P(x' \mid x, u, \mathbb{G}_{i,t}) \arg\max_{u' \in \mathcal{U}} Q^{\pi,\mu}_{i,t+1}(x', u')$$

where $\mathbb{G}_{i,t}$ is the discretized neighborhood. In the second step, it remains to learn the approximately optimal behavior of the periphery agents. We interpret the dynamics in the periphery as a standard Markov decision process (MDP) with reward $r'_k(x, u) := \mathbb{E}\left[r(X_t, U_t, \mathbb{G}^k_t) \mid X_t = x, U_t = u\right]$ and transition kernel $P'_k(x'|x, u) := \mathbb{E}\left[P(x'|X_t, U_t, \mathbb{G}^k_t) \mid X_t = x, U_t = u\right]$. Here, $Q^{k,\mu^{\tau_{\max}}}$ is defined as above with $r'_k$ and $P'_k$ instead of $r$ and $P$. Recall that in the limiting model the periphery does not influence the core agents. Therefore, an action by a periphery agent has no effect on its neighborhood. The reformulation as a standard MDP then allows us to apply the theoretically well-founded calculation of the $Q$ function via backward induction (dynamic programming).

---

**Algorithm 1 Hybrid Online Mirror Descent (HOMD)**

---

1: **First step: solve for the core equilibrium**
2: **Input**: update weight $\gamma$, number of iterations $\tau_{\max}$, number of equivalence classes $M$, $\alpha^*$
3: **Initialization**: $y_{i,t}^0 = 0$ for all $i \leq M, t \in \mathcal{T}$
4: **for** $\tau = 1, \ldots, \tau_{\max}$ **do**
5:     Forward update for all $i$ via equation (1) for given $\pi_i^\tau$: $\mu_i^\tau$
6:     Backward update for all $i$: $Q_{i,t}^{\pi_i^\tau, \mu^\tau}$
7:     Update for all $i, t, x, u$
8:     $y_{i,t}^{\tau+1}(x, u) = y_{i,t}^\tau(x, u) + \gamma \cdot Q_{i,t}^{\pi_i^\tau, \mu^\tau}(x, u)$
9:     $\pi_{i,t}^{\tau+1}(\cdot \mid x) = \text{softmax}\left(y_{t,\tau+1}^i(x, \cdot)\right)$
10: **end for**
11: **Second step: determine the optimal strategies in the periphery**
12: **Input**: maximal degree in the periphery $k_{\max}$
13: **for** $k = 1, \ldots, k_{\max}$ **do**
14:     Backward update for all $k$: $Q^{k, \mu^{\tau_{\max}}}$
15: **end for**

---

## 5 EXAMPLES

In this work, we consider three numerical examples inspired by the existing literature. In each example, a crucial modification is that the agents are now connected by a topology generated by a power law graphex. For more detailed descriptions and problem parameters, see Appendix G.

**Susceptible-Infected-Susceptible (SIS).** The SIS epidemics model is a well-known benchmark problem for learning MFGs (Laurière et al., 2022b; Becherer et al., 2023) and of practical relevance to epidemics control without immunity. Here, agents in the susceptible state ($x = S$) can choose a costly action that prevents costly infection ($x = I$), and the probability to be infected scales both with the number of connections and the fraction of infected neighbors, while the probability of recovery remains constant. Additionally, we consider the **Susceptible-Infected-Recovered (SIR)** model (Laguzet & Turinici, 2015; Lee et al., 2020; Aurell et al., 2022) as an extension of the SIS epidemics model to allow for modelling epidemics with immunity using a third, terminal state $x = R$.

**Rumor Spreading (RS).** Lastly, in a rumor spreading problem as in Cui et al. (2022), agents (e.g., on a social network) randomly contact neighbors and try to spread a rumor to neighbors unaware ($U$) of the rumor for a reputation gain, while avoiding to spread the rumor to aware neighbors ($\bar{U}$).

**Simulation on synthetic networks.** We assume the underlying graphex to be of separable power law form $W(\alpha, \beta) = (1 + \alpha)^{-1/\sigma}(1 + \beta)^{-1/\sigma}$ with $\sigma = 0.5$. In Figure 2, we show the convergence of Algorithm 1 in contrast to naive fixed point iteration (FPI), i.e. the core exploitability

$$\mathcal{E} := \max_{i \leq M} \max_{\pi \in \Pi} \left\{ \sum_{x \in \mathcal{X}} \mu_0(x) \max_{u \in \mathcal{U}} Q_{i,0}^{\pi, \mu^\tau}(x, u) - \sum_{x \in \mathcal{X}} \mu_0(x) \sum_{u \in \mathcal{U}} \pi_{i,0}^\tau(u \mid x) Q_{i,0}^{\pi_i^\tau, \mu^\tau}(x, u) \right\}$$

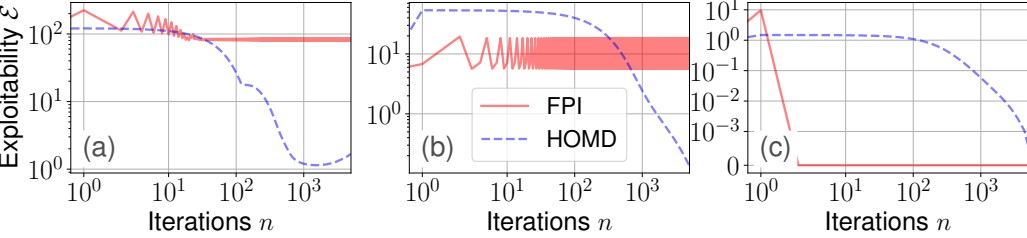

Figure 2: The core exploitability $\mathcal{E}$ is optimized with respect to the iterations of Algorithm 1. (a): SIS; (b): SIR; (c): RS.

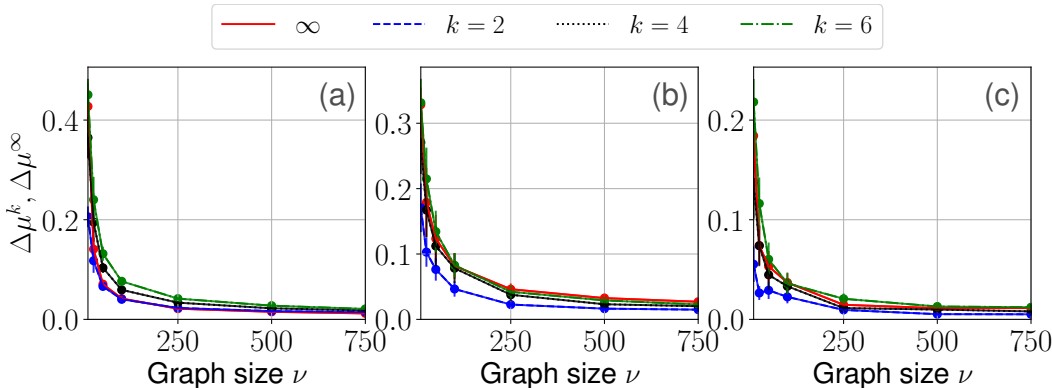

Figure 3: Difference $\Delta\mu^k = \frac{1}{2T}\mathbb{E}\left[\sum_t\|\hat\mu_t^k - \mu_t^k\|_1\right]$ and $\Delta\mu^\infty = \frac{1}{2T}\mathbb{E}\left[\sum_t\|\hat\mu_t^\infty - \mu_t^\infty\|_1\right]$ of the periphery $k$-degree and core MFs against the empirical $k$-degree and $k > k_{\max}$ MFs with respect to $\nu$ for sampled graphs ($\pm$ 95% confidence interval, 20 trials). The parameters $\nu = 10$ and $\nu = 750$ corresponds to approximately $N = 40$ and $N = 26750$ nodes. (a): SIS; (b): SIR; (c): RS.

is optimized which quantifies the limiting MFCE quality. The resulting MFCE is then applied in randomly sampled graphs for fixed $\nu$ by computing the periphery optimal strategies as in the second step of Algorithm 1. As a result of applying the obtained equilibrium in the preceding examples, in Figure 3 we observe the convergence of empirical periphery $k$-degree MFs and core MFs $k > k_{\max}$ for degree cutoff $k_{\max}$ to the limiting MFs predicted by the GXMFG equations in Section 3 as $\nu \to \infty$, validating the derived theoretical framework. For qualitative results, see also the next section and the following Figure 4.

## 6 SIMULATION ON REAL NETWORKS

Beyond synthetic data, we find that our framework is capable of producing good equilibria in real world networks. Under the separable power law form, for each empirical data set we estimate $\hat\sigma$ with the procedure proposed by Naulet et al. (2021). Although this simple estimation yields reasonable evaluation results, we believe that developing more complex estimators would also further increase the accuracy of our GXMFG learning method. If data sets contain multiple or directed edges, we substitute them by one undirected edge to obtain simple, undirected graphs. We consider the following real world networks from the KONECT database (Kunegis, 2013): Prosper loans (Redmond & Cunningham, 2013), Dogster (Kunegis, 2013), Pokec (Takac & Zabovsky, 2012), Livemocha (Zafarani & Liu, 2009), Flickr (Mislove et al., 2007), Brightkite (Cho et al., 2011),

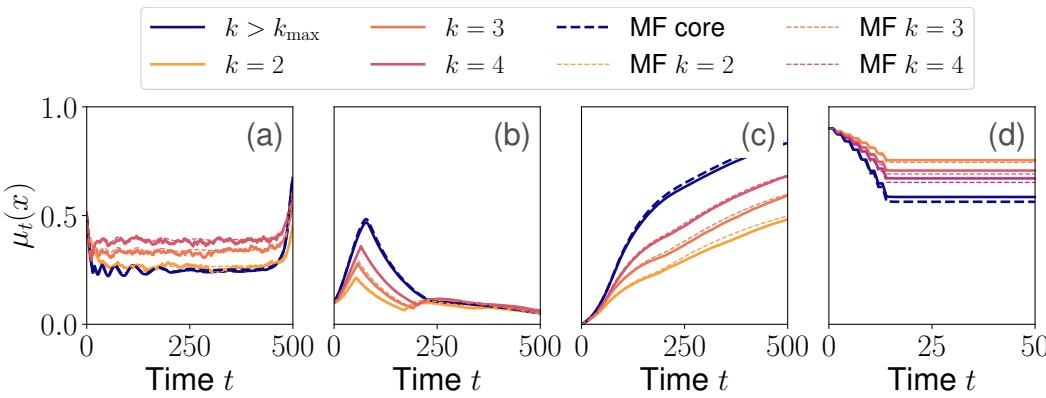

Figure 4: Comparison of the empirically observed MF on a real network with the predicted equilibrium for (a) SIS: $x = I$ on Prosper; (b-c) SIR: $x = I$ and $x = R$ on Dogster; and (d) RS: $x = U$ on Pokec.

Table 1: Expected average total variation $\Delta\mu = \frac{1}{2T}\mathbb{E}\left[\sum_t \|\hat{\mu}_t - \mu_t\|_1\right] \in [0,1]$ between the overall GXMFG MF prediction $\mu_t$ and the empirical MF $\hat{\mu}_t = \sum_i \delta_{X_t^i}$ ($\pm$ standard deviation, 5 trials).

| network | $\hat{\sigma}$ | # nodes | # edges | Expected total variation $\Delta\mu$ in % | | |
|---------|------|---------|---------|-----------|----------|---------|
| | | | | SIS (%) | SIR (%) | RS (%) |
| Prosper | 0.058 | 89269 | 3.3 MM | $0.53 \pm 0.21$ | $2.06 \pm 0.25$ | $1.58 \pm 0.31$ |
| Dogster | 0.071 | 426820 | 8.5 MM | $2.78 \pm 0.44$ | $4.93 \pm 0.98$ | $1.89 \pm 0.66$ |
| Pokec | 0.108 | 1.6 MM | 22.3 MM | $2.14 \pm 0.05$ | $4.93 \pm 0.12$ | $2.19 \pm 0.08$ |
| Livemocha | 0.075 | 104103 | 2.2 MM | $2.57 \pm 0.20$ | $4.40 \pm 1.08$ | $2.51 \pm 0.54$ |
| Flickr | 0.506 | 2.3 MM | 22.8 MM | $3.57 \pm 0.08$ | $8.58 \pm 0.24$ | $2.24 \pm 0.11$ |
| Brightkite | 0.376 | 58228 | 214078 | $1.37 \pm 0.11$ | $10.92 \pm 0.77$ | $3.37 \pm 0.25$ |
| Facebook | 0.252 | 46952 | 188453 | $2.90 \pm 0.09$ | $13.57 \pm 0.37$ | $5.01 \pm 0.27$ |
| Hyves | 0.585 | 1.4 MM | 2.8 MM | $5.07 \pm 0.24$ | $10.06 \pm 0.81$ | $2.62 \pm 0.44$ |

Facebook (Viswanath et al., 2009), and Hyves (Zafarani & Liu, 2009). See the respective papers for detailed network descriptions.

The plots in Figure 4 give a qualitative impression of the algorithmic performance of our approach on real world networks and show that the estimated equilibrium is close to the empirical MF on the real networks. The quantitative empirical results can be found in Table 1 where the overall predicted GXMFG mean field is $\mu = \sum_k p_k \mu^k + (1 - \sum_k p_k)\mu^\infty$ with $p_k = \hat{\sigma}\Gamma(k - \hat{\sigma})/(k!\Gamma(1 - \hat{\sigma}))$, see Caron et al. (2022, Section 6.3) for the choice of $p_k$. Similar results are shown for the periphery mean fields, see Table 2 in Appendix G. Furthermore, our GXMFG learning approach clearly outperforms LPGMFGs (Fabian et al., 2023) (and thereby also the subset of GMFGs) on all of the considered tasks and real world networks, see Table 3 in Appendix G for details. These results indicate that the ability of GXMFGs to depict sparse networks with finite degree agents is a crucial advantage over existing GMFG and LPGMFG models, both conceptually and empirically.

In Table 1, each row shows the estimated graphex parameter $\hat{\sigma}$ and the number of nodes and edges for each real world network. The last three columns contain the (normalized) expected total variation $\Delta\mu$ which measures how close the empirical mean field $\hat{\mu}$ is to the predicted mean field $\mu$ of the graphex learning approach for each of the three examplary models SIS, SIR and RS. One can see that the accuracy of the hybrid graphex learning algorithm is somewhere between a deviation of 1.5% up to 5% for most of the real world networks and examples. While the MFs for SIS and RS are learned accurately ($\Delta\mu < 5.1\%$) on all networks, the SIR approximation performs well on the Prosper, Dogster, Pokec, and Livemotcha networks. A possible explanation for the under average performance on the Brightkite and Facebook data sets could be their rather small number of nodes which decreases the accuracy of our asymptotic approximation. Additionally, some networks such as Hyves might not be represented optimally by a separable power law graphex. We think that proposing more evolved approximation schemes and thereby allowing for more diverse graphexes would improve the performance of our algorithm on these networks, which we leave to future work.

## 7 CONCLUSION

In this paper we have introduced the concept of graphex mean field games. To learn this challenging class of games, we have developed a novel hybrid graphex learning approach to address the high complexity of sparse real world networks. Our empirical examples have shown that learning GXMFGs could be a promising approach to understanding behavior in large and complex agent networks. For future work it would be interesting to develop and use more precise graphex estimators and to apply our setup to various research problems. Furthermore, an extension to continuous state and action spaces, as well as continuous time, could be a next step. We hope that our work contributes to the applicability and accuracy of mean field games for real world challenges.

ACKNOWLEDGMENTS

This work has been co-funded by the Hessian Ministry of Science and the Arts (HMWK) within the projects "The Third Wave of Artificial Intelligence - 3AI" and hessian.AI, and the LOEWE initiative (Hesse, Germany) within the emergenCITY center. The authors acknowledge the Lichtenberg high performance computing cluster of the TU Darmstadt for providing computational facilities for the calculations of this research. We thank the anonymous reviewers for their helpful comments on improving the manuscript.

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

## A  OVERVIEW

On the following pages, we often write $\varepsilon_{\alpha^*}$ to express an error term depending on $\alpha^*$ such that $\lim_{\alpha^* \to \infty} \varepsilon_{\alpha^*} = 0$. The following lemma will be of frequent help to us for proving various convergence results. In the subsequent sections, we often just refer to Assumption 1 but tacitly also mean its implication formalized by the next lemma.

**Lemma 2.** *Assumption 1 implies* $\sup_{-1 \le g \le 1} \int_{\mathbb{R}_+} \left| \int_{\mathbb{R}_+} \left( W(\alpha, \beta) - \widehat{W}_\nu(\alpha, \beta) \right) g(\beta) \, \mathrm{d}\beta \right| \mathrm{d}\alpha \to 0$ *as* $\nu \to \infty$ *where the supremum is taken over all measurable functions* $g : \mathbb{R}_+ \to [-1, 1]$.

*Proof of Lemma 2.* To show the statement, we adapt a proof strategy in Lovász (2012, proof of Lemma 8.11) to our case. We reformulate

$$
\sup_{-1 \le g \le 1} \int_{\mathbb{R}_+} \left| \int_{\mathbb{R}_+} \left( W(\alpha, \beta) - \widehat{W}_\nu(\alpha, \beta) \right) g(\beta) \, \mathrm{d}\beta \right| \mathrm{d}\alpha
$$

$$
= \sup_{-1 \le f, g \le 1} \int_{\mathbb{R}_+} \int_{\mathbb{R}_+} \left( W(\alpha, \beta) - \widehat{W}_\nu(\alpha, \beta) \right) f(\alpha) g(\beta) \, \mathrm{d}\beta \, \mathrm{d}\alpha
$$

$$
= \sup_{0 \le f, f', g, g' \le 1} \int_{\mathbb{R}_+} \int_{\mathbb{R}_+} \left( W(\alpha, \beta) - \widehat{W}_\nu(\alpha, \beta) \right) (f(\alpha) - f'(\alpha))(g(\beta) - g'(\beta)) \, \mathrm{d}\beta \, \mathrm{d}\alpha
$$

$$
= \sup_{0 \le f, f', g, g' \le 1} \int_{\mathbb{R}_+} \int_{\mathbb{R}_+} \left( W(\alpha, \beta) - \widehat{W}_\nu(\alpha, \beta) \right) f(\alpha) g(\beta) \, \mathrm{d}\beta \, \mathrm{d}\alpha
$$

$$
- \int_{\mathbb{R}_+} \int_{\mathbb{R}_+} \left( W(\alpha, \beta) - \widehat{W}_\nu(\alpha, \beta) \right) f(\alpha) g'(\beta) \, \mathrm{d}\beta \, \mathrm{d}\alpha
$$

$$
- \int_{\mathbb{R}_+} \int_{\mathbb{R}_+} \left( W(\alpha, \beta) - \widehat{W}_\nu(\alpha, \beta) \right) f'(\alpha) g(\beta) \, \mathrm{d}\beta \, \mathrm{d}\alpha
$$

$$
+ \int_{\mathbb{R}_+} \int_{\mathbb{R}_+} \left( W(\alpha, \beta) - \widehat{W}_\nu(\alpha, \beta) \right) f'(\alpha) g'(\beta) \, \mathrm{d}\beta \, \mathrm{d}\alpha
$$

$$
\le 4 \| W - \widehat{W}_\nu \|_\square = 4 \sup_{U, V} \left| \int_{U \times V} W(\alpha, \beta) - \widehat{W}_\nu(\alpha, \beta) \, \mathrm{d}\alpha \mathrm{d}\beta \right| \to 0 \quad \text{as} \quad \nu \to \infty
$$

where the inequality in the last line follows from Janson (2016, equation 2.5) and the convergence is a consequence of Assumption 1. $\qquad \square$

We continue with proving the first theoretical statement from the main paper.

*Proof of Lemma 1.* Since in the limiting mean field core system we 'cut off' the graphex at parameter $\alpha^*$, we can think of the remaining part of the graphex as a standard graphon, where we point to Veitch & Roy (2015, Section 3.1 and Theorem 5.6) for a detailed argument. Thus, the limiting core system can be interpreted as a standard graphon mean field game which means that the existence of a mean field core equilibrium follows from Cui & Koeppl (2021b, Theorem 1). $\qquad \square$

## B  PROOF OF THEOREM 1

We prove the claim by induction over $t = 0$. The induction start follows from a law of large numbers argument similar to the one for the first term below. For the induction, we leverage the inequality

$$
\mathbb{E}\left[ \left| \hat{\boldsymbol{\mu}}_{t+1}^\nu(f, \alpha^*) - \boldsymbol{\mu}_{t+1}^\infty(f, \alpha^*) \right| \right] \le \mathbb{E}\left[ \left| \hat{\boldsymbol{\mu}}_{t+1}^\nu(f, \alpha^*) - \hat{\boldsymbol{\mu}}_t^\nu P_{t, \hat{\boldsymbol{\mu}}, \widehat{W}}^{\hat{\boldsymbol{\pi}}, \infty}(f, \alpha^*) \right| \right]
$$

$$
+ \mathbb{E}\left[ \left| \hat{\boldsymbol{\mu}}_t^\nu P_{t, \hat{\boldsymbol{\mu}}, \widehat{W}}^{\hat{\boldsymbol{\pi}}, \infty}(f, \alpha^*) - \hat{\boldsymbol{\mu}}_t^\nu P_{t, \hat{\boldsymbol{\mu}}, W}^{\hat{\boldsymbol{\pi}}, \infty}(f, \alpha^*) \right| \right]
$$

$$
+ \mathbb{E}\left[ \left| \hat{\boldsymbol{\mu}}_t^\nu P_{t, \hat{\boldsymbol{\mu}}, W}^{\hat{\boldsymbol{\pi}}, \infty}(f, \alpha^*) - \hat{\boldsymbol{\mu}}_t^\nu P_{t, \hat{\boldsymbol{\mu}}, W}^{\boldsymbol{\pi}, \infty}(f, \alpha^*) \right| \right]
$$

$$
+ \mathbb{E}\left[ \left| \hat{\boldsymbol{\mu}}_t^\nu P_{t, \hat{\boldsymbol{\mu}}, W}^{\boldsymbol{\pi}, \infty}(f, \alpha^*) - \hat{\boldsymbol{\mu}}_t^\nu P_{t, \boldsymbol{\mu}, W}^{\boldsymbol{\pi}, \infty}(f, \alpha^*) \right| \right]
$$

$$+ \mathbb{E}\left[\left|\hat{\boldsymbol{\mu}}_t^\nu P_{t,\boldsymbol{\mu},W}^{\boldsymbol{\pi},\infty}(f,\alpha^*) - \boldsymbol{\mu}_{t+1}^\infty(f,\alpha^*)\right|\right] .$$

In the following, we upper bound each of the five terms separately.

**First term.** We start with the first term

$$\mathbb{E}\left[\left|\hat{\boldsymbol{\mu}}_{t+1}^\nu(f,\alpha^*) - \hat{\boldsymbol{\mu}}_t^\nu P_{t,\hat{\boldsymbol{\mu}},\widehat{W}}^{\hat{\boldsymbol{\pi}},\infty}(f,\alpha^*)\right|\right]$$

$$= \frac{1}{\alpha^*} \cdot \mathbb{E}\left[\left|\int_0^{\alpha^*} \sum_{x'\in\mathcal{X}} f(x',\alpha)\hat{\mu}_{\alpha,t+1}^\nu(x')\,\mathrm{d}\alpha - \int_0^{\alpha^*} \sum_{x'\in\mathcal{X}}\sum_{x\in\mathcal{X}} \hat{\mu}_{\alpha,t}^\nu(x)\right.\right.$$

$$\left.\left.\cdot \sum_{u\in\mathcal{U}} \hat{\pi}_{\alpha,t}(u\mid x) \cdot P\left(x'\mid x,u,\frac{1}{\xi_{\widehat{W},\alpha^*}(\alpha)}\int_0^{\alpha^*}\widehat{W}(\alpha,\beta)\hat{\mu}_{\beta,t}^\nu\,\mathrm{d}\beta\right)f(x',\alpha)\,\mathrm{d}\alpha\right|\right]$$

$$= \frac{\varepsilon_{\alpha^*}L_P}{\alpha^*} + \frac{1}{\alpha^*} \cdot \mathbb{E}\left[\left|\int_0^{\alpha^*} \sum_{x'\in\mathcal{X}} f(x',\alpha)\hat{\mu}_{\alpha,t+1}^\nu(x')\,\mathrm{d}\alpha - \int_0^{\alpha^*} \sum_{x'\in\mathcal{X}}\sum_{x\in\mathcal{X}} \hat{\mu}_{\alpha,t}^\nu(x)\right.\right.$$

$$\left.\left.\cdot \sum_{u\in\mathcal{U}} \hat{\pi}_{\alpha,t}(u\mid x) \cdot P\left(x'\mid x,u,\frac{1}{\xi_{\widehat{W}}(\alpha)}\int_{\mathbb{R}_+}\widehat{W}(\alpha,\beta)\hat{\mu}_{\beta,t}^\nu\,\mathrm{d}\beta\right)f(x',\alpha)\,\mathrm{d}\alpha\right|\right]$$

$$= \frac{\varepsilon_{\alpha^*}L_P}{\alpha^*} + \frac{1}{\alpha^*} \cdot \mathbb{E}\left[\left|\sum_{\substack{i\in V_\nu \\ i<\alpha^*\sqrt{2|E_\nu|}}} \left(\int_{\left(\frac{i-1}{\sqrt{2|E_\nu|}},\frac{i}{\sqrt{2|E_\nu|}}\right]} f(X_{t+1}^i,\alpha)\,\mathrm{d}\alpha\right.\right.\right.$$

$$\left.\left.\left.- \mathbb{E}\left[\int_{\left(\frac{i-1}{\sqrt{2|E_\nu|}},\frac{i}{\sqrt{2|E_\nu|}}\right]} f(X_{t+1}^i,\alpha)\,\mathrm{d}\alpha \,\middle|\, \mathbf{X}_t\right]\right)\right|\right]$$

$$\leq \frac{\varepsilon_{\alpha^*}L_P}{\alpha^*} + \frac{1}{\alpha^*} \cdot \mathbb{E}\left[\sum_{\substack{i\in V_\nu \\ i<\alpha^*\sqrt{2|E_\nu|}}} \left(\int_{\left(\frac{i-1}{\sqrt{2|E_\nu|}},\frac{i}{\sqrt{2|E_\nu|}}\right]} f(X_{t+1}^i,\alpha)\,\mathrm{d}\alpha\right.\right.$$

$$\left.\left.- \mathbb{E}\left[\int_{\left(\frac{i-1}{\sqrt{2|E_\nu|}},\frac{i}{\sqrt{2|E_\nu|}}\right]} f(X_{t+1}^i,\alpha)\,\mathrm{d}\alpha \,\middle|\, \mathbf{X}_t\right]\right)^2\right]^{\frac{1}{2}}$$

$$\leq \frac{\varepsilon_{\alpha^*}L_P}{\alpha^*} + \frac{1}{\alpha^*} \cdot \left(\sum_{\substack{i\in V_\nu \\ i<\alpha^*\sqrt{2|E_\nu|}}} \frac{\max_{(x,\alpha)\in\mathcal{X}\times\left(\frac{i-1}{\sqrt{2|E_\nu|}},\frac{i}{\sqrt{2|E_\nu|}}\right]} f(x,\alpha)^2}{|E_\nu|}\right)^{\frac{1}{2}}$$

$$\leq \frac{\varepsilon_{\alpha^*}L_P}{\alpha^*} + \underbrace{\frac{1}{\alpha^*} \cdot \left(\frac{M_f^2\alpha^*\sqrt{2|E_\nu|}}{|E_\nu|}\right)^{\frac{1}{2}}}_{\to 0 \text{ for } \nu\to\infty} \leq \varepsilon ,$$

for $\alpha^*$ and $\nu$ large enough, where we exploited the fact that $\{X_{t+1}^i\}_{i\in V_\nu}$ are independent if conditioned on $\mathbf{X}_t \equiv \{X_t^i\}_{i\in V_\nu}$.

**Second term.** For the second term we have

$$\mathbb{E}\left[\left|\hat{\boldsymbol{\mu}}_t^\nu P_{t,\hat{\boldsymbol{\mu}},\widehat{W}}^{\hat{\boldsymbol{\pi}},\infty}(f,\alpha^*) - \hat{\boldsymbol{\mu}}_t^\nu P_{t,\hat{\boldsymbol{\mu}},W}^{\hat{\boldsymbol{\pi}},\infty}(f,\alpha^*)\right|\right]$$

$$= \frac{1}{\alpha^*}\mathbb{E}\left[\left|\int_0^{\alpha^*} \sum_{x,x'\in\mathcal{X}}\sum_{u\in\mathcal{U}} \hat{\mu}_{\alpha,t}^\nu(x)\hat{\pi}_{\alpha,t}(u\mid x)\right.\right.$$

$$\cdot P\left(x' \mid x, u, \frac{1}{\xi_{\widehat{W},\alpha^*}(\alpha)} \int_0^{\alpha^*} \widehat{W}(\alpha,\beta)\hat{\mu}_{\beta,t}^\nu \, \mathrm{d}\beta\right) f(x',\alpha)$$

$$-\sum_{x,x'\in\mathcal{X}} \hat{\mu}_{\alpha,t}^\nu(x) \sum_{u\in\mathcal{U}} \hat{\pi}_{\alpha,t}(u \mid x) P\left(x' \mid x, u, \frac{1}{\xi_{W,\alpha^*}(\alpha)} \int_0^{\alpha^*} W(\alpha,\beta)\hat{\mu}_{\beta,t}^\nu \, \mathrm{d}\beta\right) f(x',\alpha)\,\mathrm{d}\alpha\Bigg|\Bigg]$$

$$\leq \frac{L_P(1+o(1))}{\alpha^*\xi_{W,\alpha^*}(\alpha^*)} \mathbb{E}\left[\int_0^{\alpha^*} \left\|\int_0^{\alpha^*} \widehat{W}(\alpha,\beta)\hat{\mu}_{\beta,t}^\nu \, \mathrm{d}\beta - \int_0^{\alpha^*} W(\alpha,\beta)\hat{\mu}_{\beta,t}^\nu \, \mathrm{d}\beta\right\| \sum_{x'\in\mathcal{X}} f(x',\alpha)\,\mathrm{d}\alpha\right]$$

$$= \frac{L_P|\mathcal{X}|M_f(1+o(1))}{\alpha^*\xi_{W,\alpha^*}(\alpha^*)} \mathbb{E}\left[\int_0^{\alpha^*} \left\|\int_0^{\alpha^*} \widehat{W}(\alpha,\beta)\hat{\mu}_{\beta,t}^\nu \, \mathrm{d}\beta - \int_0^{\alpha^*} W(\alpha,\beta)\hat{\mu}_{\beta,t}^\nu \, \mathrm{d}\beta\right\| \mathrm{d}\alpha\right]$$

$$\leq \frac{L_P|\mathcal{X}|M_f(1+o(1))}{\alpha^*\xi_{W,\alpha^*}(\alpha^*)} \max_{x\in\mathcal{X}} \mathbb{E}\left[\int_0^{\alpha^*} \left|\int_0^{\alpha^*} \widehat{W}(\alpha,\beta)\hat{\mu}_{\beta,t}^\nu(x) - W(\alpha,\beta)\hat{\mu}_{\beta,t}^\nu(x) \, \mathrm{d}\beta\right| \mathrm{d}\alpha\right]$$

which converges to $0$ as $\nu \to \infty$, where the convergence follows from Assumption 1 and $\hat{\mu}_{\beta,t}^\nu$ being bounded by $1$ by construction.

**Third term.** The third term converges to $0$ by the following argument:

$$\mathbb{E}\left[\left|\hat{\boldsymbol{\mu}}_t^\nu P_{t,\hat{\boldsymbol{\mu}},W}^{\hat{\boldsymbol{\pi}},\infty}(f,\alpha^*) - \hat{\boldsymbol{\mu}}_t^\nu P_{t,\hat{\boldsymbol{\mu}},W}^{\boldsymbol{\pi},\infty}(f,\alpha^*)\right|\right]$$

$$= \frac{1}{\alpha^*} \mathbb{E}\left[\left|\int_0^{\alpha^*} \sum_{x,x'\in\mathcal{X}} \sum_{u\in\mathcal{U}} \hat{\mu}_{\alpha,t}^\nu(x)\hat{\pi}_{\alpha,t}(u \mid x)\right.\right.$$

$$\cdot P\left(x' \mid x, u, \frac{1}{\xi_{W,\alpha^*}(\alpha)} \int_0^{\alpha^*} W(\alpha,\beta)\hat{\mu}_{\beta,t}^\nu \, \mathrm{d}\beta\right) f(x',\alpha)$$

$$-\sum_{x,x'\in\mathcal{X}} \hat{\mu}_{\alpha,t}^\nu(x) \sum_{u\in\mathcal{U}} \pi_{\alpha,t}^\infty(u \mid x) P\left(x' \mid x, u, \frac{1}{\xi_{W,\alpha^*}(\alpha)} \int_0^{\alpha^*} W(\alpha,\beta)\hat{\mu}_{\beta,t}^\nu \, \mathrm{d}\beta\right) f(x',\alpha)\,\mathrm{d}\alpha\Bigg|\Bigg]$$

$$\leq \frac{M_f|\mathcal{X}||\mathcal{U}|}{\alpha^*} \max_{(x,u)\in\mathcal{X}\times\mathcal{U}} \mathbb{E}\left[\int_0^{\alpha^*} \left|\hat{\pi}_{\alpha,t}(u \mid x) - \pi_{\alpha,t}^\infty(u \mid x)\right| \mathrm{d}\alpha\right]$$

$$= \frac{M_f|\mathcal{X}||\mathcal{U}|}{\alpha^*} \max_{(x,u)\in\mathcal{X}\times\mathcal{U}} \mathbb{E}\left[\sum_{\substack{j\in V_\nu, j\neq i \\ j < \alpha^*\sqrt{2|E_\nu|}}} \int_{\frac{j-1}{\sqrt{2|E_\nu|}}}^{\frac{j}{\sqrt{2|E_\nu|}}} \left|\hat{\pi}_{\alpha,t}(u \mid x) - \pi_{\alpha,t}^\infty(u \mid x)\right| \mathrm{d}\alpha\right]$$

$$+ \frac{M_f|\mathcal{X}||\mathcal{U}|}{\alpha^*} \max_{(x,u)\in\mathcal{X}\times\mathcal{U}} \mathbb{E}\left[\int_{\frac{i-1}{\sqrt{2|E_\nu|}}}^{\frac{i}{\sqrt{2|E_\nu|}}} \left|\hat{\pi}_{\alpha,t}(u \mid x) - \pi_{\alpha,t}^\infty(u \mid x)\right| \mathrm{d}\alpha\right]$$

$$\leq \frac{M_f|\mathcal{X}||\mathcal{U}|}{\alpha^*} \cdot \left(\frac{\alpha^*}{\sqrt{2|E_\nu|}} + \frac{2D_\pi}{\sqrt{2|E_\nu|}} + \frac{2}{\sqrt{2|E_\nu|}}\right) \to 0 \qquad \text{as } \nu \to \infty.$$

Here, $D_\pi$ denotes the finite number of discontinuities of $\pi$.

**Fourth term.** We have

$$\mathbb{E}\left[\left|\hat{\boldsymbol{\mu}}_t^\nu P_{t,\hat{\boldsymbol{\mu}},W}^{\boldsymbol{\pi},\infty}(f,\alpha^*) - \hat{\boldsymbol{\mu}}_t^\nu P_{t,\boldsymbol{\mu},W}^{\boldsymbol{\pi},\infty}(f,\alpha^*)\right|\right]$$

$$= \frac{1}{\alpha^*} \mathbb{E}\left[\left|\int_0^{\alpha^*} \sum_{x,x'\in\mathcal{X}} \hat{\mu}_{\alpha,t}^\nu(x) \sum_{u\in\mathcal{U}} \pi_{\alpha,t}^\infty(u \mid x)\right.\right.$$

$$\cdot P\left(x' \mid x, u, \frac{1}{\xi_{W,\alpha^*}(\alpha)} \int_0^{\alpha^*} W(\alpha,\beta)\hat{\mu}_{\beta,t}^\nu \, \mathrm{d}\beta\right) f(x',\alpha)$$

$$- \sum_{\substack{x,x'\in\mathcal{X} \\ u\in\mathcal{U}}} \hat{\mu}^{\nu}_{\alpha,t}(x)\pi^{\infty}_{\alpha,t}(u\mid x)\, P\left(x'\mid x,u,\frac{1}{\xi_{W,\alpha^*}(\alpha)}\int_0^{\alpha^*}W(\alpha,\beta)\mu_{\beta,t}\,\mathrm{d}\beta\right)f(x',\alpha)\,\mathrm{d}\alpha\Bigg|\Bigg]$$

$$\leq \frac{M_f L_P |\mathcal{X}|}{\alpha^* \xi_{W,\alpha^*}(\alpha^*)}\,\mathbb{E}\left[\int_0^{\alpha^*}\sum_{x'\in\mathcal{X}}\left|\int_0^{\alpha^*}W(\alpha,\beta)\hat{\mu}^{\nu}_{\beta,t}(x')\,\mathrm{d}\beta - \int_0^{\alpha^*}W(\alpha,\beta)\mu_{\beta,t}(x')\,\mathrm{d}\beta\right|\,\mathrm{d}\alpha\right]$$

$$\leq \frac{M_f L_P |\mathcal{X}|^2}{\alpha^* \xi_{W,\alpha^*}(\alpha^*)}\max_{x'\in X}\int_0^{\alpha^*}\mathbb{E}\left[\left|\int_0^{\alpha^*}W(\alpha,\beta)\hat{\mu}^{\nu}_{\beta,t}(x')\,\mathrm{d}\beta - \int_0^{\alpha^*}W(\alpha,\beta)\mu_{\beta,t}(x')\,\mathrm{d}\beta\right|\right]\,\mathrm{d}\alpha$$

which goes to zero as $\nu \to \infty$. This convergence follows from the induction assumption. To see this, we consider the functions $f'_{x',\alpha}(x,\beta) := W(\alpha,\beta)\cdot\mathbf{1}_{\{x=x'\}}$ which, combined with the induction assumption and Fubini's theorem, yields

$$\frac{1}{\alpha^*}\int_0^{\alpha^*}\mathbb{E}\left[\left|\int_0^{\alpha^*}W(\alpha,\beta)\hat{\mu}^{\nu}_{\beta,t}(x')\,\mathrm{d}\beta - \int_0^{\alpha^*}W(\alpha,\beta)\mu_{\beta,t}(x')\,\mathrm{d}\beta\right|\right]\,\mathrm{d}\alpha$$

$$= \int_0^{\alpha^*}\mathbb{E}\left[\left|\hat{\boldsymbol{\mu}}^{\nu}_t(f'_{x',\alpha},\alpha^*) - \boldsymbol{\mu}^{\infty}_t(f'_{x',\alpha},\alpha^*)\right|\right]\,\mathrm{d}\alpha \to 0 \quad \text{as } \nu \to \infty\,.$$

**Fifth term.** Finally, for the fifth term we know that

$$\mathbb{E}\left[\left|\hat{\boldsymbol{\mu}}^{\nu}_t P^{\boldsymbol{\pi},\infty}_{t,\boldsymbol{\mu},W}(f,\alpha^*) - \boldsymbol{\mu}^{\infty}_{t+1}(f,\alpha^*)\right|\right]$$

$$= \mathbb{E}\Bigg[\Bigg|\frac{1}{\alpha^*}\int_0^{\alpha^*}\sum_{\substack{x,x'\in\mathcal{X}\\u\in\mathcal{U}}}\hat{\mu}^{\nu}_{\alpha,t}(x)\pi^{\infty}_{\alpha,t}(u\mid x)$$

$$\cdot P\left(x'\mid x,u,\frac{1}{\xi_{W,\alpha^*}(\alpha)}\int_0^{\alpha^*}W(\alpha,\beta)\mu_{\beta,t}\,\mathrm{d}\beta\right)f(x',\alpha)\,\mathrm{d}\alpha$$

$$-\frac{1}{\alpha^*}\int_0^{\alpha^*}\sum_{\substack{x,x'\in\mathcal{X}\\u\in\mathcal{U}}}\mu_{\alpha,t}(x)\pi^{\infty}_{\alpha,t}(u\mid x)$$

$$\cdot P\left(x'\mid x,u,\frac{1}{\xi_{W,\alpha^*}(\alpha)}\int_0^{\alpha^*}W(\alpha,\beta)\mu_{\beta,t}\,\mathrm{d}\beta\right)f(x',\alpha)\,\mathrm{d}\alpha\Bigg|\Bigg]$$

$$\leq \frac{\alpha^*}{\xi_{W,\alpha^*}(\alpha^*)}\,\mathbb{E}\left[\left|\frac{1}{\alpha^*}\int_0^{\alpha^*}\sum_{x\in\mathcal{X}}\hat{\mu}^{\nu}_{\alpha,t}(x)f'(x,\alpha)\,\mathrm{d}\alpha - \frac{1}{\alpha^*}\int_0^{\alpha^*}\sum_{x\in\mathcal{X}}\mu_{\alpha,t}(x)f'(x,\alpha)\,\mathrm{d}\alpha\right|\right]$$

$$= \frac{\alpha^*}{\xi_{W,\alpha^*}(\alpha^*)}\,\mathbb{E}\left[\left|\hat{\boldsymbol{\mu}}^{\nu}_t(f',\alpha^*) - \boldsymbol{\mu}^{\infty}_t(f',\alpha^*)\right|\right] \leq \varepsilon$$

where we define the function

$$f'(x,\alpha) := \sum_{x'\in\mathcal{X}}\sum_{u\in\mathcal{U}}\pi^{\infty}_{\alpha,t}(u\mid x)\cdot P\left(x'\mid x,u,\frac{1}{\alpha^*}\int_0^{\alpha^*}W(\alpha,\beta)\mu^{\beta}_t\,\mathrm{d}\beta\right)f(x',\alpha)$$

and apply the induction assumption. This concludes the proof by induction.

## C  CALCULATION OF THE NEIGHBORHOOD PROBABILITY DISTRIBUTION

**Binary state space case.** Consider some uniformly, at random picked node $v_0$ with degree $k$, latent parameter $\alpha$ and state $x_0$. Furthermore, denote by $x_1,\ldots,x_k$ the states of its $k$ neighbors. Let $j$ be the number of neighbors in the first state $s_1$.

$$P^{\nu}_{\boldsymbol{\pi},\boldsymbol{\mu}}\left(\mathbb{G}^{\nu,k}_{0,t+1}(\hat{\boldsymbol{\mu}}^{\nu}_t) = j, x_{0,t+1} = x\right)$$

$$= \sum_{x' \in \mathcal{X}} \sum_{j'=0}^{k} P_{\boldsymbol{\pi}, \boldsymbol{\mu}}^{\nu} \left( \mathbb{G}_{0,t+1}^{\nu,k} (\hat{\boldsymbol{\mu}}_t^{\nu}) = j, \mathbb{G}_{0,t}^{\nu,k} = j', x_{0,t} = x', x_{0,t+1} = x \right)$$

$$= \sum_{x' \in \mathcal{X}} \sum_{j'=0}^{k} P_{\boldsymbol{\pi}, \boldsymbol{\mu}}^{\nu} \left( \mathbb{G}_{0,t}^{\nu,k} (\hat{\boldsymbol{\mu}}_t^{\nu}) = j', x_{0,t} = x' \right) \cdot P_{\boldsymbol{\pi}, \boldsymbol{\mu}}^{\nu} \left( x_{0,t+1} = x \mid \mathbb{G}_{0,t}^{\nu,k} (\hat{\boldsymbol{\mu}}_t^{\nu}) = j', x_{0,t} = x' \right)$$

$$\cdot P_{\boldsymbol{\pi}}^{\nu} \left( \mathbb{G}_{0,t+1}^{\nu,k} (\hat{\boldsymbol{\mu}}_t^{\nu}) = j \mid x_{0,t+1} = x, \mathbb{G}_{0,t}^{\nu,k} = j', x_{0,t} = x' \right)$$

$$= \sum_{x' \in \mathcal{X}} \sum_{j'=0}^{k} P_{\boldsymbol{\pi}, \boldsymbol{\mu}}^{\nu} \left( \mathbb{G}_{0,t}^{\nu,k} (\hat{\boldsymbol{\mu}}_t^{\nu}) = j', x_{0,t} = x' \right) \sum_{u \in \mathcal{U}} \pi_{0,t}^{k} (u \mid x') \cdot P \left( x \mid x', u, \frac{j'}{k} \right)$$

$$\cdot P_{\boldsymbol{\pi}}^{\nu} \left( \mathbb{G}_{0,t+1}^{\nu,k} (\hat{\boldsymbol{\mu}}_t^{\nu}) = j \mid \mathbb{G}_{0,t}^{\nu,k} (\hat{\boldsymbol{\mu}}_t^{\nu}) = j', x_{0,t} = x' \right) .$$

Now, we consider

$$P_{\boldsymbol{\pi}}^{\nu} \left( \mathbb{G}_{0,t+1}^{\nu,k} (\hat{\boldsymbol{\mu}}_t^{\nu}) = j \mid \mathbb{G}_{0,t}^{\nu,k} (\hat{\boldsymbol{\mu}}_t^{\nu}) = j', x_{0,t} = x' \right)$$

$$= P_{\boldsymbol{\pi}}^{\nu} \left( \mathbb{G}_{0,t+1}^{\nu,k} (\hat{\boldsymbol{\mu}}_t^{\nu}) = j \mid \mathbb{G}_{0,t}^{\nu,k} (\hat{\boldsymbol{\mu}}_t^{\nu}) = j' \right) + o(1)$$

$$= o(1) + \int_{\mathbb{R}_+^k} P_{\boldsymbol{\pi}}^{\nu} \left( \mathbb{G}_{0,t+1}^{\nu,k} (\hat{\boldsymbol{\mu}}_t^{\nu}) = j \mid \mathbb{G}_{0,t}^{\nu,k} (\hat{\boldsymbol{\mu}}_t^{\nu}) = j', \boldsymbol{\beta} = \beta \right) f_{\beta | \mathbb{G}_{0,t}^{\nu,k} = j}(\beta) \, \mathrm{d}\beta$$

$$= o(1) + \int_{\mathbb{R}_+^k} f_{\beta | \mathbb{G}_{0,t}^{\nu,k} = j}(\beta) \sum_{(x_1, \ldots, x_k) \in \mathcal{X}_j^k} \prod_{i=1}^{k} \sum_{u \in \mathcal{U}} \pi_{i,t} (u \mid x_i') \cdot P \left( x_i \mid x_i', u, \mathbb{G}_{i,t}^{\nu} (\hat{\boldsymbol{\mu}}_t^{\nu}) \right) \, \mathrm{d}\beta .$$

where $\mathcal{X}_j^k := \{ (x_1, \ldots, x_k) \in \mathcal{X}^k : \sum_{i=1}^{k} \mathbf{1}_{\{x_i = s_1\}} = j \}$ and $\boldsymbol{\beta} := (\boldsymbol{\beta}_1, \ldots, \boldsymbol{\beta}_k)$ is a vector of $k$ random variables where each one is the latent parameter of one of the $k$ neighbors. Here, $f_{\beta | \mathbb{G} = j}$ is the conditional density function of $\boldsymbol{\beta}$ given that the neighborhood has exactly $j$ agents in state $s_1$. Similarly, $\beta = (\beta_1, \ldots, \beta_k) \in \mathbb{R}_+^k$ is one realization of the $k$ latent neighbor parameters $\boldsymbol{\beta}$. Now, we determine

$$f_{\beta | \mathbb{G}_{0,t}^{\nu,k} = j}(\beta) = \frac{f_{\beta, \mathbb{G}_{0,t}^{\nu,k}}(\beta, j)}{P(\mathbb{G}_{0,t}^{\nu,k} = j)} = \frac{f_{\beta}(\beta) P(\mathbb{G}_{0,t}^{\nu,k} = j | \boldsymbol{\beta} = \beta)}{P(\mathbb{G}_{0,t}^{\nu,k} = j)}$$

where we tacitly assume that $P(\mathbb{G}_{0,t}^{\nu,k} = j) > 0$. The case $P(\mathbb{G}_{0,t}^{\nu,k} = j) = 0$ can be neglected since $P(\mathbb{G}_{0,t}^{\nu,k} = j) \geq P_{\boldsymbol{\pi}, \boldsymbol{\mu}}^{\nu} \left( \mathbb{G}_{0,t}^{\nu,k} (\hat{\boldsymbol{\mu}}_t^{\nu}) = j', x_{0,t} = x' \right)$ implies that $P_{\boldsymbol{\pi}, \boldsymbol{\mu}}^{\nu} \left( \mathbb{G}_{0,t}^{\nu,k} (\hat{\boldsymbol{\mu}}_t^{\nu}) = j', x_{0,t} = x' \right) = 0$ which means that the respective summands for the calculation of $P_{\boldsymbol{\pi}, \boldsymbol{\mu}}^{\nu} \left( \mathbb{G}_{0,t+1}^{\nu,k} (\hat{\boldsymbol{\mu}}_t^{\nu}) = j, x_{0,t+1} = x \right)$ are zero and hence can be neglected. We know that

$$f_{\beta}(\beta) = \prod_{i=1}^{k} \frac{W(\alpha, \beta_i)}{\xi_W(\alpha)}$$

and

$$P(\mathbb{G}_{0,t}^{\nu,k} = j | \boldsymbol{\beta} = \beta) = \mathbb{E} \left[ \sum_{(x_1, \ldots x_k) \in \mathcal{X}_j^k} \prod_{i=1}^{k} \hat{\mu}_{\beta_i, t}^{\nu}(x_i) \right] .$$

We also have

$$P(\mathbb{G}_{0,t}^{\nu,k} = j) = \int_{\mathbb{R}_+^k} \left( \prod_{i=1}^{k} \frac{W(\alpha, \beta_i)}{\xi_W(\alpha)} \right) \mathbb{E} \left[ \sum_{(x_1, \ldots x_k) \in \mathcal{X}_j^k} \prod_{i=1}^{k} \hat{\mu}_{\beta_i, t}^{\nu}(x_i) \right] \, \mathrm{d}\beta$$

$$= \int_{\mathbb{R}_+^k} \sum_{(x_1, \ldots x_k) \in \mathcal{X}_j^k} \mathbb{E} \left[ \prod_{i=1}^{k} \frac{W(\alpha, \beta_i)}{\xi_W(\alpha)} \hat{\mu}_{\beta_i, t}^{\nu}(x_i) \right] \, \mathrm{d}\beta$$

which eventually yields

$$f_{\beta|\mathbb{G}_{0,t}^{\nu,k}=j}(\beta) = \frac{\sum_{(x_1,\ldots x_k)\in\mathcal{X}_j^k} \mathbb{E}\left[\prod_{i=1}^k \frac{W(\alpha,\beta_i)}{\xi_W(\alpha)}\hat{\mu}_{\beta_i,t}^\nu(x_i)\right]}{\int_{\mathbb{R}_+^k}\sum_{(z_1,\ldots z_k)\in\mathcal{X}_j^k}\mathbb{E}\left[\prod_{i=1}^k \frac{W(\alpha,\beta_i')}{\xi_W(\alpha)}\hat{\mu}_{\beta_i',t}^\nu(z_i)\right]\,\mathrm{d}\beta'} \; .$$

Neglecting values larger than $\alpha^*$, we obtain the following formulation for the limiting system

$$P_{\boldsymbol{\pi}}\left(\mathbb{G}_{\alpha,t+1}^k(\boldsymbol{\mu}_t)=j \mid \mathbb{G}_{\alpha,t}^k(\boldsymbol{\mu}_t)=j', x_{\alpha,t}=x'\right)$$

$$= \int_{[0,\alpha^*]^k} \frac{\sum_{(y_1,\ldots y_k)\in\mathcal{X}_j^k}\prod_{i=1}^k \frac{W(\alpha,\beta_i)}{\xi_W(\alpha)}\mu_{\beta_i,t}^\infty(y_i)}{\int_{[0,\alpha^*]^k}\sum_{(z_1,\ldots z_k)\in\mathcal{X}_j^k}\prod_{i=1}^k \frac{W(\alpha,\beta_i')}{\xi_W(\alpha)}\mu_{\beta_i',t}^\infty(z_i)\,\mathrm{d}\beta'}$$

$$\cdot \sum_{(x_1,\ldots x_k)\in\mathcal{X}_j^k}\prod_{i=1}^k\sum_{u\in\mathcal{U}}\pi_{\beta_i,t}^\infty(u\mid x_i')\cdot P\left(x_i\mid x_i',u,\mathbb{G}_{\beta_i,t}^\infty(\boldsymbol{\mu}_t)\right)\,\mathrm{d}\beta \; .$$

The following inequality connects the two systems

$$\left|P_{\boldsymbol{\pi}}\left(\mathbb{G}_{\alpha,t+1}^k(\boldsymbol{\mu}_t)=j \mid \mathbb{G}_{\alpha,t}^k(\boldsymbol{\mu}_t)=j', x_{\alpha,t}=x'\right)\right.$$
$$\left.-P_{\boldsymbol{\pi}}^\nu\left(\mathbb{G}_{0,t+1}^{\nu,k}(\hat{\boldsymbol{\mu}}_t^\nu)=j \mid \mathbb{G}_{0,t}^{\nu,k}(\hat{\boldsymbol{\mu}}_t^\nu)=j', x_{0,t}=x'\right)\right| < o(1) + \varepsilon_{\alpha^*} \; . \quad (2)$$

To see this, we reformulate

$$\left|P_{\boldsymbol{\pi}}\left(\mathbb{G}_{\alpha,t+1}^k(\boldsymbol{\mu}_t)=j \mid \mathbb{G}_{\alpha,t}^k(\boldsymbol{\mu}_t)=j', x_{\alpha,t}=x'\right)\right.$$
$$\left.-P_{\boldsymbol{\pi}}^\nu\left(\mathbb{G}_{0,t+1}^{\nu,k}(\hat{\boldsymbol{\mu}}_t^\nu)=j \mid \mathbb{G}_{0,t}^{\nu,k}(\hat{\boldsymbol{\mu}}_t^\nu)=j', x_{0,t}=x'\right)\right|$$

$$= \left| \int_{[0,\alpha^*]^k} \frac{\sum_{(y_1,\ldots y_k)\in\mathcal{X}_j^k}\prod_{i=1}^k \frac{W(\alpha,\beta_i)}{\xi_W(\alpha)}\mu_{\beta_i,t}^\infty(y_i)}{\int_{[0,\alpha^*]^k}\sum_{(z_1,\ldots z_k)\in\mathcal{X}_j^k}\prod_{i=1}^k \frac{W(\alpha,\beta_i')}{\xi_W(\alpha)}\mu_{\beta_i',t}^\infty(z_i)\,\mathrm{d}\beta'} \right.$$

$$\cdot \sum_{(x_1,\ldots x_k)\in\mathcal{X}_j^k}\prod_{i=1}^k\sum_{u\in\mathcal{U}}\pi_{\beta_i,t}^\infty(u\mid x_i')\cdot P\left(x_i\mid x_i',u,\mathbb{G}_{\beta_i,t}^\infty(\boldsymbol{\mu}_t)\right)\,\mathrm{d}\beta$$

$$-\int_{\mathbb{R}_+^k}\frac{\sum_{(y_1,\ldots y_k)\in\mathcal{X}_j^k}\mathbb{E}\left[\prod_{i=1}^k \frac{W(\alpha,\beta_i)}{\xi_W(\alpha)}\hat{\mu}_{\beta_i,t}^\nu(y_i)\right]}{\int_{\mathbb{R}_+^k}\sum_{(z_1,\ldots z_k)\in\mathcal{X}_j^k}\mathbb{E}\left[\prod_{i=1}^k \frac{W(\alpha,\beta_i')}{\xi_W(\alpha)}\hat{\mu}_{\beta_i',t}^\nu(z_i)\right]\,\mathrm{d}\beta'}$$

$$\left. \cdot \sum_{(x_1,\ldots x_k)\in\mathcal{X}_j^k}\prod_{i=1}^k\sum_{u\in\mathcal{U}}\pi_{i,t}(u\mid x_i')\cdot P\left(x_i\mid x_i',u,\mathbb{G}_{i,t}^\nu(\hat{\boldsymbol{\mu}}_t^\nu)\right)\,\mathrm{d}\beta \right|$$

$$\leq T_{(\mathrm{I})} + T_{(\mathrm{II})} + T_{(\mathrm{III})} + T_{(\mathrm{IV})} + T_{(\mathrm{V})}$$

where $T_{(\mathrm{I})}, T_{(\mathrm{II})}, T_{(\mathrm{III})}, T_{(\mathrm{IV})}, T_{(\mathrm{V})}$ denote five terms which are defined and bounded next. Start with

$$T_{(\mathrm{I})} := \left| \int_{[0,\alpha^*]^k} \frac{\sum_{(y_1,\ldots y_k)\in\mathcal{X}_j^k}\prod_{i=1}^k \frac{W(\alpha,\beta_i)}{\xi_W(\alpha)}\mu_{\beta_i,t}^\infty(y_i)}{\int_{[0,\alpha^*]^k}\sum_{(z_1,\ldots z_k)\in\mathcal{X}_j^k}\prod_{i=1}^k \frac{W(\alpha,\beta_i')}{\xi_W(\alpha)}\mu_{\beta_i',t}^\infty(z_i)\,\mathrm{d}\beta'} \right.$$

$$\cdot \sum_{(x_1,\ldots x_k)\in\mathcal{X}_j^k}\prod_{i=1}^k\sum_{u\in\mathcal{U}}\pi_{\beta_i,t}^\infty(u\mid x_i')\cdot P\left(x_i\mid x_i',u,\mathbb{G}_{\beta_i,t}^\infty(\boldsymbol{\mu}_t)\right)\,\mathrm{d}\beta$$

$$-\int_{\mathbb{R}_+^k}\frac{\sum_{(y_1,\ldots y_k)\in\mathcal{X}_j^k}\mathbb{E}\left[\prod_{i=1}^k \frac{W(\alpha,\beta_i)}{\xi_W(\alpha)}\hat{\mu}_{\beta_i,t}^\nu(y_i)\right]}{\int_{[0,\alpha^*]^k}\sum_{(z_1,\ldots z_k)\in\mathcal{X}_j^k}\prod_{i=1}^k \frac{W(\alpha,\beta_i')}{\xi_W(\alpha)}\mu_{\beta_i',t}^\infty(z_i)\,\mathrm{d}\beta'}$$

$$\left. \cdot \sum_{(x_1,\ldots x_k)\in\mathcal{X}_j^k}\prod_{i=1}^k\sum_{u\in\mathcal{U}}\pi_{\beta_i,t}^\infty(u\mid x_i')\cdot P\left(x_i\mid x_i',u,\mathbb{G}_{\beta_i,t}^\infty(\boldsymbol{\mu}_t)\right)\,\mathrm{d}\beta \right|$$

$$= \varepsilon_{\alpha^*} + O(1) \left| \int_{[0,\alpha^*]^k} \sum_{(y_1,\ldots y_k) \in \mathcal{X}_j^k} \left( \left( \prod_{i=1}^k \mu_{\beta_i,t}^\infty(y_i) \right) - \mathbb{E}\left[ \prod_{i=1}^k \hat{\mu}_{\beta_i,t}^\nu(y_i) \right] \right) \right.$$

$$\left. \cdot \sum_{(x_1,\ldots x_k) \in \mathcal{X}_j^k} \prod_{i=1}^k \frac{W(\alpha,\beta_i)}{\xi_W(\alpha)} \sum_{u \in \mathcal{U}} \pi_{\beta_i,t}^\infty \left( u \mid x_i' \right) \cdot P\left( x_i \mid x_i', u, \mathbb{G}_{\beta_i,t}^\infty(\boldsymbol{\mu}_t) \right) \mathrm{d}\beta \right|$$

$$\leq \varepsilon_{\alpha^*} + O(1) \sum_{(y_1,\ldots y_k) \in \mathcal{X}_j^k} \sum_{(x_1,\ldots x_k) \in \mathcal{X}_j^k} \mathbb{E}\left[ \left| \int_{[0,\alpha^*]^k} \left( \left( \prod_{i=1}^k \mu_{\beta_i,t}^\infty(y_i) \right) - \prod_{i=1}^k \hat{\mu}_{\beta_i,t}^\nu(y_i) \right) \right. \right.$$

$$\left. \left. \cdot \prod_{i=1}^k \frac{W(\alpha,\beta_i)}{\xi_W(\alpha)} \sum_{u \in \mathcal{U}} \pi_{\beta_i,t}^\infty \left( u \mid x_i' \right) \cdot P\left( x_i \mid x_i', u, \mathbb{G}_{\beta_i,t}^\infty(\boldsymbol{\mu}_t) \right) \mathrm{d}\beta \right| \right]$$

$$\leq \varepsilon_{\alpha^*} + k(\alpha^*)^{k-1} O(1) \sum_{(y_1,\ldots y_k) \in \mathcal{X}_j^k} \sum_{(x_1,\ldots x_k) \in \mathcal{X}_j^k} \mathbb{E}\left[ \left| \int_{[0,\alpha^*]} \left( \mu_{\beta_1,t}^\infty(y_1) - \hat{\mu}_{\beta_1,t}^\nu(y_1) \right) \right. \right.$$

$$\left. \left. \cdot \frac{W(\alpha,\beta_1)}{\xi_W(\alpha)} \sum_{u \in \mathcal{U}} \pi_{\beta_1,t}^\infty \left( u \mid x_1' \right) \cdot P\left( x_1 \mid x_i', u, \mathbb{G}_{\beta_1,t}^\infty(\boldsymbol{\mu}_t) \right) \mathrm{d}\beta \right| \right]$$

$$= \varepsilon_{\alpha^*} + o(1)$$

by applying Theorem 1 in the last line. Moving on to the second term

$$T_{(\mathrm{II})} := \left| \int_{\mathbb{R}_+^k} \frac{\sum_{(y_1,\ldots y_k) \in \mathcal{X}_j^k} \mathbb{E}\left[ \prod_{i=1}^k \frac{W(\alpha,\beta_i)}{\xi_W(\alpha)} \hat{\mu}_{\beta_i,t}^\nu(y_i) \right]}{\int_{[0,\alpha^*]^k} \sum_{(z_1,\ldots z_k) \in \mathcal{X}_j^k} \prod_{i=1}^k \frac{W(\alpha,\beta_i')}{\xi_W(\alpha)} \mu_{\beta_i',t}^\infty(z_i) \mathrm{d}\beta'} \right.$$

$$\cdot \sum_{(x_1,\ldots x_k) \in \mathcal{X}_j^k} \prod_{i=1}^k \sum_{u \in \mathcal{U}} \pi_{\beta_i,t}^\infty \left( u \mid x_i' \right) \cdot P\left( x_i \mid x_i', u, \mathbb{G}_{\beta_i,t}^\infty(\boldsymbol{\mu}_t) \right) \mathrm{d}\beta$$

$$- \int_{\mathbb{R}_+^k} \frac{\sum_{(y_1,\ldots y_k) \in \mathcal{X}_j^k} \mathbb{E}\left[ \prod_{i=1}^k \frac{W(\alpha,\beta_i)}{\xi_W(\alpha)} \hat{\mu}_{\beta_i,t}^\nu(y_i) \right]}{\int_{\mathbb{R}_+^k} \sum_{(z_1,\ldots z_k) \in \mathcal{X}_j^k} \mathbb{E}\left[ \prod_{i=1}^k \frac{W(\alpha,\beta_i')}{\xi_W(\alpha)} \hat{\mu}_{\beta_i',t}^\nu(z_i) \right] \mathrm{d}\beta'}$$

$$\left. \cdot \sum_{(x_1,\ldots x_k) \in \mathcal{X}_j^k} \prod_{i=1}^k \sum_{u \in \mathcal{U}} \pi_{\beta_i,t}^\infty \left( u \mid x_i' \right) \cdot P\left( x_i \mid x_i', u, \mathbb{G}_{\beta_i,t}^\infty(\boldsymbol{\mu}_t) \right) \mathrm{d}\beta \right|$$

we note that the difference of the two denominators is bounded by

$$\left| \int_{[0,\alpha^*]^k} \sum_{(z_1,\ldots z_k) \in \mathcal{X}_j^k} \prod_{i=1}^k \frac{W(\alpha,\beta_i')}{\xi_W(\alpha)} \mu_{\beta_i',t}^\infty(z_i) \mathrm{d}\beta' \right.$$

$$\left. - \int_{\mathbb{R}_+^k} \sum_{(z_1,\ldots z_k) \in \mathcal{X}_j^k} \mathbb{E}\left[ \prod_{i=1}^k \frac{W(\alpha,\beta_i')}{\xi_W(\alpha)} \hat{\mu}_{\beta_i',t}^\nu(z_i) \right] \mathrm{d}\beta' \right| \leq \varepsilon_{\alpha^*} + o(1)$$

by an argument as for the term $T_{(\mathrm{I})}$. Hence, the second term is also bounded by

$$T_{(\mathrm{II})} \leq \varepsilon_{\alpha^*} + o(1).$$

Finally, we are left with defining and bounding the third, fourth and fifth terms $T_{(\mathrm{III})}, T_{(\mathrm{IV})}, T_{(\mathrm{V})}$. They are given by

$$T_{(\mathrm{III})} := \left| \int_{\mathbb{R}_+^k} \frac{\sum_{(y_1,\ldots y_k) \in \mathcal{X}_j^k} \mathbb{E}\left[ \prod_{i=1}^k \frac{W(\alpha,\beta_i)}{\xi_W(\alpha)} \hat{\mu}_{\beta_i,t}^\nu(y_i) \right]}{\int_{\mathbb{R}_+^k} \sum_{(z_1,\ldots z_k) \in \mathcal{X}_j^k} \mathbb{E}\left[ \prod_{i=1}^k \frac{W(\alpha,\beta_i')}{\xi_W(\alpha)} \hat{\mu}_{\beta_i',t}^\nu(z_i) \right] \mathrm{d}\beta'} \right.$$

$$\cdot \sum_{(x_1,\ldots x_k)\in \mathcal{X}_j^k} \prod_{i=1}^{k} \sum_{u\in\mathcal{U}} \pi_{\beta_i,t}^{\infty}\left(u \mid x_i'\right) \cdot P\left(x_i \mid x_i', u, \mathbb{G}_{\beta_i,t}^{\infty}\left(\boldsymbol{\mu}_t\right)\right)\, \mathrm{d}\beta$$

$$-\int_{\mathbb{R}_+^k} \frac{\sum_{(y_1,\ldots y_k)\in \mathcal{X}_j^k} \mathbb{E}\left[\prod_{i=1}^{k} \frac{W(\alpha,\beta_i)}{\xi_W(\alpha)} \hat{\mu}_{\beta_i,t}^{\nu}(y_i)\right]}{\int_{\mathbb{R}_+^k} \sum_{(z_1,\ldots z_k)\in \mathcal{X}_j^k} \mathbb{E}\left[\prod_{i=1}^{k} \frac{W(\alpha,\beta_i')}{\xi_W(\alpha)} \hat{\mu}_{\beta_i',t}^{\nu}(z_i)\right]\, \mathrm{d}\beta'}$$

$$\left. \cdot \sum_{(x_1,\ldots x_k)\in \mathcal{X}_j^k} \prod_{i=1}^{k} \sum_{u\in\mathcal{U}} \pi_{\beta_i,t}^{\infty}\left(u \mid x_i'\right) \cdot P\left(x_i \mid x_i', u, \mathbb{G}_{\beta_i,t}^{\infty}\left(\hat{\boldsymbol{\mu}}_t^{\nu}\right)\right)\, \mathrm{d}\beta \right|$$

and

$$T_{(\mathrm{IV})} := \left| \int_{\mathbb{R}_+^k} \frac{\sum_{(y_1,\ldots y_k)\in \mathcal{X}_j^k} \mathbb{E}\left[\prod_{i=1}^{k} \frac{W(\alpha,\beta_i)}{\xi_W(\alpha)} \hat{\mu}_{\beta_i,t}^{\nu}(y_i)\right]}{\int_{\mathbb{R}_+^k} \sum_{(z_1,\ldots z_k)\in \mathcal{X}_j^k} \mathbb{E}\left[\prod_{i=1}^{k} \frac{W(\alpha,\beta_i')}{\xi_W(\alpha)} \hat{\mu}_{\beta_i',t}^{\nu}(z_i)\right]\, \mathrm{d}\beta'} \right.$$

$$\cdot \sum_{(x_1,\ldots x_k)\in \mathcal{X}_j^k} \prod_{i=1}^{k} \sum_{u\in\mathcal{U}} \pi_{\beta_i,t}^{\infty}\left(u \mid x_i'\right) \cdot P\left(x_i \mid x_i', u, \mathbb{G}_{\beta_i,t}^{\infty}\left(\hat{\boldsymbol{\mu}}_t^{\nu}\right)\right)\, \mathrm{d}\beta$$

$$-\int_{\mathbb{R}_+^k} \frac{\sum_{(y_1,\ldots y_k)\in \mathcal{X}_j^k} \mathbb{E}\left[\prod_{i=1}^{k} \frac{W(\alpha,\beta_i)}{\xi_W(\alpha)} \hat{\mu}_{\beta_i,t}^{\nu}(y_i)\right]}{\int_{\mathbb{R}_+^k} \sum_{(z_1,\ldots z_k)\in \mathcal{X}_j^k} \mathbb{E}\left[\prod_{i=1}^{k} \frac{W(\alpha,\beta_i')}{\xi_W(\alpha)} \hat{\mu}_{\beta_i',t}^{\nu}(z_i)\right]\, \mathrm{d}\beta'}$$

$$\left. \cdot \sum_{(x_1,\ldots x_k)\in \mathcal{X}_j^k} \prod_{i=1}^{k} \sum_{u\in\mathcal{U}} \pi_{\beta_i,t}^{\infty}\left(u \mid x_i'\right) \cdot P\left(x_i \mid x_i', u, \mathbb{G}_{i,t}^{\nu}\left(\hat{\boldsymbol{\mu}}_t^{\nu}\right)\right)\, \mathrm{d}\beta \right|$$

and

$$T_{(\mathrm{V})} := \left| \int_{\mathbb{R}_+^k} \frac{\sum_{(y_1,\ldots y_k)\in \mathcal{X}_j^k} \mathbb{E}\left[\prod_{i=1}^{k} \frac{W(\alpha,\beta_i)}{\xi_W(\alpha)} \hat{\mu}_{\beta_i,t}^{\nu}(y_i)\right]}{\int_{\mathbb{R}_+^k} \sum_{(z_1,\ldots z_k)\in \mathcal{X}_j^k} \mathbb{E}\left[\prod_{i=1}^{k} \frac{W(\alpha,\beta_i')}{\xi_W(\alpha)} \hat{\mu}_{\beta_i',t}^{\nu}(z_i)\right]\, \mathrm{d}\beta'} \right.$$

$$\cdot \sum_{(x_1,\ldots x_k)\in \mathcal{X}_j^k} \prod_{i=1}^{k} \sum_{u\in\mathcal{U}} \pi_{\beta_i,t}^{\infty}\left(u \mid x_i'\right) \cdot P\left(x_i \mid x_i', u, \mathbb{G}_{i,t}^{\nu}\left(\hat{\boldsymbol{\mu}}_t^{\nu}\right)\right)\, \mathrm{d}\beta$$

$$-\int_{\mathbb{R}_+^k} \frac{\sum_{(y_1,\ldots y_k)\in \mathcal{X}_j^k} \mathbb{E}\left[\prod_{i=1}^{k} \frac{W(\alpha,\beta_i)}{\xi_W(\alpha)} \hat{\mu}_{\beta_i,t}^{\nu}(y_i)\right]}{\int_{\mathbb{R}_+^k} \sum_{(z_1,\ldots z_k)\in \mathcal{X}_j^k} \mathbb{E}\left[\prod_{i=1}^{k} \frac{W(\alpha,\beta_i')}{\xi_W(\alpha)} \hat{\mu}_{\beta_i',t}^{\nu}(z_i)\right]\, \mathrm{d}\beta'}$$

$$\left. \cdot \sum_{(x_1,\ldots x_k)\in \mathcal{X}_j^k} \prod_{i=1}^{k} \sum_{u\in\mathcal{U}} \pi_{\beta_i,t}\left(u \mid x_i'\right) \cdot P\left(x_i \mid x_i', u, \mathbb{G}_{i,t}^{\nu}\left(\hat{\boldsymbol{\mu}}_t^{\nu}\right)\right)\, \mathrm{d}\beta \right|$$

where each of the three terms is bounded by arguments as in the proof of Theorem 2. This establishes inequality (2). The just proved inequality (2) also implies by induction that

$$\left| P_{\boldsymbol{\pi},\boldsymbol{\mu}}\left(\mathbb{G}_{\alpha,t}^{k}\left(\boldsymbol{\mu}_t\right)=j, x_{\alpha,t}=x\right) - P_{\boldsymbol{\pi},\boldsymbol{\mu}}^{\nu}\left(\mathbb{G}_{0,t}^{\nu,k}\left(\hat{\boldsymbol{\mu}}_t^{\nu}\right)=j, x_{0,t}=x\right) \right| < o(1) + \varepsilon_{\alpha^*} \qquad (3)$$

where $\varepsilon_{\alpha^*}$ is an error term depending on $\alpha^*$ with $\lim_{\alpha^*\to\infty} \varepsilon_{\alpha^*} = 0$. Exploiting the symmetry of the limiting model, we can further reformulate the probability in the limiting system as

$$\sum_{(x_1,\ldots x_k)\in \mathcal{X}_j^k} \prod_{i=1}^{k} \sum_{u\in\mathcal{U}} \pi_{\beta_i,t}^{\infty}\left(u \mid x_i'\right) \cdot P\left(x_i \mid x_i', u, \mathbb{G}_{\beta_i,t}^{\infty}\left(\boldsymbol{\mu}_t\right)\right)$$

$$= \sum_{(x_1,\ldots x_k)\in \mathcal{X}_j^k} \left( \prod_{i=1}^{j'} \sum_{u\in\mathcal{U}} \pi_{\beta_i,t}^{\infty}\left(u \mid s_1\right) \cdot P\left(x_i \mid s_1, u, \mathbb{G}_{\beta_i,t}^{\infty}\left(\hat{\boldsymbol{\mu}}_t^{\nu}\right)\right)\, \mathrm{d}\beta_i \right)$$

$$
\cdot \left( \prod_{i=j'+1}^{k} \sum_{u \in \mathcal{U}} \pi_{\beta_i,t}^{\infty} (u \mid s_2) \cdot P\left(x_i \mid s_2, u, \mathbb{G}_{\beta_i,t}^{\infty}(\hat{\boldsymbol{\mu}}_t^{\nu})\right) \, \mathrm{d}\beta_i \right)
$$

$$
= \sum_{\ell=\max\{0,j+j'-k\}}^{\min\{j,j'\}} \binom{j'}{\ell} \left( \prod_{i=1}^{\ell} \sum_{u \in \mathcal{U}} \pi_{\beta_i,t}^{\infty} (u \mid s_1) \cdot P\left(s_1 \mid s_1, u, \mathbb{G}_{\beta_i,t}^{\infty}(\boldsymbol{\mu}_t)\right) \right)
$$

$$
\cdot \left( \prod_{i=\ell+1}^{j'} \sum_{u \in \mathcal{U}} \pi_{\beta_i,t}^{\infty} (u \mid s_1) \cdot P\left(s_2 \mid s_1, u, \mathbb{G}_{\beta_i,t}^{\infty}(\boldsymbol{\mu}_t)\right) \right)
$$

$$
\cdot \binom{k-j'}{j-\ell} \left( \prod_{i=j'+1}^{j+j'-\ell} \sum_{u \in \mathcal{U}} \pi_{\beta_i,t}^{\infty} (u \mid s_2) \cdot P\left(s_1 \mid s_2, u, \mathbb{G}_{\beta_i,t}^{\infty}(\boldsymbol{\mu}_t)\right) \right)
$$

$$
\cdot \prod_{i=j+j'-\ell+1}^{k} \sum_{u \in \mathcal{U}} \pi_{\beta_i,t}^{\infty} (u \mid s_2) \cdot P\left(s_2 \mid s_2, u, \mathbb{G}_{\beta_i,t}^{\infty}(\boldsymbol{\mu}_t)\right).
$$

Combining these findings we arrive at

$$
P_{\boldsymbol{\pi},\boldsymbol{\mu}} \left( \mathbb{G}_{\alpha,t+1}^k (\boldsymbol{\mu}_t) = j, x_{\alpha,t+1} = x \right)
$$

$$
= \sum_{x' \in \mathcal{X}} \sum_{j'=0}^{k} P_{\boldsymbol{\pi},\boldsymbol{\mu}} \left( \mathbb{G}_{\alpha,t}^k (\boldsymbol{\mu}_t) = j', x_{\alpha,t} = x' \right) \sum_{u \in \mathcal{U}} \pi_{\alpha,t}^k (u \mid x') \cdot P\left( x \mid x', u, \frac{j'}{k} \right)
$$

$$
\cdot \int_{[0,\alpha^*]^k} \frac{\sum_{(y_1,\ldots y_k) \in \mathcal{X}_j^k} \prod_{i=1}^{k} \frac{W(\alpha,\beta_i)}{\xi_W(\alpha)} \mu_{\beta_i,t}^{\infty}(y_i)}{\int_{[0,\alpha^*]^k} \sum_{(z_1,\ldots z_k) \in \mathcal{X}_j^k} \prod_{i=1}^{k} \frac{W(\alpha,\beta_i')}{\xi_W(\alpha)} \mu_{\beta_i',t}^{\infty}(z_i) \, \mathrm{d}\beta'}
$$

$$
\cdot \sum_{\ell=\max\{0,j+j'-k\}}^{\min\{j,j'\}} \binom{j'}{\ell} \left( \prod_{i=1}^{\ell} \sum_{u \in \mathcal{U}} \pi_{\beta_i,t}^{\infty} (u \mid s_1) \cdot P\left(s_1 \mid s_1, u, \mathbb{G}_{\beta_i,t}^{\infty}(\boldsymbol{\mu}_t)\right) \right)
$$

$$
\cdot \left( \prod_{i=\ell+1}^{j'} \sum_{u \in \mathcal{U}} \pi_{\beta_i,t}^{\infty} (u \mid s_1) \cdot P\left(s_2 \mid s_1, u, \mathbb{G}_{\beta_i,t}^{\infty}(\boldsymbol{\mu}_t)\right) \right)
$$

$$
\cdot \binom{k-j'}{j-\ell} \left( \prod_{i=j'+1}^{j+j'-\ell} \sum_{u \in \mathcal{U}} \pi_{\beta_i,t}^{\infty} (u \mid s_2) \cdot P\left(s_1 \mid s_2, u, \mathbb{G}_{\beta_i,t}^{\infty}(\boldsymbol{\mu}_t)\right) \right)
$$

$$
\cdot \prod_{i=j+j'-\ell+1}^{k} \sum_{u \in \mathcal{U}} \pi_{\beta_i,t}^{\infty} (u \mid s_2) \cdot P\left(s_2 \mid s_2, u, \mathbb{G}_{\beta_i,t}^{\infty}(\boldsymbol{\mu}_t)\right) \, \mathrm{d}\beta_1 \cdots \mathrm{d}\beta_k
$$

for the limiting system.

**Extension to general finite state spaces.** Consider some uniformly, at random picked node $v_0$ with degree $k$, state $x_0$ and latent parameter $\alpha$. Furthermore, denote by $x_1, \ldots, x_k$ the states of its $k$ neighbors. Let $G = \left( \frac{g_1}{k}, \ldots, \frac{g_d}{k} \right) \in \boldsymbol{\mathcal{G}}^k$ be the neighborhood and $\mathcal{X} = \{s_1, \ldots, s_d\}$ the finite state space, where $d$ is the number of different states, and $g_1$ is the number of neighbors in the first state $s_1$, $g_2$ in the second state $s_2$, and so on.

$$
P_{\boldsymbol{\pi},\boldsymbol{\mu}}^{\nu} \left( \mathbb{G}_{0,t+1}^{\nu,k} (\hat{\boldsymbol{\mu}}_t^{\nu}) = G, x_{0,t+1} = x \right)
$$

$$
= \sum_{x' \in \mathcal{X}} \sum_{G' \in \boldsymbol{\mathcal{G}}^k} P_{\boldsymbol{\pi},\boldsymbol{\mu}}^{\nu} \left( \mathbb{G}_{0,t+1}^{\nu,k} (\hat{\boldsymbol{\mu}}_t^{\nu}) = G, \mathbb{G}_{0,t}^{\nu,k} = G', x_{0,t} = x', x_{0,t+1} = x \right)
$$

$$
= \sum_{x' \in \mathcal{X}} \sum_{G' \in \boldsymbol{\mathcal{G}}^k} P_{\boldsymbol{\pi},\boldsymbol{\mu}}^{\nu} \left( \mathbb{G}_{0,t}^{\nu,k} (\hat{\boldsymbol{\mu}}_t^{\nu}) = G', x_{0,t} = x' \right)
$$

$$
\cdot P_{\boldsymbol{\pi},\boldsymbol{\mu}}^{\nu} \left( x_{0,t+1} = x \mid \mathbb{G}_{0,t}^{\nu,k} (\hat{\boldsymbol{\mu}}_t^{\nu}) = G', x_{0,t} = x' \right)
$$

$$\cdot P_{\boldsymbol{\pi}}^{\nu}\left(\mathbb{G}_{0,t+1}^{\nu,k}\left(\hat{\boldsymbol{\mu}}_t^{\nu}\right) = G \mid x_{0,t+1} = x, \mathbb{G}_{0,t}^{\nu,k} = G', x_{0,t} = x'\right)$$

$$= \sum_{x' \in \mathcal{X}} \sum_{G' \in \mathcal{G}^k} P_{\boldsymbol{\pi},\boldsymbol{\mu}}^{\nu}\left(\mathbb{G}_{0,t}^{\nu,k}\left(\hat{\boldsymbol{\mu}}_t^{\nu}\right) = G', x_{0,t} = x'\right) \sum_{u \in \mathcal{U}} \pi_{0,t}^k\left(u \mid x'\right) \cdot P\left(x \mid x', u, G'\right)$$

$$\cdot P_{\boldsymbol{\pi}}^{\nu}\left(\mathbb{G}_{0,t+1}^{\nu,k}\left(\hat{\boldsymbol{\mu}}_t^{\nu}\right) = G \mid \mathbb{G}_{0,t}^{\nu,k}\left(\hat{\boldsymbol{\mu}}_t^{\nu}\right) = G', x_{0,t} = x'\right).$$

We focus on

$$P_{\boldsymbol{\pi}}^{\nu}\left(\mathbb{G}_{0,t+1}^{\nu,k}\left(\hat{\boldsymbol{\mu}}_t^{\nu}\right) = G \mid \mathbb{G}_{0,t}^{\nu,k}\left(\hat{\boldsymbol{\mu}}_t^{\nu}\right) = G', x_{0,t} = x'\right)$$

$$= P_{\boldsymbol{\pi}}^{\nu}\left(\mathbb{G}_{0,t+1}^{\nu,k}\left(\hat{\boldsymbol{\mu}}_t^{\nu}\right) = G \mid \mathbb{G}_{0,t}^{\nu,k}\left(\hat{\boldsymbol{\mu}}_t^{\nu}\right) = G'\right) + o(1)$$

$$= o(1) + \int_{\mathbb{R}_+^k} P_{\boldsymbol{\pi}}^{\nu}\left(\mathbb{G}_{0,t+1}^{\nu,k}\left(\hat{\boldsymbol{\mu}}_t^{\nu}\right) = G \mid \mathbb{G}_{0,t}^{\nu,k}\left(\hat{\boldsymbol{\mu}}_t^{\nu}\right) = G', \boldsymbol{\beta} = \beta\right) f_{\beta \mid \mathbb{G}_{0,t}^{\nu,k}=G}(\beta)\, \mathrm{d}\beta$$

$$= o(1) + \int_{\mathbb{R}_+^k} f_{\beta \mid \mathbb{G}_{0,t}^{\nu,k}=G}(\beta) \sum_{(x_1,\ldots x_k) \in \mathcal{X}_G^k} \prod_{i=1}^{k} \sum_{u \in \mathcal{U}} \pi_{i,t}\left(u \mid x_i'\right) \cdot P\left(x_i \mid x_i', u, \mathbb{G}_{i,t}^{\nu}\left(\hat{\boldsymbol{\mu}}_t^{\nu}\right)\right)\, \mathrm{d}\beta.$$

where $\mathcal{X}_G^k := \{(x_1, \ldots x_k) \in \mathcal{X}^k : \forall j \in [d] : \sum_{i=1}^{k} \mathbf{1}_{\{x_i = s_j\}} = g_j\}$. Similar to the binary state case, we can calculate

$$f_{\beta \mid \mathbb{G}_{0,t}^{\nu,k}=G}(\beta) = \frac{\sum_{(x_1,\ldots x_k) \in \mathcal{X}_G^k} \mathbb{E}\left[\prod_{i=1}^{k} \frac{W(\alpha, \beta_i)}{\xi_W(\alpha)} \hat{\mu}_{\beta_i, t}^{\nu}(x_i)\right]}{\int_{\mathbb{R}_+^k} \sum_{(z_1,\ldots z_k) \in \mathcal{X}_G^k} \mathbb{E}\left[\prod_{i=1}^{k} \frac{W(\alpha, \beta_i')}{\xi_W(\alpha)} \hat{\mu}_{\beta_i', t}^{\nu}(z_i)\right]\, \mathrm{d}\beta'}.$$

This brings us to the limiting case

$$P_{\boldsymbol{\pi}}\left(\mathbb{G}_{\alpha, t+1}^k\left(\boldsymbol{\mu}_t\right) = G \mid \mathbb{G}_{\alpha, t}^k\left(\boldsymbol{\mu}_t\right) = G', x_{\alpha, t} = x'\right)$$

$$= \int_{[0, \alpha^*]^k} \frac{\sum_{(y_1,\ldots y_k) \in \mathcal{X}_G^k} \prod_{i=1}^{k} \frac{W(\alpha, \beta_i)}{\xi_W(\alpha)} \mu_{\beta_i, t}^{\infty}(y_i)}{\int_{[0, \alpha^*]^k} \sum_{(z_1,\ldots z_k) \in \mathcal{X}_G^k} \prod_{i=1}^{k} \frac{W(\alpha, \beta_i')}{\xi_W(\alpha)} \mu_{\beta_i', t}^{\infty}(z_i)\, \mathrm{d}\beta'}$$

$$\cdot \sum_{(x_1,\ldots x_k) \in \mathcal{X}_G^k} \prod_{i=1}^{k} \sum_{u \in \mathcal{U}} \pi_{\beta_i, t}^{\infty}\left(u \mid x_i'\right) \cdot P\left(x_i \mid x_i', u, \mathbb{G}_{\beta_i, t}^{\infty}\left(\boldsymbol{\mu}_t\right)\right)\, \mathrm{d}\beta.$$

Following the argumentation in the binary case, we can establish the two inequalities

$$\left| P_{\boldsymbol{\pi}}\left(\mathbb{G}_{\alpha, t+1}^k\left(\boldsymbol{\mu}_t\right) = G \mid \mathbb{G}_{\alpha, t}^k\left(\boldsymbol{\mu}_t\right) = G', x_{\alpha, t} = x'\right)\right.$$

$$\left. - P_{\boldsymbol{\pi}}^{\nu}\left(\mathbb{G}_{0,t+1}^{\nu,k}\left(\hat{\boldsymbol{\mu}}_t^{\nu}\right) = G \mid \mathbb{G}_{0,t}^{\nu,k}\left(\hat{\boldsymbol{\mu}}_t^{\nu}\right) = G', x_{0,t} = x'\right)\right| < o(1) + \varepsilon_{\alpha^*} \quad (4)$$

and

$$\left| P_{\boldsymbol{\pi},\boldsymbol{\mu}}\left(\mathbb{G}_{\alpha, t}^k\left(\boldsymbol{\mu}_t\right) = G, x_{\alpha, t} = x\right) - P_{\boldsymbol{\pi},\boldsymbol{\mu}}^{\nu}\left(\mathbb{G}_{0,t}^{\nu,k}\left(\hat{\boldsymbol{\mu}}_t^{\nu}\right) = G, x_{0,t} = x\right)\right| < o(1) + \varepsilon_{\alpha^*}. \quad (5)$$

For notational convenience, let us define a set of matrices $\mathcal{A}^k(G, G')$ for each pair of vectors $G, G' \in \mathcal{G}^k$ and finite $k \in \mathbb{N}$ as

$$\mathcal{A}^k(G, G') := \left\{ A = (a_{ij})_{i,j \in [d]} \in \mathbb{N}_0^{d \times d} : \sum_{\ell=1}^{d} a_{i\ell} = g_i \text{ and } \sum_{\ell=1}^{d} a_{\ell j} = g_j', \quad \forall i, j \in [d] \right\}$$

Keeping in mind the symmetry of the limiting model, we have

$$\sum_{(x_1,\ldots x_k) \in \mathcal{X}_G^k} \prod_{i=1}^{k} \sum_{u \in \mathcal{U}} \pi_{\beta_i, t}^{\infty}\left(u \mid x_i'\right) \cdot P\left(x_i \mid x_i', u, \mathbb{G}_{\beta_i, t}^{\infty}\left(\boldsymbol{\mu}_t\right)\right)$$

$$= \sum_{(x_1,\ldots x_k) \in \mathcal{X}_G^k} \prod_{j=1}^{d} \prod_{i=1}^{k} \mathbf{1}_{\{x_i' = s_j\}} \sum_{u \in \mathcal{U}} \pi_{\beta_i, t}^{\infty}\left(u \mid s_j\right) \cdot P\left(x_i \mid s_j, u, \mathbb{G}_{\beta_i, t}^{\infty}\left(\boldsymbol{\mu}_t\right)\right)$$

$$= \sum_{(a_{ij})_{i,j\in[d]}\in\mathcal{A}^k(G,G')} \prod_{j=1}^{d} \binom{g'_j}{a_{1j},\ldots,a_{dj}} \prod_{i=1}^{d} \left( \sum_{u\in\mathcal{U}} \pi^\infty_{\beta_i,t}(u\mid s_j) \cdot P\left(s_i\mid s_j,u,\mathbb{G}^\infty_{\beta_i,t}(\hat{\boldsymbol{\mu}}^\nu_t)\right) \right)^{a_{ij}}.$$

Eventually, the previous results yield

$$P_{\boldsymbol{\pi},\boldsymbol{\mu}}\left(\mathbb{G}^k_{\alpha,t+1}(\boldsymbol{\mu}_t) = G, x_{\alpha,t+1} = x\right)$$

$$= \sum_{x'\in\mathcal{X}} \sum_{\substack{(g'_1,\ldots,g'_d)\in\mathbb{N}_0:\\ g'_1+\ldots+g'_d=k}} P_{\boldsymbol{\pi},\boldsymbol{\mu}}\left(\mathbb{G}^k_{\alpha,t}(\boldsymbol{\mu}_t) = G', x_{\alpha,t} = x'\right) \sum_{u\in\mathcal{U}} \pi^k_{\alpha,t}(u\mid x') \cdot P(x\mid x',u,G')$$

$$\cdot \int_{[0,\alpha^*]^k} \frac{\sum_{(y_1,\ldots y_k)\in\mathcal{X}^k_G} \prod_{i=1}^{k} \frac{W(\alpha,\beta_i)}{\xi_W(\alpha)} \mu^\infty_{\beta_i,t}(y_i)}{\int_{[0,\alpha^*]^k} \sum_{(z_1,\ldots z_k)\in\mathcal{X}^k_G} \prod_{i=1}^{k} \frac{W(\alpha,\beta'_i)}{\xi_W(\alpha)} \mu^\infty_{\beta'_i,t}(z_i)\,d\beta'} \sum_{(a_{ij})_{i,j\in[d]}\in\mathcal{A}^k(G,G')}$$

$$\cdot \prod_{j=1}^{d} \binom{g'_j}{a_{1j},\ldots,a_{dj}} \prod_{i=1}^{d} \left( \sum_{u\in\mathcal{U}} \pi^\infty_{\beta_i,t}(u\mid s_j) \cdot P\left(s_i\mid s_j,u,\mathbb{G}^\infty_{\beta_i,t}(\hat{\boldsymbol{\mu}}^\nu_t)\right) \right)^{a_{ij}} d\beta_1\cdots d\beta_k.$$

## D  PROOF OF THEOREM 2

This proof focuses on binary state spaces $\mathcal{X} = \{s_1, s_2\}$. The extension of the proof given below to arbitrary finite state spaces is straightforward, although the notations are sometimes less convenient. We prove the claim via induction over $t$ and start with the case $t = 0$

$$\mathbb{E}\left[\left|\hat{\boldsymbol{\mu}}^{\nu,k}_0(f) - \boldsymbol{\mu}^k_0(f,\nu)\right|\right]$$

$$= \mathbb{E}\left[\left| \frac{\sqrt{2|E_\nu|}}{|V_{\nu,k}|} \cdot \sum_{i\in V_\nu} \mathbf{1}_{\{\deg(v_i)=k\}} \int_{\frac{i-1}{\sqrt{2|E_\nu|}}}^{\frac{i}{\sqrt{2|E_\nu|}}} \sum_{x\in\mathcal{X}} f(x,\alpha)\hat{\mu}^\nu_{\alpha,0}(x)\,d\alpha \right.\right.$$

$$\left.\left. - \frac{\sqrt{\bar{\xi}_W}}{\int_0^\infty \text{Poi}^W_{\nu,\alpha}(k)d\alpha} \cdot \int_0^\infty \text{Poi}^W_{\nu,\alpha}(k) \sum_{x\in\mathcal{X}} f(x,\alpha)\mu^k_{\alpha,0}(x)\,d\alpha \right|\right]$$

$$\leq \mathbb{E}\left[\left| \frac{(1+o(1))\sqrt{\bar{\xi}_W}}{\int_0^\infty \text{Poi}^W_{\nu,\alpha}(k)\,d\alpha} \cdot \sum_{i\in V_\nu} \mathbf{1}_{\{\deg(v_i)=k\}} \int_{\frac{i-1}{\sqrt{2|E_\nu|}}}^{\frac{i}{\sqrt{2|E_\nu|}}} \sum_{x\in\mathcal{X}} f(x,\alpha)\hat{\mu}^\nu_{\alpha,0}(x)\,d\alpha \right.\right.$$

$$\left.\left. - \frac{\sqrt{\bar{\xi}_W}}{\int_0^\infty \text{Poi}^W_{\nu,\alpha}(k)d\alpha} \cdot \int_0^\infty \text{Poi}^W_{\nu,\alpha}(k) \sum_{x\in\mathcal{X}} f(x,\alpha)\hat{\mu}^\nu_{\alpha,0}(x)\,d\alpha \right|\right]$$

$$+ \mathbb{E}\left[\left| \frac{\sqrt{\bar{\xi}_W}}{\int_0^\infty \text{Poi}^W_{\nu,\alpha}(k)d\alpha} \cdot \int_0^\infty \text{Poi}^W_{\nu,\alpha}(k) \sum_{x\in\mathcal{X}} f(x,\alpha)\hat{\mu}^\nu_{\alpha,0}(x)\,d\alpha \right.\right.$$

$$\left.\left. - \frac{\sqrt{\bar{\xi}_W}}{\int_0^\infty \text{Poi}^W_{\nu,\alpha}(k)d\alpha} \cdot \int_0^\infty \text{Poi}^W_{\nu,\alpha}(k) \sum_{x\in\mathcal{X}} f(x,\alpha)\mu^k_{\alpha,0}(x)\,d\alpha \right|\right]$$

$$= \frac{\sqrt{\bar{\xi}_W}}{\int_0^\infty \text{Poi}^W_{\nu,\alpha}(k)\,d\alpha} \cdot \mathbb{E}\left[\left| \sum_{i\in V_\nu} \mathbf{1}_{\{\deg(v_i)=k\}} \int_{\frac{i-1}{\sqrt{2|E_\nu|}}}^{\frac{i}{\sqrt{2|E_\nu|}}} \sum_{x\in\mathcal{X}} f(x,\alpha)\hat{\mu}^\nu_{\alpha,t}(x)\,d\alpha \right.\right.$$

$$\left.\left. - \int_0^\infty \text{Poi}^W_{\nu,\alpha}(k) \sum_{x\in\mathcal{X}} f(x,\alpha)\hat{\mu}^\nu_{\alpha,t}(x)\,d\alpha \right|\right] + o(1) = o(1)$$

where the second summand goes to zero by a standard LLN argument. To see the convergence of the first summand, we keep in mind Assumption 1 and recall Caron et al. (2022, Proposition 1) which states that

$$|E_\nu| = (1 + o(1))\nu^2 \frac{\bar{\xi}_W}{2}.$$

Furthermore, by Caron et al. (2022, Theorem 2) and Veitch & Roy (2015, Theorem 5.5) we know that

$$|V_{\nu,k}| = (1 + o(1))\nu \int_0^\infty \text{Poi}_{\nu,\alpha}^W(k) \, d\alpha \, .$$

Combining these insights, we obtain

$$\frac{\sqrt{2|E_\nu|}}{|V_{\nu,k}|} = \frac{(1 + o(1))\nu\sqrt{\bar\xi_W}}{(1 + o(1))\nu \int_0^\infty \text{Poi}_{\nu,\alpha}^W(k) \, d\alpha} = (1 + o(1)) \frac{\sqrt{\bar\xi_W}}{\int_0^\infty \text{Poi}_{\nu,\alpha}^W(k) \, d\alpha} \, .$$

Thus, by Kolmogorov's strong law of large numbers (see, e.g. Feller (1991) for details) and Veitch & Roy (2015, Lemma 5.1) we have

$$\frac{\sqrt{\bar\xi_W}}{\int_0^\infty \text{Poi}_{\nu,\alpha}^W(k) d\alpha} \cdot \mathbb{E}\left[\left|\sum_{i \in V_\nu} \mathbf{1}_{\{\deg(v_i)=k\}} \int_{\frac{i-1}{\sqrt{2|E_\nu|}}}^{\frac{i}{\sqrt{2|E_\nu|}}} \sum_{x \in \mathcal{X}} f(x, \alpha)\hat\mu_{\alpha,t}^\nu(x) \, d\alpha\right.\right.$$
$$\left.\left. - \int_{\frac{i-1}{\sqrt{2|E_\nu|}}}^{\frac{i}{\sqrt{2|E_\nu|}}} \text{Poi}_{\nu,\alpha}^W(k) \sum_{x \in \mathcal{X}} f(x, \alpha)\hat\mu_{\alpha,t}^\nu(x) \, d\alpha\right|\right] = o(1)$$

which concludes the induction start.

Now, it remains to show the induction step which we do by leveraging the following upper bound on the term of interest.

$$\mathbb{E}\left[\left|\hat{\boldsymbol{\mu}}_{t+1}^{\nu,k}(f) - \boldsymbol{\mu}_{t+1}^k(f, \nu)\right|\right] \leq \mathbb{E}\left[\left|\hat{\boldsymbol{\mu}}_{t+1}^{\nu,k}(f) - \hat{\boldsymbol{\mu}}_t^{\nu,k} P_{t,\hat{\boldsymbol{\mu}},\widehat{W}}^{\hat{\boldsymbol{\pi}},\infty}(f)\right|\right]$$
$$+ \mathbb{E}\left[\left|\hat{\boldsymbol{\mu}}_t^{\nu,k} P_{t,\hat{\boldsymbol{\mu}},\widehat{W}}^{\hat{\boldsymbol{\pi}},\infty}(f) - \hat{\boldsymbol{\mu}}_t^{\nu,k} P_{t,\hat{\boldsymbol{\mu}},\widehat{W}}^{\hat{\boldsymbol{\pi}},k}(f)\right|\right]$$
$$+ \mathbb{E}\left[\left|\hat{\boldsymbol{\mu}}_t^{\nu,k} P_{t,\hat{\boldsymbol{\mu}},\widehat{W}}^{\hat{\boldsymbol{\pi}},k}(f) - \hat{\boldsymbol{\mu}}_t^{\nu,k} P_{t,\hat{\boldsymbol{\mu}},W}^{\hat{\boldsymbol{\pi}},k}(f)\right|\right]$$
$$+ \mathbb{E}\left[\left|\hat{\boldsymbol{\mu}}_t^{\nu,k} P_{t,\hat{\boldsymbol{\mu}},W}^{\hat{\boldsymbol{\pi}},k}(f) - \hat{\boldsymbol{\mu}}_t^{\nu,k} P_{t,\hat{\boldsymbol{\mu}},W}^{\boldsymbol{\pi},k}(f)\right|\right]$$
$$+ \mathbb{E}\left[\left|\hat{\boldsymbol{\mu}}_t^{\nu,k} P_{t,\hat{\boldsymbol{\mu}},W}^{\boldsymbol{\pi},k}(f) - \hat{\boldsymbol{\mu}}_t^{\nu,k} P_{t,\boldsymbol{\mu},W}^{\boldsymbol{\pi},k}(f)\right|\right]$$
$$+ \mathbb{E}\left[\left|\hat{\boldsymbol{\mu}}_t^{\nu,k} P_{t,\boldsymbol{\mu},W}^{\boldsymbol{\pi},k}(f) - \boldsymbol{\mu}_{t+1}^k(f, \nu)\right|\right] \, .$$

**First term.** By the law of total expectation we have

$$\mathbb{E}\left[\left|\hat{\boldsymbol{\mu}}_{t+1}^{\nu,k}(f) - \hat{\boldsymbol{\mu}}_t^{\nu,k} P_{t,\hat{\boldsymbol{\mu}},\widehat{W}}^{\hat{\boldsymbol{\pi}},\infty}(f)\right|\right]$$

$$= \mathbb{E}\left[\left|\frac{\sqrt{2|E_\nu|}}{|V_{\nu,k}|}\left(\sum_{i \in V_\nu} \mathbf{1}_{\{\deg(v_i)=k\}} \int_{\frac{i-1}{\sqrt{2|E_\nu|}}}^{\frac{i}{\sqrt{2|E_\nu|}}} \sum_{x \in \mathcal{X}} f(x, \alpha)\hat\mu_{\alpha,t+1}^\nu(x) \, d\alpha\right.\right.\right.$$

$$- \sum_{i \in V_\nu} \mathbf{1}_{\{\deg(v_i)=k\}} \int_{\frac{i-1}{\sqrt{2|E_\nu|}}}^{\frac{i}{\sqrt{2|E_\nu|}}} \sum_{x \in \mathcal{X}} \hat\mu_{\alpha,t}^\nu(x) \sum_{u \in \mathcal{U}} \hat\pi_{\alpha,t}(u \mid x)$$

$$\left.\left.\left. \cdot \sum_{x' \in \mathcal{X}} P\left(x' \mid x, u, \frac{1}{\xi_{\widehat{W},\alpha^*}(\alpha)} \int_0^{\alpha^*} \widehat{W}(\alpha, \beta)\hat\mu_{\beta,t} \, d\beta\right) f(x', \alpha) \, d\alpha\right)\right|\right]$$

$$= \varepsilon_{\alpha^*} O(1) + \mathbb{E}\left[\left|\frac{\sqrt{2|E_\nu|}}{|V_{\nu,k}|}\left(\sum_{i \in V_\nu} \mathbf{1}_{\{\deg(v_i)=k\}} \int_{\frac{i-1}{\sqrt{2|E_\nu|}}}^{\frac{i}{\sqrt{2|E_\nu|}}} \sum_{x \in \mathcal{X}} f(x, \alpha)\hat\mu_{\alpha,t+1}^\nu(x) \, d\alpha\right.\right.\right.$$

$$- \sum_{i \in V_\nu} \mathbf{1}_{\{\deg(v_i)=k\}} \int_{\frac{i-1}{\sqrt{2|E_\nu|}}}^{\frac{i}{\sqrt{2|E_\nu|}}} \sum_{x \in \mathcal{X}} \hat\mu_{\alpha,t}^\nu(x) \sum_{u \in \mathcal{U}} \hat\pi_{\alpha,t}(u \mid x)$$

$$\left.\left.\left. \cdot \sum_{x' \in \mathcal{X}} P\left(x' \mid x, u, \frac{1}{\xi_{\widehat{W}}(\alpha)} \int_{\mathbb{R}_+} \widehat{W}(\alpha, \beta)\hat\mu_{\beta,t} \, d\beta\right) f(x', \alpha) \, d\alpha\right)\right|\right]$$

$$
= \varepsilon_{\alpha^*} O(1) + \frac{\sqrt{2|E_\nu|}}{|V_{\nu,k}|} \, \mathbb{E}\Bigg[ \Bigg| \sum_{i \in V_\nu} \mathbf{1}_{\{\deg(v_i)=k\}} \left( \int_{\frac{i-1}{\sqrt{2|E_\nu|}}}^{\frac{i}{\sqrt{2|E_\nu|}}} f(X_{t+1}^i, \alpha) \, d\alpha \right.
$$

$$
\left. - \, \mathbb{E}\left[ \int_{\frac{i-1}{\sqrt{2|E_\nu|}}}^{\frac{i}{\sqrt{2|E_\nu|}}} f(X_{t+1}^i, \alpha) \, d\alpha \, \Bigg| \, \mathbf{X}_t \right] \right) \Bigg| \Bigg]
$$

$$
\leq \varepsilon_{\alpha^*} O(1) + \frac{\sqrt{2|E_\nu|}}{|V_{\nu,k}|} \, \mathbb{E}\Bigg[ \Bigg( \sum_{i \in V_\nu} \mathbf{1}_{\{\deg(v_i)=k\}} \left( \int_{\frac{i-1}{\sqrt{2|E_\nu|}}}^{\frac{i}{\sqrt{2|E_\nu|}}} f(X_{t+1}^i, \alpha) \, d\alpha \right.
$$

$$
\left. - \, \mathbb{E}\left[ \int_{\frac{i-1}{\sqrt{2|E_\nu|}}}^{\frac{i}{\sqrt{2|E_\nu|}}} f(X_{t+1}^i, \alpha) \, d\alpha \, \Bigg| \, \mathbf{X}_t \right] \right) \Bigg)^2 \Bigg]^{\frac{1}{2}}
$$

$$
= \varepsilon_{\alpha^*} O(1) + \frac{\sqrt{2|E_\nu|}}{|V_{\nu,k}|} \, \mathbb{E}\Bigg[ \sum_{i \in V_\nu} \mathbf{1}_{\{\deg(v_i)=k\}} \left( \int_{\frac{i-1}{\sqrt{2|E_\nu|}}}^{\frac{i}{\sqrt{2|E_\nu|}}} f(X_{t+1}^i, \alpha) \, d\alpha \right.
$$

$$
\left. - \, \mathbb{E}\left[ \int_{\frac{i-1}{\sqrt{2|E_\nu|}}}^{\frac{i}{\sqrt{2|E_\nu|}}} f(X_{t+1}^i, \alpha) \, d\alpha \, \Bigg| \, \mathbf{X}_t \right] \right)^2 \Bigg]^{\frac{1}{2}}
$$

$$
\leq \varepsilon_{\alpha^*} O(1) + \frac{\sqrt{2|E_\nu|}}{|V_{\nu,k}|} \left( \sum_{i \in V_\nu} \mathbf{1}_{\{\deg(v_i)=k\}} \left( \frac{2M_f}{\sqrt{2|E_\nu|}} \right)^2 \right)^{\frac{1}{2}}
$$

$$
= \varepsilon_{\alpha^*} O(1) + \frac{2M_f \sqrt{2|E_\nu|}}{|V_{\nu,k}|} \cdot \left( \frac{|V_{\nu,k}|}{2|E_\nu|} \right)^{\frac{1}{2}} = \varepsilon_{\alpha^*} O(1) + \underbrace{\frac{2M_f}{\sqrt{|V_{\nu,k}|}}}_{\to 0 \text{ as } \nu \to \infty} \leq \varepsilon
$$

for $\alpha^*$ and $\nu$ large enough, where we point out that the $\{X_{t+1}^i\}_{i \in V_\nu}$ are independent if conditioned on $\mathbf{X}_t \equiv \{X_t^i\}_{i \in V_\nu}$.

**Second term.** The second term can be bounded by

$$
\mathbb{E}\left[ \left| \hat{\boldsymbol{\mu}}_t^{\nu,k} P_{t,\hat{\boldsymbol{\mu}},\widehat{W}}^{\hat{\boldsymbol{\pi}},\infty}(f) - \hat{\boldsymbol{\mu}}_t^{\nu,k} P_{t,\hat{\boldsymbol{\mu}},\widehat{W}}^{\hat{\boldsymbol{\pi}},k}(f) \right| \right]
$$

$$
= \frac{\sqrt{2|E_\nu|}}{|V_{\nu,k}|} \, \mathbb{E}\Bigg[ \Bigg| \sum_{i \in V_\nu} \mathbf{1}_{\{\deg(v_i)=k\}} \int_{\frac{i-1}{\sqrt{2|E_\nu|}}}^{\frac{i}{\sqrt{2|E_\nu|}}} \sum_{x \in \mathcal{X}} \hat{\mu}_{\alpha,t}^\nu(x) \sum_{u \in \mathcal{U}} \hat{\pi}_{\alpha,t}(u \mid x)
$$

$$
\cdot \sum_{x' \in \mathcal{X}} P\left( x' \mid x, u, \frac{1}{\xi_{\widehat{W},\alpha^*}(\alpha)} \int_0^{\alpha^*} \widehat{W}(\alpha, \beta) \hat{\mu}_{\beta,t}^\nu \, d\beta \right) f(x', \alpha) \, d\alpha
$$

$$
- \sum_{i \in V_\nu} \mathbf{1}_{\{\deg(v_i)=k\}} \int_{\frac{i-1}{\sqrt{2|E_\nu|}}}^{\frac{i}{\sqrt{2|E_\nu|}}} \sum_{x \in \mathcal{X}} \hat{\mu}_{\alpha,t}^\nu(x) \sum_{j=0}^{k} P_{\hat{\boldsymbol{\pi}}}\left( \widehat{\mathbb{G}}_{\alpha,t}^{\nu,k}(\hat{\boldsymbol{\mu}}_t^\nu) = j \mid x_{\alpha,t} = x \right) \sum_{u \in \mathcal{U}} \hat{\pi}_{\alpha,t}(u \mid x)
$$

$$
\cdot \sum_{x' \in \mathcal{X}} P\left( x' \mid x, u, \frac{1}{k}(j, k-j) \right) f(x', \alpha) \, d\alpha \Bigg| \Bigg]
$$

$$
< o(1) + O(1)\varepsilon_{\alpha^*} + \frac{\sqrt{2|E_\nu|}}{|V_{\nu,k}|} \, \mathbb{E}\Bigg[ \Bigg| \sum_{i \in V_\nu} \mathbf{1}_{\{\deg(v_i)=k\}} \int_{\frac{i-1}{\sqrt{2|E_\nu|}}}^{\frac{i}{\sqrt{2|E_\nu|}}} \sum_{x \in \mathcal{X}} \hat{\mu}_{\alpha,t}^\nu(x) \sum_{u \in \mathcal{U}} \hat{\pi}_{\alpha,t}(u \mid x)
$$

$$
\cdot \sum_{x' \in \mathcal{X}} P\left( x' \mid x, u, \frac{1}{\xi_{\widehat{W}}(\alpha)} \int_{\mathbb{R}_+} \widehat{W}(\alpha, \beta) \hat{\mu}_{\beta,t}^\nu \, d\beta \right) f(x', \alpha) \, d\alpha
$$

$$- \sum_{i \in V_\nu} \mathbf{1}_{\{\deg(v_i)=k\}} \int_{\frac{i-1}{\sqrt{2|E_\nu|}}}^{\frac{i}{\sqrt{2|E_\nu|}}} \sum_{x \in \mathcal{X}} \hat{\mu}_{\alpha,t}^\nu(x) \sum_{j=0}^k P_{\hat{\boldsymbol{\pi}}}^\nu \left( \widehat{\mathbb{G}}_{i,t}^{\nu,k}(\hat{\boldsymbol{\mu}}_t^\nu) = j \mid x_{i,t} = x \right) \sum_{u \in \mathcal{U}} \hat{\pi}_{\alpha,t}(u \mid x)$$

$$\cdot \sum_{x' \in \mathcal{X}} P\left( x' \mid x, u, \frac{1}{k}(j, k-j) \right) f(x', \alpha) \, d\alpha \Bigg| \Bigg]$$

$$= o(1) + O(1)\varepsilon_{\alpha^*} \leq \varepsilon$$

for $\alpha^*$ and $\nu$ large enough, where the inequality above leverages inequality (3).

**Third term.** This term can be reformulated as

$$\mathbb{E}\left[ \left| \hat{\boldsymbol{\mu}}_t^{\nu,k} P_{t,\hat{\boldsymbol{\mu}},\widehat{W}}^{\hat{\boldsymbol{\pi}},k}(f) - \hat{\boldsymbol{\mu}}_t^{\nu,k} P_{t,\hat{\boldsymbol{\mu}},W}^{\hat{\boldsymbol{\pi}},k}(f) \right| \right]$$

$$= \frac{\sqrt{2|E_\nu|}}{|V_{\nu,k}|} \mathbb{E}\left[ \left| \sum_{i \in V_\nu} \mathbf{1}_{\{\deg(v_i)=k\}} \int_{\frac{i-1}{\sqrt{2|E_\nu|}}}^{\frac{i}{\sqrt{2|E_\nu|}}} \sum_{j=0}^k \sum_{x \in \mathcal{X}} P_{\hat{\boldsymbol{\pi}},\hat{\boldsymbol{\mu}}} \left( \widehat{\mathbb{G}}_{\alpha,t}^{\nu,k}(\hat{\boldsymbol{\mu}}_t^\nu) = j, \hat{x}_{\alpha,t} = x \right) \right. \right.$$

$$\cdot \sum_{u \in \mathcal{U}} \hat{\pi}_{\alpha,t}(u \mid x) \sum_{x' \in \mathcal{X}} P\left( x' \mid x, u, \frac{1}{k}(j, k-j) \right) f(x', \alpha) \, d\alpha$$

$$- \frac{\sqrt{2|E_\nu|}}{|V_{\nu,k}|} \sum_{j=0}^k \sum_{x \in \mathcal{X}} P_{\hat{\boldsymbol{\pi}},\hat{\boldsymbol{\mu}}} \left( \mathbb{G}_{\alpha,t}^k(\hat{\boldsymbol{\mu}}_t^\nu) = j, \hat{x}_{\alpha,t} = x \right) \sum_{u \in \mathcal{U}} \hat{\pi}_{\alpha,t}(u \mid x)$$

$$\left. \left. \cdot \sum_{x' \in \mathcal{X}} P\left( x' \mid x, u, \frac{1}{k}(j, k-j) \right) f(x', \alpha) \, d\alpha \right| \right]$$

$$\leq \frac{M_f \sqrt{2|E_\nu|}}{|V_{\nu,k}|} \mathbb{E}\left[ \sum_{i \in V_\nu} \mathbf{1}_{\{\deg(v_i)=k\}} \int_{\frac{i-1}{\sqrt{2|E_\nu|}}}^{\frac{i}{\sqrt{2|E_\nu|}}} \sum_{j=0}^k \sum_{x \in \mathcal{X}} \left| P_{\hat{\boldsymbol{\pi}},\hat{\boldsymbol{\mu}}} \left( \widehat{\mathbb{G}}_{\alpha,t}^{\nu,k}(\hat{\boldsymbol{\mu}}_t^\nu) = j, \hat{x}_{\alpha,t} = x \right) \right. \right.$$

$$\left. \left. - P_{\hat{\boldsymbol{\pi}},\hat{\boldsymbol{\mu}}} \left( \mathbb{G}_{\alpha,t}^k(\hat{\boldsymbol{\mu}}_t^\nu) = j, \hat{x}_{\alpha,t} = x \right) \right| \, d\alpha \right].$$

We focus on

$$\sum_{j=0}^k \sum_{x \in \mathcal{X}} \mathbb{E}\left[ \left| P_{\hat{\boldsymbol{\pi}},\hat{\boldsymbol{\mu}}} \left( \widehat{\mathbb{G}}_{\alpha,t}^{\nu,k}(\hat{\boldsymbol{\mu}}_t^\nu) = j, \hat{x}_{\alpha,t} = x \right) - P_{\hat{\boldsymbol{\pi}},\hat{\boldsymbol{\mu}}} \left( \mathbb{G}_{\alpha,t}^k(\hat{\boldsymbol{\mu}}_t^\nu) = j, \hat{x}_{\alpha,t} = x \right) \right| \right]$$

and recall

$$P_{\hat{\boldsymbol{\pi}},\hat{\boldsymbol{\mu}}} \left( \mathbb{G}_{\alpha,t+1}^k(\hat{\boldsymbol{\mu}}_t^\nu) = j, \hat{x}_{\alpha,t+1} = x \right)$$

$$= \int_{[0,\alpha^*]^k} \sum_{x' \in \mathcal{X}} \sum_{j'=0}^k \sum_{u \in \mathcal{U}} \sum_{\ell=\max\{0,j+j'-k\}}^{\min\{j,j'\}} \frac{\sum_{(y_1,\dots y_k) \in \mathcal{X}_j^k} \prod_{i=1}^k \frac{W(\alpha,\beta_i)}{\xi_W(\alpha)} \hat{\mu}_{\beta_i,t}^\nu(y_i)}{\int_{[0,\alpha^*]^k} \sum_{(z_1,\dots z_k) \in \mathcal{X}_j^k} \prod_{i=1}^k \frac{W(\alpha,\beta_i')}{\xi_W(\alpha)} \hat{\mu}_{\beta_i',t}^\nu(z_i) \, d\beta'}$$

$$\cdot P_{\hat{\boldsymbol{\pi}},\hat{\boldsymbol{\mu}}} \left( \mathbb{G}_{\alpha,t}^k(\hat{\boldsymbol{\mu}}_t^\nu) = j', x_{\alpha,t} = x' \right) \hat{\pi}_{\alpha,t}(u \mid x') \cdot P\left( x \mid x', u, \frac{j'}{k} \right)$$

$$\cdot \binom{j'}{\ell} \left( \sum_{u \in \mathcal{U}} \hat{\pi}_{\beta_i,t}(u \mid s_1) \cdot P\left( s_1 \mid s_1, u, \mathbb{G}_{\beta_i,t}^\infty(\hat{\boldsymbol{\mu}}_t^\nu) \right) \right)^\ell$$

$$\cdot \left( \sum_{u \in \mathcal{U}} \hat{\pi}_{\beta_i,t}(u \mid s_2) \cdot P\left( s_1 \mid s_2, u, \mathbb{G}_{\beta_i,t}^\infty(\hat{\boldsymbol{\mu}}_t^\nu) \right) \right)^{j-\ell}$$

$$\cdot \binom{k-j'}{j-\ell} \left( \sum_{u \in \mathcal{U}} \hat{\pi}_{\beta_i,t}(u \mid s_1) \cdot P\left( s_2 \mid s_1, u, \mathbb{G}_{\beta_i,t}^\infty(\hat{\boldsymbol{\mu}}_t^\nu) \right) \right)^{j'-\ell}$$

$$\cdot \left( \sum_{u \in \mathcal{U}} \hat{\pi}_{\beta_i,t}(u \mid s_2) \cdot P\left( s_2 \mid s_2, u, \mathbb{G}_{\beta_i,t}^\infty(\hat{\boldsymbol{\mu}}_t^\nu) \right) \right)^{k-j'-j+\ell} \, d\beta_1 \cdots d\beta_k.$$

For sake of simplicity, we introduce the following auxiliary notations (where we fix some $x', j', u, \ell$):

$$
\text{(I)} = P_{\hat{\boldsymbol{\pi}},\hat{\boldsymbol{\mu}}}\left(\mathbb{G}_{\alpha,t}^{k}\left(\hat{\boldsymbol{\mu}}_{t}^{\nu}\right) = j', x_{\alpha,t} = x'\right) \hat{\pi}_{\alpha,t}\left(u \mid x'\right) \cdot P\left(x \mid x', u, \frac{j'}{k}\right)
$$
$$
\cdot \frac{\sum_{(y_1,\dots y_k)\in\mathcal{X}_j^k} \prod_{i=1}^{k} \frac{W(\alpha,\beta_i)}{\xi_W(\alpha)} \hat{\mu}_{\beta_i,t}^{\nu}(y_i)}{\int_{[0,\alpha^*]^k}\sum_{(z_1,\dots z_k)\in\mathcal{X}_j^k} \prod_{i=1}^{k} \frac{W(\alpha,\beta_i')}{\xi_W(\alpha)} \hat{\mu}_{\beta_i',t}^{\nu}(z_i)\,\mathrm{d}\beta'}
$$
$$
\text{(II)} = \binom{j'}{\ell}\left(\sum_{u\in\mathcal{U}} \hat{\pi}_{\beta_i,t}\left(u \mid s_1\right)\cdot P\left(s_1 \mid s_1, u, \mathbb{G}_{\beta_i,t}^{\infty}\left(\hat{\boldsymbol{\mu}}_t^{\nu}\right)\right)\right)^{\ell}
$$
$$
\text{(III)} = \left(\sum_{u\in\mathcal{U}} \hat{\pi}_{\beta_i,t}\left(u \mid s_2\right)\cdot P\left(s_1 \mid s_2, u, \mathbb{G}_{\beta_i,t}^{\infty}\left(\hat{\boldsymbol{\mu}}_t^{\nu}\right)\right)\right)^{j-\ell}
$$
$$
\text{(IV)} = \binom{k-j'}{j-\ell}\left(\sum_{u\in\mathcal{U}} \hat{\pi}_{\beta_i,t}\left(u \mid s_1\right)\cdot P\left(s_2 \mid s_1, u, \mathbb{G}_{\beta_i,t}^{\infty}\left(\hat{\boldsymbol{\mu}}_t^{\nu}\right)\right)\right)^{j'-\ell}
$$
$$
\text{(V)} = \left(\sum_{u\in\mathcal{U}} \hat{\pi}_{\beta_i,t}\left(u \mid s_2\right)\cdot P\left(s_2 \mid s_2, u, \mathbb{G}_{\beta_i,t}^{\infty}\left(\hat{\boldsymbol{\mu}}_t^{\nu}\right)\right)\right)^{k-j'-j+\ell}
$$

such that

$$
P_{\hat{\boldsymbol{\pi}},\hat{\boldsymbol{\mu}}}\left(\mathbb{G}_{i,t+1}^{k}\left(\hat{\boldsymbol{\mu}}_t^{\nu}\right) = j, x_{1,t+1} = x\right)
$$
$$
= \int_{[0,\alpha^*]^k} \sum_{x'\in\mathcal{X}} \sum_{j'=0}^{k} \sum_{u\in\mathcal{U}} \sum_{\ell=\max\{0,j+j'-k\}}^{\min\{j,j'\}} \text{(I)}\cdot\text{(II)}\cdot\text{(III)}\cdot\text{(IV)}\cdot\text{(V)}\,\mathrm{d}\beta_1\cdots\mathrm{d}\beta_k\,.
$$

Analogously, we define $\text{(I}'), \text{(II}'), \text{(III}'), \text{(IV}'), \text{(V}')$ such that

$$
P_{\hat{\boldsymbol{\pi}},\hat{\boldsymbol{\mu}}}\left(\widehat{\mathbb{G}}_{i,t+1}^{\nu,k}\left(\hat{\boldsymbol{\mu}}_t^{\nu}\right) = j, x_{1,t+1} = x\right)
$$
$$
= \int_{[0,\alpha^*]^k} \sum_{x'\in\mathcal{X}} \sum_{j'=0}^{k} \sum_{u\in\mathcal{U}} \sum_{\ell=\max\{0,j+j'-k\}}^{\min\{j,j'\}} \text{(I}')\cdot\text{(II}')\cdot\text{(III}')\cdot\text{(IV}')\cdot\text{(V}')\,\mathrm{d}\beta_1\cdots\mathrm{d}\beta_k\,.
$$

Thus, using a telescope sum we can reformulate the term of interest as

$$
\sum_{j=0}^{k} \sum_{x\in\mathcal{X}} \mathbb{E}\left[\left|P_{\hat{\boldsymbol{\pi}},\hat{\boldsymbol{\mu}}}\left(\widehat{\mathbb{G}}_{\alpha,t}^{\nu,k}\left(\hat{\boldsymbol{\mu}}_t^{\nu}\right) = j, \hat{x}_{\alpha,t} = x\right) - P_{\hat{\boldsymbol{\pi}},\hat{\boldsymbol{\mu}}}\left(\mathbb{G}_{\alpha,t}^{k}\left(\hat{\boldsymbol{\mu}}_t^{\nu}\right) = j, \hat{x}_{\alpha,t} = x\right)\right|\right]
$$
$$
\leq \sum_{j=0}^{k} \sum_{x\in\mathcal{X}} \int_{[0,\alpha^*]^k} \sum_{x'\in\mathcal{X}} \sum_{j'=0}^{k} \sum_{u\in\mathcal{U}} \sum_{\ell=\max\{0,j+j'-k\}}^{\min\{j,j'\}} \mathbb{E}\left[|\text{(I)}-\text{(I}')|\cdot\text{(II)}\cdot\text{(III)}\cdot\text{(IV)}\cdot\text{(V)}\right]
$$
$$
+ \mathbb{E}\left[|\text{(II)}-\text{(II}')|\cdot\text{(I}')\cdot\text{(III)}\cdot\text{(IV)}\cdot\text{(V)}\right] + \mathbb{E}\left[|\text{(III)}-\text{(III}')|\cdot\text{(I}')\cdot\text{(II}')\cdot\text{(IV)}\cdot\text{(V)}\right]
$$
$$
+ \mathbb{E}\left[|\text{(IV)}-\text{(IV}')|\cdot\text{(I}')\cdot\text{(II}')\cdot\text{(III}')\cdot\text{(V)}\right] + \mathbb{E}\left[|\text{(V)}-\text{(V}')|\cdot\text{(I}')\cdot\text{(II}')\cdot\text{(III}')\cdot\text{(IV}')\right]
$$
$$
\mathrm{d}\beta_1\cdots\mathrm{d}\beta_k
$$
$$
\leq \sum_{j,j'=0}^{k} \sum_{x\in\mathcal{X}} \int_{[0,\alpha^*]^k} \sum_{x'\in\mathcal{X}} \sum_{u\in\mathcal{U}} \sum_{\ell=\max\{0,j+j'-k\}}^{\min\{j,j'\}} \mathbb{E}\left[|\text{(I)}-\text{(I}')|\right]\cdot O(1) + \mathbb{E}\left[|\text{(II)}-\text{(II}')|\right]\cdot O(1)
$$
$$
+ \mathbb{E}\left[|\text{(III)}-\text{(III}')|\right]\cdot O(1) + \mathbb{E}\left[|\text{(IV)}-\text{(IV}')|\right]\cdot O(1) + \mathbb{E}\left[|\text{(V)}-\text{(V}')|\right]\cdot O(1)\mathrm{d}\beta_1\cdots\mathrm{d}\beta_k
$$

The first term converges to 0 for $\nu\to\infty$ by induction, i.e.

$$
\mathbb{E}\left[|\text{(I)}-\text{(I}')|\right]\cdot O(1) = o(1)\,.
$$

Since the remaining four terms are structurally very similar, we just prove convergence for the second term and point out that convergence for the terms three to five can be established analogously. Thus,

for the second term we have

$$
\mathbb{E}\left[|(\mathrm{II}) - (\mathrm{II}')|\right] \cdot O(1) = \mathbb{E}\left[\left|\left(\sum_{u \in \mathcal{U}} \hat{\pi}_{\beta_i,t}\left(u \mid s_1\right) \cdot P\left(s_1 \mid s_1, u, \mathbb{G}_{\beta_i,t}^{\infty}\left(\hat{\boldsymbol{\mu}}_t^{\nu}\right)\right)\right)^{\ell}\right.\right.
$$
$$
\left.\left. - \left(\sum_{u \in \mathcal{U}} \hat{\pi}_{\beta_i,t}\left(u \mid s_1\right) \cdot P\left(s_1 \mid s_1, u, \widehat{\mathbb{G}}_{\beta_i,t}^{\nu,\infty}\left(\hat{\boldsymbol{\mu}}_t^{\nu}\right)\right)\right)^{\ell}\right|\right] \cdot O(1).
$$

We briefly recall that for arbitrary $p, q < \infty$ and finite $\ell \in \mathbb{N}$ we have

$$
p^{\ell} - q^{\ell} = (p-q)p^{\ell-1} + qp^{\ell-1} - q^{\ell} = \ldots = (p-q)\sum_{j=1}^{\ell} p^{\ell-j}q^{j-1} = (p-q)O(1) \qquad (6)
$$

which brings us to

$$
\mathbb{E}\left[|(\mathrm{II}) - (\mathrm{II}')|\right] \cdot O(1)
$$
$$
= \mathbb{E}\left[\left|\left(\sum_{u \in \mathcal{U}} \hat{\pi}_{\beta_i,t}\left(u \mid s_1\right) \cdot P\left(s_1 \mid s_1, u, \mathbb{G}_{\beta_i,t}^{\nu,\infty}\left(\hat{\boldsymbol{\mu}}_t^{\nu}\right)\right)\right)^{\ell}\right.\right.
$$
$$
\left.\left. - \left(\sum_{u \in \mathcal{U}} \hat{\pi}_{\beta_i,t}\left(u \mid s_1\right) \cdot P\left(s_1 \mid s_1, u, \widehat{\mathbb{G}}_{\beta_i,t}^{\nu,\infty}\left(\hat{\boldsymbol{\mu}}_t^{\nu}\right)\right)\right)^{\ell}\right|\right] \cdot O(1)
$$
$$
= \mathbb{E}\left[\left|\sum_{u \in \mathcal{U}} \hat{\pi}_{\beta_i,t}\left(u \mid s_1\right) \cdot \left(P\left(s_1 \mid s_1, u, \mathbb{G}_{\beta_i,t}^{\nu,\infty}\left(\hat{\boldsymbol{\mu}}_t^{\nu}\right)\right) - P\left(s_1 \mid s_1, u, \widehat{\mathbb{G}}_{\beta_i,t}^{\nu,\infty}\left(\hat{\boldsymbol{\mu}}_t^{\nu}\right)\right)\right)\right|\right] \cdot O(1)
$$
$$
\leq \sum_{u \in \mathcal{U}} \hat{\pi}_{\beta_i,t}\left(u \mid s_1\right) \cdot \mathbb{E}\left[\left|P\left(s_1 \mid s_1, u, \mathbb{G}_{\beta_i,t}^{\nu,\infty}\left(\hat{\boldsymbol{\mu}}_t^{\nu}\right)\right) - P\left(s_1 \mid s_1, u, \widehat{\mathbb{G}}_{\beta_i,t}^{\nu,\infty}\left(\hat{\boldsymbol{\mu}}_t^{\nu}\right)\right)\right|\right] \cdot O(1)
$$
$$
\leq o(1) + \varepsilon_{\alpha^*} \cdot O(1).
$$

Combining all of these findings, we eventually arrive at

$$
\mathbb{E}\left[\left|\hat{\boldsymbol{\mu}}_t^{\nu,k} P_{t,\hat{\boldsymbol{\mu}},\widehat{W}}^{\hat{\boldsymbol{\pi}},k}(f) - \hat{\boldsymbol{\mu}}_t^{\nu,k} P_{t,\hat{\boldsymbol{\mu}},W}^{\hat{\boldsymbol{\pi}},k}(f)\right|\right]
$$
$$
\leq \frac{M_f \sqrt{2|E_{\nu}|}}{|V_{\nu,k}|} \mathbb{E}\left[\sum_{i \in V_{\nu}} \mathbf{1}_{\{\deg(v_i)=k\}} \int_{\frac{i-1}{\sqrt{2|E_{\nu}|}}}^{\frac{i}{\sqrt{2|E_{\nu}|}}} \sum_{j=0}^{k} \sum_{x \in \mathcal{X}} \left|P_{\hat{\boldsymbol{\pi}},\hat{\boldsymbol{\mu}}}\left(\widehat{\mathbb{G}}_{\alpha,t}^{\nu,k}\left(\hat{\boldsymbol{\mu}}_t^{\nu}\right) = j, \hat{x}_{\alpha,t} = x\right)\right.\right.
$$
$$
\left.\left. - P_{\hat{\boldsymbol{\pi}},\hat{\boldsymbol{\mu}}}\left(\mathbb{G}_{\alpha,t}^{\nu,k}\left(\hat{\boldsymbol{\mu}}_t^{\nu}\right) = j, \hat{x}_{\alpha,t} = x\right)\right| \mathrm{d}\alpha\right]
$$
$$
\leq \frac{M_f \sqrt{2|E_{\nu}|}}{|V_{\nu,k}|} \mathbb{E}\left[\sum_{i \in V_{\nu,k}} \int_{\frac{i-1}{\sqrt{2|E_{\nu}|}}}^{\frac{i}{\sqrt{2|E_{\nu}|}}} \left(o(1) + \varepsilon_{\alpha^*} \cdot O(1)\right) \mathrm{d}\alpha\right] = o(1) + \varepsilon_{\alpha^*} \cdot O(1).
$$

**Fourth term.** For the fourth term we have

$$
\mathbb{E}\left[\left|\hat{\boldsymbol{\mu}}_t^{\nu,k} P_{t,\hat{\boldsymbol{\mu}},W}^{\hat{\boldsymbol{\pi}},k}(f) - \hat{\boldsymbol{\mu}}_t^{\nu,k} P_{t,\hat{\boldsymbol{\mu}},W}^{\boldsymbol{\pi},k}(f)\right|\right]
$$
$$
= \mathbb{E}\left[\left|\frac{\sqrt{2|E_{\nu}|}}{|V_{\nu,k}|} \cdot \sum_{i \in V_{\nu,k}} \int_{\frac{i-1}{\sqrt{2|E_{\nu}|}}}^{\frac{i}{\sqrt{2|E_{\nu}|}}} \sum_{j=0}^{k} \sum_{x \in \mathcal{X}} P_{\hat{\boldsymbol{\pi}},\hat{\boldsymbol{\mu}}}\left(\mathbb{G}_{\alpha,t}^{k}\left(\hat{\boldsymbol{\mu}}_t^{\nu}\right) = j, \hat{x}_{\alpha,t} = x\right)\right.\right.
$$
$$
\left. \cdot \sum_{u \in \mathcal{U}} \hat{\pi}_{\alpha,t}\left(u \mid x\right) \cdot P\left(x' \mid x, u, \frac{1}{k}(j, k-j)\right) f(x', \alpha) \, \mathrm{d}\alpha
$$
$$
\left. - \frac{\sqrt{2|E_{\nu}|}}{|V_{\nu,k}|} \cdot \sum_{i \in V_{\nu,k}} \int_{\frac{i-1}{\sqrt{2|E_{\nu}|}}}^{\frac{i}{\sqrt{2|E_{\nu}|}}} \sum_{j=0}^{k} \sum_{x \in \mathcal{X}} P_{\boldsymbol{\pi},\hat{\boldsymbol{\mu}}}\left(\mathbb{G}_{\alpha,t}^{k}\left(\hat{\boldsymbol{\mu}}_t^{\nu}\right) = j, \hat{x}_{\alpha,t} = x\right)\right.
$$

$$\cdot \sum_{u \in \mathcal{U}} \pi_{\alpha,t}^k (u \mid x) \cdot P\left(x' \mid x, u, \frac{1}{k}(j, k-j)\right) f(x', \alpha)\, \mathrm{d}\alpha \Bigg| \Bigg]$$

$$\leq \mathbb{E}\Bigg[ \frac{\sqrt{2|E_\nu|}}{|V_{\nu,k}|} \cdot \sum_{i \in V_{\nu,k}} \int_{\frac{i-1}{\sqrt{2|E_\nu|}}}^{\frac{i}{\sqrt{2|E_\nu|}}} \sum_{j=0}^{k} \sum_{x \in \mathcal{X}} P_{\boldsymbol{\pi}, \hat{\boldsymbol{\mu}}}\left(\mathbb{G}_{\alpha,t}^k (\hat{\boldsymbol{\mu}}_t^\nu) = j, \hat{x}_{\alpha,t} = x\right)$$

$$\cdot \sum_{u \in \mathcal{U}} \left| \pi_{\alpha,t}^k (u \mid x) - \hat{\pi}_{\alpha,t} (u \mid x) \right| \cdot P\left(x' \mid x, u, \frac{1}{k}(j, k-j)\right) f(x', \alpha)\, \mathrm{d}\alpha \Bigg]$$

$$+ \mathbb{E}\Bigg[ \frac{\sqrt{2|E_\nu|}}{|V_{\nu,k}|} \cdot \sum_{i \in V_{\nu,k}} \int_{\frac{i-1}{\sqrt{2|E_\nu|}}}^{\frac{i}{\sqrt{2|E_\nu|}}} \sum_{j=0}^{k} \sum_{x \in \mathcal{X}} \sum_{u \in \mathcal{U}} \hat{\pi}_{\alpha,t} (u \mid x) \cdot P\left(x' \mid x, u, \frac{1}{k}(j, k-j)\right)$$

$$\cdot \left| P_{\hat{\boldsymbol{\pi}}, \hat{\boldsymbol{\mu}}}\left(\mathbb{G}_{\alpha,t}^k (\hat{\boldsymbol{\mu}}_t^\nu) = j, \hat{x}_{\alpha,t} = x\right) - P_{\boldsymbol{\pi}, \hat{\boldsymbol{\mu}}}\left(\mathbb{G}_{\alpha,t}^k (\hat{\boldsymbol{\mu}}_t^\nu) = j, \hat{x}_{\alpha,t} = x\right) \right| f(x', \alpha)\, \mathrm{d}\alpha \Bigg].$$

We start with analyzing the first summand, i.e.

$$\mathbb{E}\Bigg[ \frac{\sqrt{2|E_\nu|}}{|V_{\nu,k}|} \cdot \sum_{i \in V_{\nu,k}} \int_{\frac{i-1}{\sqrt{2|E_\nu|}}}^{\frac{i}{\sqrt{2|E_\nu|}}} \sum_{j=0}^{k} \sum_{x \in \mathcal{X}} P_{\boldsymbol{\pi}, \hat{\mu}}\left(\mathbb{G}_{\alpha,t}^k (\hat{\boldsymbol{\mu}}_t^\nu) = j, \hat{x}_{\alpha,t} = x\right)$$

$$\cdot \sum_{u \in \mathcal{U}} \left| \pi_{\alpha,t}^k (u \mid x) - \hat{\pi}_{\alpha,t} (u \mid x) \right| \cdot P\left(x' \mid x, u, \frac{1}{k}(j, k-j)\right) f(x', \alpha)\, \mathrm{d}\alpha \Bigg]$$

$$\leq \mathbb{E}\Bigg[ \frac{\sqrt{2|E_\nu|}}{|V_{\nu,k}|} \cdot \sum_{\substack{i \in V_{\nu,k} \\ i \neq i'}} \int_{\frac{i-1}{\sqrt{2|E_\nu|}}}^{\frac{i}{\sqrt{2|E_\nu|}}} \sum_{j=0}^{k} \sum_{x \in \mathcal{X}} P_{\boldsymbol{\pi}, \hat{\mu}}\left(\mathbb{G}_{\alpha,t}^k (\hat{\boldsymbol{\mu}}_t^\nu) = j, \hat{x}_{\alpha,t} = x\right)$$

$$\cdot \sum_{u \in \mathcal{U}} \left| \pi_{\frac{i}{\sqrt{2|E_\nu|}}, t}^k (u \mid x) - \hat{\pi}_{\alpha,t} (u \mid x) \right| \cdot P\left(x' \mid x, u, \frac{1}{k}(j, k-j)\right) f(x', \alpha)\, \mathrm{d}\alpha \Bigg]$$

$$+ \frac{2 M_f \int_{\frac{i-1}{\sqrt{2|E_\nu|}}}^{\frac{i}{\sqrt{2|E_\nu|}}} \mathrm{Poi}_{\nu^*, \alpha}^W (k)\, \mathrm{d}\alpha}{\int_0^\infty \mathrm{Poi}_{\nu^*, \alpha}^W (k)\, \mathrm{d}\alpha}$$

$$\leq \mathbb{E}\Bigg[ \frac{\sqrt{2|E_\nu|}}{|V_{\nu,k}|} \cdot \sum_{i \in V_{\nu,k}} \int_{\frac{i-1}{\sqrt{2|E_\nu|}}}^{\frac{i}{\sqrt{2|E_\nu|}}} \frac{L_\pi M_f}{\sqrt{2|E_\nu|}}\, \mathrm{d}\alpha \Bigg] + O\left(\frac{1}{\sqrt{|E_\nu|}}\right)$$

$$= O\left(\frac{1}{\sqrt{|E_\nu|}}\right) + O\left(\frac{1}{\sqrt{|E_\nu|}}\right) = O\left(\frac{1}{\sqrt{|E_\nu|}}\right) \to 0 \qquad \text{as} \quad \nu \to \infty.$$

Now it remains to bound the second summand, which is done as follows

$$\mathbb{E}\Bigg[ \frac{\sqrt{2|E_\nu|}}{|V_{\nu,k}|} \cdot \sum_{i \in V_{\nu,k}} \int_{\frac{i-1}{\sqrt{2|E_\nu|}}}^{\frac{i}{\sqrt{2|E_\nu|}}} \sum_{j=0}^{k} \sum_{x \in \mathcal{X}} \sum_{u \in \mathcal{U}} \hat{\pi}_{\alpha,t} (u \mid x) \cdot P\left(x' \mid x, u, \frac{1}{k}(j, k-j)\right)$$

$$\cdot \left| P_{\hat{\boldsymbol{\pi}}, \hat{\boldsymbol{\mu}}}\left(\mathbb{G}_{\alpha,t}^k (\hat{\boldsymbol{\mu}}_t^\nu) = j, \hat{x}_{\alpha,t} = x\right) - P_{\boldsymbol{\pi}, \hat{\mu}}\left(\mathbb{G}_{\alpha,t}^k (\hat{\boldsymbol{\mu}}_t^\nu) = j, \hat{x}_{\alpha,t} = x\right) \right| f(x', \alpha)\, \mathrm{d}\alpha \Bigg]$$

$$\leq \frac{M_f \sqrt{2|E_\nu|}}{|V_{\nu,k}|} \cdot \sum_{i \in V_{\nu,k}} \int_{\frac{i-1}{\sqrt{2|E_\nu|}}}^{\frac{i}{\sqrt{2|E_\nu|}}} \sum_{j=0}^{k} \sum_{x \in \mathcal{X}} \sum_{u \in \mathcal{U}}$$

$$\cdot \mathbb{E}\left[ \left| P_{\hat{\boldsymbol{\pi}}, \hat{\boldsymbol{\mu}}}\left(\mathbb{G}_{\alpha,t}^k (\hat{\boldsymbol{\mu}}_t^\nu) = j, \hat{x}_{\alpha,t} = x\right) - P_{\boldsymbol{\pi}, \hat{\mu}}\left(\mathbb{G}_{\alpha,t}^k (\hat{\boldsymbol{\mu}}_t^\nu) = j, \hat{x}_{\alpha,t} = x\right) \right| \right]\, \mathrm{d}\alpha.$$

To further analyze

$$\mathbb{E}\left[ \left| P_{\boldsymbol{\pi}, \hat{\mu}}\left(\mathbb{G}_{\alpha,t}^k (\hat{\boldsymbol{\mu}}_t^\nu) = j, \hat{x}_{\alpha,t} = x\right) - P_{\hat{\boldsymbol{\pi}}, \hat{\mu}}\left(\mathbb{G}_{\alpha,t}^k (\hat{\boldsymbol{\mu}}_t^\nu) = j, \hat{x}_{\alpha,t} = x\right) \right| \right]$$

we define for notational convenience (and tacitly assuming that $\alpha \notin \left( \frac{i'-1}{\sqrt{2|E_\nu|}}, \frac{i'}{\sqrt{2|E_\nu|}} \right]$)

$$P_{\boldsymbol{\pi}, \hat{\mu}}\left(\mathbb{G}_{\alpha,t}^k (\hat{\boldsymbol{\mu}}_t^\nu) = j, \hat{x}_{\alpha,t} = x\right)$$

$$= \int_{[0,\alpha^*]^k} \sum_{x'\in\mathcal{X}} \sum_{j'=0}^{k} \sum_{u\in\mathcal{U}} \sum_{\ell=\max\{0,j+j'-k\}}^{\min\{j,j'\}} (\mathrm{I})\cdot(\mathrm{II})\cdot(\mathrm{III})\cdot(\mathrm{IV})\cdot(\mathrm{V})\, \mathrm{d}\beta_1\cdots\mathrm{d}\beta_k$$

and

$$P_{\hat{\boldsymbol{\pi}},\hat{\boldsymbol{\mu}}}\left(\mathbb{G}_{\alpha,t}^k\left(\hat{\boldsymbol{\mu}}_t^\nu\right)=j,\hat{x}_{\alpha,t}=x\right)$$

$$= \int_{[0,\alpha^*]^k} \sum_{x'\in\mathcal{X}} \sum_{j'=0}^{k} \sum_{u\in\mathcal{U}} \sum_{\ell=\max\{0,j+j'-k\}}^{\min\{j,j'\}} (\mathrm{I}')\cdot(\mathrm{II}')\cdot(\mathrm{III}')\cdot(\mathrm{IV}')\cdot(\mathrm{V}')\, \mathrm{d}\beta_1\cdots\mathrm{d}\beta_k$$

with

$$(\mathrm{I}) = P_{\boldsymbol{\pi},\hat{\boldsymbol{\mu}}}\left(\mathbb{G}_{\alpha,t}^k\left(\hat{\boldsymbol{\mu}}_t^\nu\right)=j',x_{\alpha,t}=x'\right)\pi_{\alpha,t}^k\left(u\mid x'\right)\cdot P\left(x\mid x',u,\frac{j'}{k}\right)$$

$$\cdot \frac{\sum_{(y_1,\ldots y_k)\in\mathcal{X}_j^k}\prod_{i=1}^{k}\frac{W(\alpha,\beta_i)}{\xi_W(\alpha)}\hat{\mu}_{\beta_i,t}^\nu(y_i)}{\int_{[0,\alpha^*]^k}\sum_{(z_1,\ldots z_k)\in\mathcal{X}_j^k}\prod_{i=1}^{k}\frac{W(\alpha,\beta_i')}{\xi_W(\alpha)}\hat{\mu}_{\beta_i',t}^\nu(z_i)\,\mathrm{d}\beta'}$$

$$(\mathrm{II}) = \binom{j'}{\ell}\left(\sum_{u\in\mathcal{U}}\pi_{\beta_i,t}^\infty\left(u\mid s_1\right)\cdot P\left(s_1\mid s_1,u,\mathbb{G}_{\beta_i,t}^\infty\left(\hat{\boldsymbol{\mu}}_t^\nu\right)\right)\right)^\ell$$

$$(\mathrm{III}) = \left(\sum_{u\in\mathcal{U}}\pi_{\beta_i,t}^\infty\left(u\mid s_2\right)\cdot P\left(s_1\mid s_2,u,\mathbb{G}_{\beta_i,t}^\infty\left(\hat{\boldsymbol{\mu}}_t^\nu\right)\right)\right)^{j-\ell}$$

$$(\mathrm{IV}) = \binom{k-j'}{j-\ell}\left(\sum_{u\in\mathcal{U}}\pi_{\beta_i,t}^\infty\left(u\mid s_1\right)\cdot P\left(s_2\mid s_1,u,\mathbb{G}_{\beta_i,t}^\infty\left(\hat{\boldsymbol{\mu}}_t^\nu\right)\right)\right)^{j'-\ell}$$

$$(\mathrm{V}) = \left(\sum_{u\in\mathcal{U}}\pi_{\beta_i,t}^\infty\left(u\mid s_2\right)\cdot P\left(s_2\mid s_2,u,\mathbb{G}_{\beta_i,t}^\infty\left(\hat{\boldsymbol{\mu}}_t^\nu\right)\right)\right)^{k-j'-j+\ell}$$

and $(\mathrm{I}'),(\mathrm{II}'),(\mathrm{III}'),(\mathrm{IV}')$, and $(\mathrm{V}')$ defined accordingly. As on the previous pages, we use a telescope sum to arrive at the bound

$$\sum_{j=0}^{k}\sum_{x\in\mathcal{X}}\mathbb{E}\left[\left|P_{\boldsymbol{\pi},\hat{\boldsymbol{\mu}}}\left(\mathbb{G}_{\alpha,t}^k\left(\hat{\boldsymbol{\mu}}_t^\nu\right)=j,\hat{x}_{\alpha,t}=x\right)-P_{\hat{\boldsymbol{\pi}},\hat{\boldsymbol{\mu}}}\left(\mathbb{G}_{\alpha,t}^k\left(\hat{\boldsymbol{\mu}}_t^\nu\right)=j,\hat{x}_{\alpha,t}=x\right)\right|\right]$$

$$\leq \sum_{j=0}^{k}\sum_{x\in\mathcal{X}}\int_{[0,\alpha^*]^k}\sum_{x'\in\mathcal{X}}\sum_{j'=0}^{k}\sum_{u\in\mathcal{U}}\sum_{\ell=\max\{0,j+j'-k\}}^{\min\{j,j'\}}\mathbb{E}\left[\left|(\mathrm{I})-(\mathrm{I}')\right|\right]\cdot O(1)+\mathbb{E}\left[\left|(\mathrm{II})-(\mathrm{II}')\right|\right]\cdot O(1)$$

$$+\mathbb{E}\left[\left|(\mathrm{III})-(\mathrm{III}')\right|\right]\cdot O(1)+\mathbb{E}\left[\left|(\mathrm{IV})-(\mathrm{IV}')\right|\right]\cdot O(1)+\mathbb{E}\left[\left|(\mathrm{V})-(\mathrm{V}')\right|\right]\cdot O(1)\,\mathrm{d}\beta_1\cdots\mathrm{d}\beta_k$$

where the first term goes to $0$ for $\nu\to\infty$ by induction, i.e.

$$\mathbb{E}\left[\left|(\mathrm{I})-(\mathrm{I}')\right|\right]\cdot O(1)=o(1)\,.$$

For the remaining four terms, we recall equation (6) which yields

$$\sum_{j=0}^{k}\sum_{x\in\mathcal{X}}\mathbb{E}\left[\left|P_{\boldsymbol{\pi},\hat{\boldsymbol{\mu}}}\left(\mathbb{G}_{\alpha,t}^k\left(\hat{\boldsymbol{\mu}}_t^\nu\right)=j,\hat{x}_{\alpha,t}=x\right)-P_{\hat{\boldsymbol{\pi}},\hat{\boldsymbol{\mu}}}\left(\mathbb{G}_{\alpha,t}^k\left(\hat{\boldsymbol{\mu}}_t^\nu\right)=j,\hat{x}_{\alpha,t}=x\right)\right|\right]$$

$$\leq o(1)+\sum_{j=0}^{k}\sum_{x\in\mathcal{X}}\int_{[0,\alpha^*]^k}\sum_{x'\in\mathcal{X}}\sum_{j'=0}^{k}\sum_{u\in\mathcal{U}}\sum_{\ell=\max\{0,j+j'-k\}}^{\min\{j,j'\}}O(1)$$

$$\cdot\left(\mathbb{E}\left[\sum_{u'\in\mathcal{U}}\left|\hat{\pi}_{\beta_i,t}\left(u'\mid s_1\right)-\pi_{\beta_i,t}^\infty\left(u'\mid s_1\right)\right|\right]+\mathbb{E}\left[\sum_{u'\in\mathcal{U}}\left|\hat{\pi}_{\beta_i,t}\left(u'\mid s_2\right)-\pi_{\beta_i,t}^\infty\left(u'\mid s_2\right)\right|\right]\right)$$

$$\mathrm{d}\beta_1\cdots\mathrm{d}\beta_k\,.$$

Eventually, this brings us to

$$
\frac{M_f \sqrt{2|E_\nu|}}{|V_{\nu,k}|} \cdot \sum_{i \in V_{\nu,k}} \int_{\frac{i-1}{\sqrt{2|E_\nu|}}}^{\frac{i}{\sqrt{2|E_\nu|}}} \sum_{j=0}^{k} \sum_{x \in \mathcal{X}} \sum_{u \in \mathcal{U}}
$$

$$
\cdot \mathbb{E}\left[\left|P_{\hat{\boldsymbol{\pi}},\hat{\boldsymbol{\mu}}}\left(\mathbb{G}_{\alpha,t}^k\left(\hat{\boldsymbol{\mu}}_t^\nu\right) = j, \hat{x}_{\alpha,t} = x\right) - P_{\boldsymbol{\pi},\hat{\boldsymbol{\mu}}}\left(\mathbb{G}_{\alpha,t}^k\left(\hat{\boldsymbol{\mu}}_t^\nu\right) = j, \hat{x}_{\alpha,t} = x\right)\right|\right] \, \mathrm{d}\alpha
$$

$$
\leq o(1) + \frac{M_f \sqrt{2|E_\nu|}}{|V_{\nu,k}|} \cdot \sum_{i \in V_{\nu,k}} \int_{\frac{i-1}{\sqrt{2|E_\nu|}}}^{\frac{i}{\sqrt{2|E_\nu|}}} \sum_{j=0}^{k} \sum_{x \in \mathcal{X}} \int_{[0,\alpha^*]^k} \sum_{x' \in \mathcal{X}} \sum_{j'=0}^{k} \sum_{u \in \mathcal{U}} \sum_{\ell=\max\{0,j+j'-k\}}^{\min\{j,j'\}} O(1)
$$

$$
\cdot \left( \mathbb{E}\left[\sum_{u' \in \mathcal{U}} \left|\hat{\pi}_{i,t}^\infty\left(u' \mid s_1\right) - \pi_{i,t}^\infty\left(u' \mid s_1\right)\right|\right] + \mathbb{E}\left[\sum_{u' \in \mathcal{U}} \left|\hat{\pi}_{i,t}^\infty\left(u' \mid s_2\right) - \pi_{i,t}^\infty\left(u' \mid s_2\right)\right|\right] \right)
$$

$$
\mathrm{d}\beta_1 \cdots \mathrm{d}\beta_k \mathrm{d}\alpha
$$

which is upper bounded by $\varepsilon$ for $\nu \to \infty$ by an argument as for the first term. This concludes this part of the proof.

**Fifth term.** We start with

$$
\mathbb{E}\left[\left|\hat{\boldsymbol{\mu}}_t^{\nu,k} P_{t,\hat{\boldsymbol{\mu}},W}^{\boldsymbol{\pi},k}(f) - \hat{\boldsymbol{\mu}}_t^{\nu,k} P_{t,\boldsymbol{\mu},W}^{\boldsymbol{\pi},k}(f)\right|\right]
$$

$$
= \mathbb{E}\left[\left|\frac{\sqrt{2|E_\nu|}}{|V_{\nu,k}|} \cdot \sum_{i \in V_{\nu,k}} \int_{\frac{i-1}{\sqrt{2|E_\nu|}}}^{\frac{i}{\sqrt{2|E_\nu|}}} \sum_{j=0}^{k} \sum_{x \in \mathcal{X}} P_{\boldsymbol{\pi},\hat{\boldsymbol{\mu}}}\left(\mathbb{G}_{\alpha,t}^k\left(\hat{\boldsymbol{\mu}}_t^\nu\right) = j, \hat{x}_{\alpha,t} = x\right)\right.\right.
$$

$$
\sum_{u \in \mathcal{U}} \pi_{\alpha,t}^k(u \mid x) \sum_{x' \in \mathcal{X}} P\left(x' \mid x, u, \frac{1}{k}(j, k - j)\right) f(x', \alpha) \, \mathrm{d}\alpha
$$

$$
- \frac{\sqrt{2|E_\nu|}}{|V_{\nu,k}|} \cdot \sum_{i \in V_{\nu,k}} \int_{\frac{i-1}{\sqrt{2|E_\nu|}}}^{\frac{i}{\sqrt{2|E_\nu|}}} \sum_{j=0}^{k} \sum_{x \in \mathcal{X}} P_{\boldsymbol{\pi},\hat{\boldsymbol{\mu}}}\left(\mathbb{G}_{\alpha,t}^k\left(\boldsymbol{\mu}_t^\nu\right) = j, \hat{x}_{\alpha,t} = x\right)
$$

$$
\left.\left.\sum_{u \in \mathcal{U}} \pi_{\alpha,t}^k(u \mid x) \sum_{x' \in \mathcal{X}} P\left(x' \mid x, u, \frac{1}{k}(j, k - j)\right) f(x', \alpha) \, \mathrm{d}\alpha\right|\right]
$$

$$
\leq \frac{M_f \sqrt{2|E_\nu|}}{|V_{\nu,k}|} \cdot \sum_{i \in V_{\nu,k}} \int_{\frac{i-1}{\sqrt{2|E_\nu|}}}^{\frac{i}{\sqrt{2|E_\nu|}}} \mathbb{E}\left[\sum_{j=0}^{k} \sum_{x \in \mathcal{X}} \left|P_{\boldsymbol{\pi},\hat{\boldsymbol{\mu}}}\left(\mathbb{G}_{\alpha,t}^k\left(\hat{\boldsymbol{\mu}}_t^\nu\right) = j, \hat{x}_{\alpha,t} = x\right)\right.\right.
$$

$$
\left.\left.- P_{\boldsymbol{\pi},\hat{\boldsymbol{\mu}}}\left(\mathbb{G}_{\alpha,t}^k\left(\boldsymbol{\mu}_t^\nu\right) = j, \hat{x}_{\alpha,t} = x\right)\right|\right] \, \mathrm{d}\alpha.
$$

Next, we reformulate

$$
\mathbb{E}\left[\sum_{j=0}^{k} \sum_{x \in \mathcal{X}} \left|P_{\boldsymbol{\pi},\hat{\boldsymbol{\mu}}}\left(\mathbb{G}_{\alpha,t}^k\left(\hat{\boldsymbol{\mu}}_t^\nu\right) = j, \hat{x}_{\alpha,t} = x\right) - P_{\boldsymbol{\pi},\hat{\boldsymbol{\mu}}}\left(\mathbb{G}_{\alpha,t}^k\left(\boldsymbol{\mu}_t^\nu\right) = j, \hat{x}_{\alpha,t} = x\right)\right|\right]
$$

by applying a telescope sum trick as for the third term, namely we state

$$
P_{\boldsymbol{\pi},\hat{\boldsymbol{\mu}}}\left(\mathbb{G}_{\alpha,t}^k\left(\hat{\boldsymbol{\mu}}_t^\nu\right) = j, \hat{x}_{\alpha,t} = x\right)
$$

$$
= \int_{[0,\alpha^*]^k} \sum_{x' \in \mathcal{X}} \sum_{j'=0}^{k} \sum_{u \in \mathcal{U}} \sum_{\ell=\max\{0,j+j'-k\}}^{\min\{j,j'\}} (\mathrm{I}) \cdot (\mathrm{II}) \cdot (\mathrm{III}) \cdot (\mathrm{IV}) \cdot (\mathrm{V}) \, \mathrm{d}\beta_1 \cdots \mathrm{d}\beta_k
$$

and also

$$
P_{\boldsymbol{\pi},\hat{\boldsymbol{\mu}}}\left(\mathbb{G}_{\alpha,t}^k\left(\boldsymbol{\mu}_t^\nu\right) = j, \hat{x}_{\alpha,t} = x\right)
$$

$$
= \int_{[0,\alpha^*]^k} \sum_{x' \in \mathcal{X}} \sum_{j'=0}^{k} \sum_{u \in \mathcal{U}} \sum_{\ell=\max\{0,j+j'-k\}}^{\min\{j,j'\}} (\mathrm{I}') \cdot (\mathrm{II}') \cdot (\mathrm{III}') \cdot (\mathrm{IV}') \cdot (\mathrm{V}') \, \mathrm{d}\beta_1 \cdots \mathrm{d}\beta_k
$$

where we define

$$\text{(I)} = P_{\boldsymbol{\pi},\hat{\boldsymbol{\mu}}}\left(\mathbb{G}^k_{\alpha,t}\left(\hat{\boldsymbol{\mu}}^\nu_t\right) = j', x_{\alpha,t} = x'\right)\pi^k_{\alpha,t}\left(u \mid x'\right)\cdot P\left(x \mid x', u, \frac{j'}{k}\right)$$

$$\cdot \frac{\sum_{(y_1,\ldots y_k)\in\mathcal{X}^k_j}\prod_{i=1}^k \frac{W(\alpha,\beta_i)}{\xi_W(\alpha)}\hat{\mu}^\nu_{\beta_i,t}(y_i)}{\int_{[0,\alpha^*]^k}\sum_{(z_1,\ldots z_k)\in\mathcal{X}^k_j}\prod_{i=1}^k \frac{W(\alpha,\beta'_i)}{\xi_W(\alpha)}\hat{\mu}^\nu_{\beta'_i,t}(z_i)\,\mathrm{d}\beta'}$$

$$\text{(II)} = \binom{j'}{\ell}\left(\sum_{u\in\mathcal{U}}\pi^\infty_{\beta_i,t}\left(u \mid s_1\right)\cdot P\left(s_1 \mid s_1, u, \mathbb{G}^\infty_{\beta_i,t}\left(\hat{\boldsymbol{\mu}}^\nu_t\right)\right)\right)^\ell$$

$$\text{(III)} = \left(\sum_{u\in\mathcal{U}}\pi^\infty_{\beta_i,t}\left(u \mid s_2\right)\cdot P\left(s_1 \mid s_2, u, \mathbb{G}^\infty_{\beta_i,t}\left(\hat{\boldsymbol{\mu}}^\nu_t\right)\right)\right)^{j-\ell}$$

$$\text{(IV)} = \binom{k-j'}{j-\ell}\left(\sum_{u\in\mathcal{U}}\pi^\infty_{\beta_i,t}\left(u \mid s_1\right)\cdot P\left(s_2 \mid s_1, u, \mathbb{G}^\infty_{\beta_i,t}\left(\hat{\boldsymbol{\mu}}^\nu_t\right)\right)\right)^{j'-\ell}$$

$$\text{(V)} = \left(\sum_{u\in\mathcal{U}}\pi^\infty_{\beta_i,t}\left(u \mid s_2\right)\cdot P\left(s_2 \mid s_2, u, \mathbb{G}^\infty_{\beta_i,t}\left(\hat{\boldsymbol{\mu}}^\nu_t\right)\right)\right)^{k-j'-j+\ell}$$

and $(\text{I}'), (\text{II}'), (\text{III}'), (\text{IV}'), (\text{V}')$ are defined correspondingly. Thus, we obtain the useful upper bound

$$\mathbb{E}\left[\sum_{j=0}^k\sum_{x\in\mathcal{X}}\left|P_{\boldsymbol{\pi},\hat{\boldsymbol{\mu}}}\left(\mathbb{G}^k_{\alpha,t}\left(\hat{\boldsymbol{\mu}}^\nu_t\right) = j, \hat{x}_{\alpha,t} = x\right) - P_{\boldsymbol{\pi},\hat{\boldsymbol{\mu}}}\left(\mathbb{G}^k_{\alpha,t}\left(\boldsymbol{\mu}^\nu_t\right) = j, \hat{x}_{\alpha,t} = x\right)\right|\right]$$

$$\leq \sum_{j=0}^k\sum_{x\in\mathcal{X}}\int_{[0,\alpha^*]^k}\sum_{x'\in\mathcal{X}}\sum_{j'=0}^k\sum_{u\in\mathcal{U}}\sum_{\ell=\max\{0,j+j'-k\}}^{\min\{j,j'\}}\mathbb{E}\left[\left|(\text{I})-(\text{I}')\right|\right]\cdot O(1) + \mathbb{E}\left[\left|(\text{II})-(\text{II}')\right|\right]\cdot O(1)$$

$$+\,\mathbb{E}\left[\left|(\text{III})-(\text{III}')\right|\right]\cdot O(1) + \mathbb{E}\left[\left|(\text{IV})-(\text{IV}')\right|\right]\cdot O(1) + \mathbb{E}\left[\left|(\text{V})-(\text{V}')\right|\right]\cdot O(1)\mathrm{d}\beta_1\cdots\mathrm{d}\beta_k$$

Here,

$$\mathbb{E}\left[\left|(\text{I})-(\text{I}')\right|\right]\cdot O(1)$$

converges to $0$ by induction and an argument as in the proof of inequality (2). Next, we focus on the term

$$\mathbb{E}\left[\left|(\text{II})-(\text{II}')\right|\right]\cdot O(1)$$

$$= \mathbb{E}\left[\left|\binom{j'}{\ell}\left(\sum_{u\in\mathcal{U}}\pi^\infty_{\beta_i,t}\left(u \mid s_1\right)\cdot P\left(s_1 \mid s_1, u, \mathbb{G}^\infty_{\beta_i,t}\left(\hat{\boldsymbol{\mu}}^\nu_t\right)\right)\right)^\ell\right.\right.$$

$$\left.\left.-\binom{j'}{\ell}\left(\sum_{u\in\mathcal{U}}\pi^\infty_{\beta_i,t}\left(u \mid s_1\right)\cdot P\left(s_1 \mid s_1, u, \mathbb{G}^\infty_{\beta_i,t}\left(\boldsymbol{\mu}^\nu_t\right)\right)\right)^\ell\right|\right]\cdot O(1)$$

where we once again use equation (6) to obtain

$$\mathbb{E}\left[\left|(\text{II})-(\text{II}')\right|\right]\cdot O(1)$$

$$= \mathbb{E}\left[\left|\sum_{u\in\mathcal{U}}\pi^\infty_{\beta_i,t}\left(u \mid s_1\right)\cdot P\left(s_1 \mid s_1, u, \mathbb{G}^\infty_{\beta_i,t}\left(\hat{\boldsymbol{\mu}}^\nu_t\right)\right)\right.\right.$$

$$\left.\left.-\sum_{u\in\mathcal{U}}\pi^\infty_{\beta_i,t}\left(u \mid s_1\right)\cdot P\left(s_1 \mid s_1, u, \mathbb{G}^\infty_{\beta_i,t}\left(\boldsymbol{\mu}^\nu_t\right)\right)\right|\right]\cdot O(1)$$

$$\leq \sum_{u\in\mathcal{U}}\pi^\infty_{\beta_i,t}\left(u \mid s_1\right)\cdot\mathbb{E}\left[\left|P\left(s_1 \mid s_1, u, \mathbb{G}^\infty_{\beta_i,t}\left(\hat{\boldsymbol{\mu}}^\nu_t\right)\right) - P\left(s_1 \mid s_1, u, \mathbb{G}^\infty_{\beta_i,t}\left(\boldsymbol{\mu}^\nu_t\right)\right)\right|\right]\cdot O(1)$$

$$\leq \sum_{u \in \mathcal{U}} \pi^{\infty}_{\beta_i,t}(u \mid s_1) \cdot \max_{x \in \mathcal{X}} \mathbb{E}\left[\left\|\frac{1}{\xi_W(\alpha)} \int_{\mathbb{R}_+} W(\alpha,\beta)\hat{\mu}_{\beta,t}\,\mathrm{d}\beta - \frac{1}{\xi_W(\alpha)}\int_{\mathbb{R}_+} W(\alpha,\beta)\mu_{\beta,t}\,\mathrm{d}\beta\right\|\right]$$
$$\cdot\, O(1) + \varepsilon_{\alpha^*}$$
$$= o(1) + \varepsilon_{\alpha^*}$$

where the convergence of the first term for $\nu \to \infty$ follows from Theorem 1. Finally, the terms

$$\mathbb{E}\left[|(\mathrm{III}) - (\mathrm{III}')|\right], \qquad \mathbb{E}\left[|(\mathrm{IV}) - (\mathrm{IV}')|\right], \qquad \text{and } \mathbb{E}\left[|(\mathrm{V}) - (\mathrm{V}')|\right]$$

can be upper bounded in a similar fashion which completes this part of the proof.

**Sixth term.** We have

$$\mathbb{E}\left[\left\|\hat{\boldsymbol{\mu}}_t^{\nu,k} P_{t,\boldsymbol{\mu},W}^{\boldsymbol{\pi},k}(f) - \boldsymbol{\mu}_{t+1}^k(f,\nu)\right\|\right]$$
$$= \mathbb{E}\left[\left\|\hat{\boldsymbol{\mu}}_t^{\nu,k} P_{t,\boldsymbol{\mu},W}^{\boldsymbol{\pi},k}(f) - \boldsymbol{\mu}_t^k P_{t,\boldsymbol{\mu},W}^{\boldsymbol{\pi},k}(f,\nu)\right\|\right]$$
$$= \mathbb{E}\left[\left\|\frac{\sqrt{2|E_\nu|}}{|V_{\nu,k}|}\sum_{i \in V_\nu} \mathbf{1}_{\{\deg(v_i)=k\}} \int_{\frac{i-1}{\sqrt{2|E_\nu|}}}^{\frac{i}{\sqrt{2|E_\nu|}}} \sum_{x \in \mathcal{X}} \hat{\mu}_{\alpha,t}^{\nu}(x) \sum_{j=0}^{k} P_{\boldsymbol{\pi}}\left(\mathbb{G}_{\alpha,t}^k(\boldsymbol{\mu}_t^\nu) = j \mid x_{\alpha,t} = x\right)\right.\right.$$
$$\sum_{u \in \mathcal{U}} \pi_{\alpha,t}^k(u \mid x) \sum_{x' \in \mathcal{X}} P\left(x' \mid x, u, \frac{1}{k}(j, k-j)\right) f(x',\alpha)\,\mathrm{d}\alpha$$
$$- \frac{\sqrt{\bar{\xi}_W}}{\int_0^\infty \mathrm{Poi}_{\nu,\alpha}^W(k)\,\mathrm{d}\alpha} \cdot \int_0^\infty \mathrm{Poi}_{\nu,\alpha}^W(k) \sum_{x \in \mathcal{X}} \mu_{\alpha,t}^k(x) \sum_{j=0}^{k} P_{\boldsymbol{\pi}}\left(\mathbb{G}_{\alpha,t}^k(\boldsymbol{\mu}_t^\nu) = j \mid x_{\alpha,t} = x\right)$$
$$\left.\left.\sum_{u \in \mathcal{U}} \pi_{\alpha,t}^k(u \mid x) \sum_{x' \in \mathcal{X}} P\left(x' \mid x, u, \frac{1}{k}(j, k-j)\right) f(x',\alpha)\,\mathrm{d}\alpha\right\|\right]$$
$$= \mathbb{E}\left[\left\|\frac{\sqrt{2|E_\nu|}}{|V_{\nu,k}|} \cdot \sum_{i \in V_\nu} \mathbf{1}_{\{\deg(v_i)=k\}} \int_{\frac{i-1}{\sqrt{2|E_\nu|}}}^{\frac{i}{\sqrt{2|E_\nu|}}} \sum_{x \in \mathcal{X}} \hat{\mu}_{\alpha,t}^{\nu}(x) f'(x,\alpha)\,\mathrm{d}\alpha\right.\right.$$
$$\left.\left.- \frac{\sqrt{\bar{\xi}_W}}{\int_0^\infty \mathrm{Poi}_{\nu,\alpha}^W(k)\,\mathrm{d}\alpha} \cdot \int_0^\infty \mathrm{Poi}_{\nu,\alpha}^W(k) \sum_{x \in \mathcal{X}} \mu_{\alpha,t}^k(x) f'(x,\alpha)\,\mathrm{d}\alpha\right\|\right]$$
$$= \mathbb{E}\left[\left\|\hat{\boldsymbol{\mu}}_t^{\nu,k}(f') - \boldsymbol{\mu}_t^k(f',\nu)\right\|\right] \leq \varepsilon$$

where we apply the induction assumption and define the functions

$$f'(x,\alpha) := \sum_{j=0}^{k} P_{\boldsymbol{\pi}}\left(\mathbb{G}_{\alpha,t}^k(\hat{\boldsymbol{\mu}}_t^\nu) = j \mid x_{\alpha,t} = x\right) \sum_{u \in \mathcal{U}} \pi_{\alpha,t}^k(u \mid x)$$
$$\sum_{x' \in \mathcal{X}} P\left(x' \mid x, u, \frac{1}{k}(j, k-j)\right) f(x',\alpha).$$

This concludes the proof.

## E    PROOF OF THEOREM 3

For notational convenience we define $\alpha(i) := \frac{i}{\sqrt{2|E_\nu|}}$. Furthermore, we define

$$V_{\nu,\alpha^*} := \left\{i \in V_\nu : \frac{i}{\sqrt{2|E_\nu|}} \leq \alpha^*\right\} \subset V_\nu.$$

The next lemma is crucial for proving Theorem 3.

**Lemma 3.** *Consider Lipschitz continuous $\boldsymbol{\pi} \in \boldsymbol{\Pi}$ up to a finite number of discontinuities $D_\pi$, with $\boldsymbol{\mu} = \Psi(\boldsymbol{\pi})$. Under Assumptions 1 and 2 and the policy $\Gamma_{|V_\nu|}(\boldsymbol{\pi}, \bar{\pi}_i) \in \Pi^{|V_\nu|}$, $\bar{\pi}_i \in \Pi$ arbitrary, for any uniformly bounded family of functions $\mathcal{G}$ from $\mathcal{X}$ to $\mathbb{R}$ and any $\varepsilon, p > 0$, $t \in \mathcal{T}$, there exist $\nu', \alpha' > 0$ such that for all $\nu > \nu'$ and $\alpha^* > \alpha'$*

$$\sup_{g \in \mathcal{G}} \left| \mathbb{E}\left[g(X_{i,t})\right] - \mathbb{E}\left[g(\bar{X}_{\alpha(i),t})\right] \right| < \varepsilon \tag{7}$$

*holds uniformly over $\bar{\pi}_i \in \Pi$, $i \in V_\nu'$ for some $V_\nu' \subseteq V_{\nu,\alpha^*}$ with $|V_\nu'| \geq (1-p)\,|V_{\nu,\alpha^*}|$. Furthermore, for any uniformly Lipschitz, uniformly bounded family of measurable functions $\mathcal{H}$ from $\mathcal{X} \times \mathcal{B}(\mathcal{X})$ to $\mathbb{R}$ and any $\varepsilon, p > 0$, $t \in \mathcal{T}$, there exist $\nu', \alpha' > 0$ such that for all $\nu > \nu'$ and $\alpha^* > \alpha'$*

$$\sup_{h \in \mathcal{H}} \left| \mathbb{E}\left[h(X_{i,t}, \mathbb{G}_{\alpha(i),t}^\nu(\hat{\boldsymbol{\mu}}_t^\nu))\right] - \mathbb{E}\left[h(\bar{X}_{\alpha(i),t}, \mathbb{G}_{\alpha(i),t}^\infty(\boldsymbol{\mu}_t))\right] \right| < \varepsilon \tag{8}$$

*holds uniformly over $\bar{\pi} \in \Pi$, $i \in V_\nu'$ for some $V_\nu' \subseteq V_{\nu,\alpha^*}$ with $|V_\nu'| \geq (1-p)\,|V_{\nu,\alpha^*}|$.*

*Proof.* Let us start by showing that (7) implies (8). Therefore, we upper bound the term of interest by

$$\left| \mathbb{E}\left[h(X_{i,t}, \mathbb{G}_{i,t}^\nu(\hat{\boldsymbol{\mu}}_t^\nu))\right] - \mathbb{E}\left[h(\bar{X}_{\alpha(i),t}, \mathbb{G}_{\alpha(i),t}^\infty(\boldsymbol{\mu}_t))\right] \right|$$

$$\leq \left| \mathbb{E}\left[h(X_{i,t}, \mathbb{G}_{i,t}^\nu(\hat{\boldsymbol{\mu}}_t^\nu))\right] - \mathbb{E}\left[h(X_{i,t}, \mathbb{G}_{i,t}^\nu(\boldsymbol{\mu}_t))\right] \right|$$

$$+ \left| \mathbb{E}\left[h(X_{i,t}, \mathbb{G}_{i,t}^\nu(\boldsymbol{\mu}_t))\right] - \mathbb{E}\left[h(X_{i,t}, \mathbb{G}_{\alpha(i),t}^\infty(\boldsymbol{\mu}_t))\right] \right|$$

$$+ \left| \mathbb{E}\left[h(X_{i,t}, \mathbb{G}_{\alpha(i),t}^\infty(\boldsymbol{\mu}_t))\right] - \mathbb{E}\left[h(\bar{X}_{\alpha(i),t}, \mathbb{G}_{\alpha(i),t}^\infty(\boldsymbol{\mu}_t))\right] \right| .$$

**First term.** We start with

$$\left| \mathbb{E}\left[h(X_{i,t}, \mathbb{G}_{i,t}^\nu(\hat{\boldsymbol{\mu}}_t^\nu))\right] - \mathbb{E}\left[h(X_{i,t}, \mathbb{G}_{i,t}^\nu(\boldsymbol{\mu}_t))\right] \right|$$

$$\leq L_h \sum_{x \in \mathcal{X}} \left| \mathbb{G}_{i,t}^\nu(\hat{\boldsymbol{\mu}}_t^\nu)(x) - \mathbb{G}_{i,t}^\nu(\boldsymbol{\mu}_t)(x) \right|$$

$$= \frac{L_h}{\xi_{\widehat{W}}(\alpha(i))} \sum_{x \in \mathcal{X}} \left| \int_0^\infty \widehat{W}(\alpha(i), \beta) \hat{\mu}_{\beta,t}^\nu(x)\, \mathrm{d}\beta - \int_0^\infty \widehat{W}(\alpha(i), \beta) \mu_{\beta,t}(x)\, \mathrm{d}\beta \right|$$

$$= \varepsilon_{\alpha^*} + \frac{L_h}{\xi_{\widehat{W}}(\alpha(i))} \sum_{x \in \mathcal{X}} \left| \int_0^{\alpha^*} \widehat{W}(\alpha(i), \beta) \hat{\mu}_{\beta,t}^\nu(x)\, \mathrm{d}\beta - \int_0^{\alpha^*} \widehat{W}(\alpha(i), \beta) \mu_{\beta,t}(x)\, \mathrm{d}\beta \right|$$

$$= \varepsilon_{\alpha^*} + o(1)$$

where the second summand goes to zero for $\nu \to \infty$ by Theorem 1 and by choosing $\alpha^*$ sufficiently high, the sum can be bounded by $\varepsilon$.

**Second term.** For this term we have

$$\left| \mathbb{E}\left[h(X_{i,t}, \mathbb{G}_{i,t}^\nu(\boldsymbol{\mu}_t))\right] - \mathbb{E}\left[h(X_{i,t}, \mathbb{G}_{\alpha(i),t}^\infty(\boldsymbol{\mu}_t))\right] \right|$$

$$\leq L_h \sum_{x \in \mathcal{X}} \left| \mathbb{G}_{i,t}^\nu(\boldsymbol{\mu}_t)(x) - \mathbb{G}_{\alpha(i),t}^\infty(\boldsymbol{\mu}_t)(x) \right|$$

$$= L_h \sum_{x \in \mathcal{X}} \left| \frac{1}{\xi_{\widehat{W}}(\alpha(i))} \int_0^\infty \widehat{W}(\alpha(i), \beta) \mu_{\beta,t}(x)\, \mathrm{d}\beta \right.$$

$$\left. - \frac{1}{\xi_{W,\alpha^*}(\alpha(i))} \int_0^{\alpha^*} W(\alpha(i), \beta) \mu_{\beta,t}(x)\, \mathrm{d}\beta \right|$$

$$\leq \varepsilon_{\alpha^*} + L_h \sum_{x \in \mathcal{X}} \left| \left( \frac{1}{\xi_{\widehat{W}}(\alpha(i))} - \frac{1}{\xi_{W,\alpha^*}(\alpha(i))} \right) \int_0^\infty \widehat{W}(\alpha(i), \beta) \mu_{\beta,t}(x)\, \mathrm{d}\beta \right|$$

$$+ \frac{L_h}{\xi_{W,\alpha^*}(\alpha(i))} \sum_{x \in \mathcal{X}} \left| \int_0^{\alpha^*} \widehat{W}(\alpha(i), \beta) \mu_{\beta,t}(x)\, \mathrm{d}\beta - \int_0^{\alpha^*} W(\alpha(i), \beta) \mu_{\beta,t}(x)\, \mathrm{d}\beta \right|$$

$$= \varepsilon_{\alpha^*} + o(1)$$

where the second and third summand converge to zero for $\nu \to \infty$ by Assumption 1. Thus, for sufficiently high $\alpha^*$, the second term is upper bounded by $\varepsilon$.

**Third term.** Finally, the third term is bounded by (7), i.e.

$$\left| \mathbb{E}\left[ h(X_{i,t}, \mathbb{G}^\infty_{\alpha(i),t}(\boldsymbol{\mu}_t)) \right] - \mathbb{E}\left[ h(\bar{X}_{\alpha(i),t}, \mathbb{G}^\infty_{\alpha(i),t}(\boldsymbol{\mu}_t)) \right] \right| < \varepsilon \,.$$

This concludes the first part of the proof. Now it remains to establish (7) via induction over $t$. At $t = 0$, the induction assumption holds by the construction of the model. Thus, it remains to show the induction step

$$\sup_{g \in \mathcal{G}} \left| \mathbb{E}\left[ g(X_{i,t+1}) \right] - \mathbb{E}\left[ g(\bar{X}_{\alpha(i),t+1}) \right] \right| < \varepsilon \,.$$

We define

$$h'_{\nu,t}(x, G) := \sum_{u \in \mathcal{U}} \hat{\pi}_t(u|x) \sum_{x' \in \mathcal{X}} P(x'|x, u, G) g(x')$$

which allows us to apply the induction assumption and thereby (8) (by the implication we have proved above) to our term of interest, i.e.

$$\sup_{g \in \mathcal{G}} \left| \mathbb{E}\left[ g(X_{i,t+1}) \right] - \mathbb{E}\left[ g(\bar{X}_{\alpha(i),t+1}) \right] \right|$$

$$= \left| \mathbb{E}\left[ h'_{\nu,t}(X_{i,t}, \mathbb{G}^\nu_{i,t}(\hat{\boldsymbol{\mu}}^\nu_t)) \right] - \mathbb{E}\left[ h'_{\nu,t}(\bar{X}_{\alpha(i),t}, \mathbb{G}^\infty_{\alpha(i),t}(\boldsymbol{\mu}_t)) \right] \right| < \varepsilon$$

and concludes the proof. $\qquad\square$

*Proof of Theorem 3.* The proof leverages Lemma 3. More precisely, for the situation as in Lemma 3 we have

$$\sup_{\bar{\pi}_i \in \Pi} \left| J^\nu_i(\pi_1, \dots, \pi_{|V_\nu|}) - J^\nu_i(\pi_1, \dots, \bar{\pi}_i, \dots, \pi_{|V_\nu|}) \right|$$

$$\leq \left| J^\nu_i(\pi_1, \dots, \pi_{|V_\nu|}) - J^{\boldsymbol{\mu}}_{\alpha(i)}(\pi_{\alpha(i)}) \right| + \underbrace{\sup_{\bar{\pi}_i \in \Pi} \left| J^{\boldsymbol{\mu}}_{\alpha(i)}(\pi_{\alpha(i)}) - J^{\boldsymbol{\mu}}_{\alpha(i)}(\bar{\pi}_i) \right|}_{=0}$$

$$+ \sup_{\bar{\pi}_i \in \Pi} \left| J^{\boldsymbol{\mu}}_{\alpha(i)}(\bar{\pi}_i) - J^\nu_i(\pi_1, \dots, \bar{\pi}_i, \dots, \pi_{|V_\nu|}) \right|$$

$$\leq \sum_{t=0}^{T-1} \left| \mathbb{E}\left[ r_{\pi_{\alpha(i)}}(X_{i,t}, \mathbb{G}^\nu_{i,t}(\hat{\boldsymbol{\mu}}^\nu_t)) \right] - \mathbb{E}\left[ r_{\pi_{\alpha(i)}}(\bar{X}_{\alpha(i),t}, \mathbb{G}^\infty_{\alpha(i),t}(\boldsymbol{\mu}_t)) \right] \right|$$

$$+ \sup_{\bar{\pi}_i \in \Pi} \sum_{t=0}^{T-1} \left| \mathbb{E}\left[ r_{\bar{\pi}_i}(X_{i,t}, \mathbb{G}^\nu_{i,t}(\hat{\boldsymbol{\mu}}^\nu_t)) \right] - \mathbb{E}\left[ r_{\bar{\pi}_i}(\bar{X}_{\alpha(i),t}, \mathbb{G}^\infty_{\alpha(i),t}(\boldsymbol{\mu}_t)) \right] \right| \leq \varepsilon$$

where the last inequality follows from Lemma 3 by defining

$$r_\pi(x, G) := \sum_{u \in \mathcal{U}} \pi(x|u) r(x, u, G)$$

for all $\pi \in \Pi$. This concludes the proof. $\qquad\square$

## F  PROOF OF THEOREM 4

The proof is similar to the proof of Theorem 3. As before, we start with a helpful lemma and define $\alpha(i) := \frac{i}{\sqrt{2|E_\nu|}}$.

**Lemma 4.** *Consider Lipschitz continuous $\boldsymbol{\pi} \in \boldsymbol{\Pi}$ up to a finite number of discontinuities $D_\pi$, with $\boldsymbol{\mu} = \Psi(\boldsymbol{\pi})$. Under Assumptions 1 and 2 and the policy $(\pi^1, \dots, \pi^{|V_\nu|}) = \Gamma_{|V_\nu|}(\boldsymbol{\pi}, \bar{\pi}_i) \in \Pi^{|V_\nu|}$, for any uniformly bounded family of functions $\mathcal{G}$ from $\mathcal{X}$ to $\mathbb{R}$ and any $\varepsilon, p > 0$, $t \in \mathcal{T}$, there exist $\nu', \alpha' > 0$ such that for all $\nu > \nu'$ and $\alpha^* > \alpha'$*

$$\sup_{g \in \mathcal{G}} \left| \mathbb{E}\left[ g(X_{i,t}) \right] - \mathbb{E}\left[ g(\bar{X}_{\alpha(i),t}) \right] \right| < \varepsilon \tag{9}$$

*holds uniformly over $\bar{\pi}_i \in \Pi, i \in V'_\nu$ for some $V'_\nu \subseteq V_\nu$ with $|V'_\nu| \geq (1-p)|V_\nu|$. Furthermore, for any uniformly Lipschitz, uniformly bounded family of measurable functions $\mathcal{H}$ from $\mathcal{X} \times \mathcal{B}(\mathcal{X})$ to $\mathbb{R}$ and any $\varepsilon, p > 0$, $t \in \mathcal{T}$, $k \in \mathbb{N}$, there exist $\nu', \alpha' > 0$ such that for all $\nu > \nu'$ and $\alpha^* > \alpha'$*

$$\sup_{h \in \mathcal{H}} \left| \mathbb{E}\left[h(X_{i,t}, \mathbb{G}^{\nu,k}_{\alpha(i),t}(\hat{\boldsymbol{\mu}}^\nu_t))\right] - \mathbb{E}\left[h(\bar{X}_{\alpha(i),t}, \mathbb{G}^k_{\alpha(i),t}(\boldsymbol{\mu}_t))\right] \right| < \varepsilon \tag{10}$$

*holds uniformly over $\bar{\pi}_i \in \Pi, i \in V'_\nu$ for some $V'_\nu \subseteq V_\nu$ with $|V'_\nu| \geq (1-p)|V_\nu|$.*

*Proof.* To show that (9) implies (10), we use the upper bound

$$\left| \mathbb{E}\left[h(X_{i,t}, \mathbb{G}^{\nu,k}_{i,t}(\hat{\boldsymbol{\mu}}^\nu_t))\right] - \mathbb{E}\left[h(\bar{X}_{\alpha(i),t}, \mathbb{G}^k_{\alpha(i),t}(\boldsymbol{\mu}_t))\right] \right|$$
$$\leq \left| \mathbb{E}\left[h(X_{i,t}, \mathbb{G}^{\nu,k}_{i,t}(\hat{\boldsymbol{\mu}}^\nu_t))\right] - \mathbb{E}\left[h(X_{i,t}, \mathbb{G}^{\nu,k}_{i,t}(\boldsymbol{\mu}_t))\right] \right|$$
$$+ \left| \mathbb{E}\left[h(X_{i,t}, \mathbb{G}^{\nu,k}_{i,t}(\boldsymbol{\mu}_t))\right] - \mathbb{E}\left[h(X_{i,t}, \mathbb{G}^k_{\alpha(i),t}(\boldsymbol{\mu}_t))\right] \right|$$
$$+ \left| \mathbb{E}\left[h(X_{i,t}, \mathbb{G}^k_{\alpha(i),t}(\boldsymbol{\mu}_t))\right] - \mathbb{E}\left[h(\bar{X}_{\alpha(i),t}, \mathbb{G}^k_{\alpha(i),t}(\boldsymbol{\mu}_t))\right] \right| .$$

**First term.** We start with

$$\left| \mathbb{E}\left[h(X_{i,t}, \mathbb{G}^{\nu,k}_{i,t}(\hat{\boldsymbol{\mu}}^\nu_t))\right] - \mathbb{E}\left[h(X_{i,t}, \mathbb{G}^{\nu,k}_{i,t}(\boldsymbol{\mu}_t))\right] \right|$$
$$\leq L_h \sum_{x \in \mathcal{X}} \left| \mathbb{G}^{\nu,k}_{i,t}(\hat{\boldsymbol{\mu}}^\nu_t)(x) - \mathbb{G}^{\nu,k}_{i,t}(\boldsymbol{\mu}_t)(x) \right|$$
$$= \frac{L_h}{\xi_{\widehat{W}}(\alpha(i))} \sum_{x \in \mathcal{X}} \left| \int_0^\infty \widehat{W}(\alpha(i), \beta)\hat{\mu}^\nu_{\beta,t}(x)\,\mathrm{d}\beta - \int_0^\infty \widehat{W}(\alpha(i), \beta)\mu_{\beta,t}(x)\,\mathrm{d}\beta \right|$$
$$= \varepsilon_{\alpha^*} + \frac{L_h}{\xi_{\widehat{W}}(\alpha(i))} \sum_{x \in \mathcal{X}} \left| \int_0^{\alpha^*} \widehat{W}(\alpha(i), \beta)\hat{\mu}^\nu_{\beta,t}(x)\,\mathrm{d}\beta - \int_0^{\alpha^*} \widehat{W}(\alpha(i), \beta)\mu_{\beta,t}(x)\,\mathrm{d}\beta \right|$$
$$= \varepsilon_{\alpha^*} + o(1)$$

where the first term goes to zero for $\nu \to \infty$ by Theorem 1 and by choosing $\alpha^*$ sufficiently high, the sum can be bounded by $\varepsilon$.

**Second term.** Moving on to the second term, we know that

$$\left| \mathbb{E}\left[h(X_{i,t}, \mathbb{G}^{\nu,k}_{i,t}(\boldsymbol{\mu}_t))\right] - \mathbb{E}\left[h(X_{i,t}, \mathbb{G}^k_{\alpha(i),t}(\boldsymbol{\mu}_t))\right] \right|$$
$$\leq L_h \sum_{x \in \mathcal{X}} \left| \mathbb{E}\left[\mathbb{G}^{\nu,k}_{i,t}(\boldsymbol{\mu}_t)(x)\right] - \mathbb{E}\left[\mathbb{G}^k_{\alpha(i),t}(\boldsymbol{\mu}_t)(x)\right] \right| = \varepsilon_{\alpha^*} + o(1)$$

where the last line follows from applying inequality (5). Therefore, by choosing a sufficiently high $\alpha^*$, the second term is smaller than $\varepsilon$.

**Third term.** Finally, the third term is bounded by (9) analogously to the proof of Theorem 3

$$\left| \mathbb{E}\left[h(X_{i,t}, \mathbb{G}^k_{\alpha(i),t}(\boldsymbol{\mu}_t))\right] - \mathbb{E}\left[h(\bar{X}_{\alpha(i),t}, \mathbb{G}^k_{\alpha(i),t}(\boldsymbol{\mu}_t))\right] \right| < \varepsilon .$$

This concludes the first part of the proof. Thus, we still have to prove (9) via induction over $t$. This can be done as in Lemma 3, so we point to the preceding proof for details. $\qquad\square$

*Proof of Theorem 4.* This proof is analogous to the one of Theorem 3. The only difference is that instead of leveraging Lemma 3, we now use Lemma 4. Then, the result follows immediately. $\qquad\square$

## G    EXPERIMENT DETAILS

In this section, we describe problem and algorithm details, and give additional periphery MF comparisons on the real network datasets that were omitted in the main text. Note that for separable power-law graphexes used in this work, the computations in Algorithm 1 simplify to calculating the same MF and neighborhood probabilities for all $\alpha$, improving the complexity of the algorithm in practice. For Algorithm 1, we use $\mathbb{G}_t^k \sim \text{Multinomial}(k, \boldsymbol{\mu}_t^\infty)$. The code is available in the supplementary material.

**Algorithm hyperparameters**    For the implementation of Algorithm 1, we use an inverse step size (temperature) of $\gamma = 50$ and perform 5000 iterations as seen in Figure 2. As degree cutoffs for the periphery, we use $k_{\max} = 8$ for SIS and SIR, and $k_{\max} = 6$ for RS, i.e. all nodes with degree $k > k_{\max}$ follow the MF policy.

**SIS problem.**    In the SIS problem, we assume a state space $\mathcal{X} = \{S, I\}$ consisting of the susceptible state $x = S$ and the infected state $x = I$. The time horizon $T$ is the time until a cure is found. Each agent may choose to avoid infections, taking actions $\mathcal{U} = \{\bar{P}, P\}$ to protect themselves ($P$) or not ($\bar{P}$). The resulting infection dynamics are then defined by the transition kernels

$$P^k(S \mid S, P, G) = 1, \quad P^k(I \mid S, \bar{P}, G) = \tau_I G(\mathbf{1}_I) \left( \frac{2}{1 + \exp(-\frac{k}{2})} - 1 \right), \quad \forall G \in \boldsymbol{\mathcal{G}}^k \quad (11)$$

for some infection rate $\tau_I > 0$, where the last term is a scaling factor that scales linearly with the number of neighbors around $k = 0$ (i.e. randomly meeting neighboring contacts) and eventually saturates to 1 as $k \to \infty$ (too many contacts to meet). The recovery dynamics are given by

$$P^k(S \mid I, u, G) = \tau_R, \quad \forall u \in \mathcal{U}, G \in \boldsymbol{\mathcal{G}}^k$$

for some recovery rate $\tau_R \in [0, 1]$. Finally, we assume costs for infection and protective actions

$$r(x, u, G) = c_I \mathbf{1}_I(x) + c_P \mathbf{1}_P(u)$$

for some coefficients $c_I, c_P > 0$.

In our experiments, we use $\tau_I = 0.2$, $\tau_R = 0.05$, $T = 500$, $\mu_0(I) = 0.5$, $c_I = 1$ and $c_P = 0.5$.

**SIR problem.**    In the SIR problem, in contrast to the SIS problem, we use the state space $\mathcal{X} = \{S, I, R\}$, i.e. we extend (11) by the new recovery dynamics

$$P^k(R \mid I, u, G) = 1 - P^k(I \mid I, u, G) = \tau_R, \quad \forall u \in \mathcal{U}, G \in \boldsymbol{\mathcal{G}}^k,$$

$$P^k(R \mid R, u, G) = 1, \quad \forall u \in \mathcal{U}, G \in \boldsymbol{\mathcal{G}}^k$$

and the same cost function as in SIS. For experiments, we use the parameters $\tau_I = 0.05$, $\tau_R = 0.01$, $T = 500$, $1 - \mu_0(S) = \mu_0(I) = 0.1$, $c_I = 1$ and $c_P = 0.25$.

**RS problem.**    Lastly, in the RS problem we have agents that are either aware ($A$) or unaware ($\bar{A}$) of the rumor. Agents that are aware may choose to propagate ($P$) the rumor to unaware agents or not ($\bar{P}$), resulting in actions $\mathcal{U} = \{\bar{P}, P\}$. The spreading probability is then dependent on the number of propagating agents in the neighborhood, which can be modelled by using the state space $\mathcal{X} = \{A, \bar{A}, \bar{P}, P\}$, where $\bar{P}$ and $P$ are aware agents trying to spread the rumor. As a result, the overall dynamics are given by

$$P^k(u \mid A, u, G) = 1, \quad P^k(A \mid u, u', G) = 1, \quad P^k(A \mid u, u', G) = 1, \quad \forall u, u' \in \mathcal{U}, G \in \boldsymbol{\mathcal{G}}^k$$

$$P^k(A \mid \bar{A}, u, G) = \tau_I G(\mathbf{1}_P) \left( \frac{2}{1 + \exp(-\frac{k}{2})} - 1 \right), \quad \forall u \in \mathcal{U}, G \in \boldsymbol{\mathcal{G}}^k.$$

Since the goal is to spread to neighbors that are unaware of the rumor, the rewards for propagation are given by positive rewards scaling with the number of unaware agents, and costs scaling with the number of aware agents.

$$r(x, u, G) = \big(c_P(G(\bar{P}) + G(P)) + r_P G(A)\big) \cdot \mathbf{1}_P(x)$$

We use $\tau_I = 0.3$, $T = 50$, $1 - \mu_0(S) = \mu_0(I) = 0.1$, $c_P = 0.8$ and $r_P = 0.5$.

Table 2: Expected time-averaged total variation $\Delta\mu^k = \frac{1}{2T}\mathbb{E}\left[\sum_t\|\hat{\mu}_t^k - \mu_t^k\|_1\right] \in [0,1]$ between the overall GXMFG MF prediction $\mu_t^k$ and the empirical MF $\hat{\mu}_t^k = \frac{1}{\sum_{i:\ \deg(v_i)=k}1}\sum_{i:\ \deg(v_i)=k}\delta_{X_t^i}$, exemplarily for $k=2$ and $k=5$ ($\pm$ standard deviation, 5 trials).

| network | Expected total variation $\Delta\mu^2$ in % | | | Expected total variation $\Delta\mu^5$ in % | | |
|---|---|---|---|---|---|---|
| | SIS (%) | SIR (%) | RS (%) | SIS (%) | SIR (%) | RS (%) |
| Prosper | $0.80 \pm 0.22$ | $0.86 \pm 0.43$ | $0.76 \pm 0.45$ | $1.01 \pm 0.04$ | $1.21 \pm 0.51$ | $1.03 \pm 0.74$ |
| Dogster | $1.36 \pm 0.25$ | $1.73 \pm 0.79$ | $1.02 \pm 0.53$ | $2.29 \pm 0.55$ | $3.02 \pm 0.89$ | $1.67 \pm 1.43$ |
| Pokec | $0.95 \pm 0.05$ | $2.39 \pm 0.03$ | $1.00 \pm 0.13$ | $1.56 \pm 0.09$ | $3.76 \pm 0.21$ | $1.45 \pm 0.13$ |
| Livemocha | $1.16 \pm 0.17$ | $2.08 \pm 0.90$ | $0.92 \pm 0.59$ | $1.03 \pm 0.13$ | $1.99 \pm 0.95$ | $1.02 \pm 0.67$ |
| Flickr | $4.26 \pm 0.11$ | $7.47 \pm 0.28$ | $2.16 \pm 0.10$ | $3.37 \pm 0.13$ | $12.03 \pm 0.47$ | $3.75 \pm 0.12$ |
| Brightkite | $1.23 \pm 0.16$ | $8.17 \pm 0.24$ | $2.46 \pm 0.34$ | $2.17 \pm 0.22$ | $11.19 \pm 0.96$ | $2.83 \pm 0.64$ |
| Facebook | $1.32 \pm 0.10$ | $10.20 \pm 0.67$ | $3.56 \pm 0.21$ | $3.00 \pm 0.27$ | $12.71 \pm 1.42$ | $4.06 \pm 0.72$ |
| Hyves | $5.99 \pm 0.35$ | $6.17 \pm 1.26$ | $1.65 \pm 0.56$ | $4.44 \pm 0.44$ | $9.03 \pm 1.47$ | $2.44 \pm 0.62$ |

**Additional comparisons on real data.** In Table 2, we also show the resulting mean field deviations for low $k$-degree nodes in the periphery. Similarly to the results in the main text in Table 1, we find that the mean field predicted by our GXMFG framework matches the empirical observations in the system on the real networks.

**Empirical comparison to the LPGMFG model** In this paragraph, we compare our GXMFG approach to the LPGMFG model proposed by Fabian et al. (2023). Note that LPGMFGs include the more restricted class of GMFGs (Cui & Koeppl, 2021b) which makes a separate comparison of our approach to GMFGs obsolete. The detailed results can be found in Table 3 where we repeat the GXMFG results from the main text for readability and easy comparison. Since the LPGMFG model does not allow for specific policies for agents with finite degree $k$, all agents follow the policy learned for agents with infinite degree. For implementation details on the LPGMFG model we refer to Fabian et al. (2023).

Table 3 shows that our GXMFG approach clearly outperforms LPGMFGs on all tasks and empirical networks. While the difference on some network-task combinations is relatively moderate, e.g. 4.28 (LPGMFG) vs. 2.78 (GXMFG) on Dogster SIS, the GXMFG performance on other networks and tasks, such as Flickr RS (15.80 vs. 2.24), SIS Brightkite (17.07 vs. 1.37), or SIR Hyves (39.94 vs. 10.06), is many times better then for the LPGMFG benchmark. These results strongly support the conceptual advantage of the hybrid graphex learning approach over existing work such as LPGMFGs. The GXMFGs ability to depict finite degree agents in the limiting model proves to be a crucial advantage over GMFGs and LPGMFGs that assume exclusively infinite degree agents. This is reflected by its superior performance on various tasks and real world networks as seen in Table 3.

Table 3: Average expected total variation $\Delta\mu = \frac{1}{2T} \mathbb{E}\left[\sum_t \|\hat{\mu}_t - \mu_t\|_1\right] \in [0, 1]$ between the overall GXMFG MF prediction $\mu_t$ and the empirical MF $\hat{\mu}_t = \sum_i \delta_{X_t^i}$ ($\pm$ standard deviation, 5 trials).

| network | $\Delta\mu$ in % for LPGMFG | | | $\Delta\mu$ in % for GXMFG | | |
|---|---|---|---|---|---|---|
| | SIS (%) | SIR (%) | RS (%) | SIS (%) | SIR (%) | RS (%) |
| Prosper | $2.66 \pm 0.22$ | $5.41 \pm 0.35$ | $2.80 \pm 0.43$ | $0.53 \pm 0.21$ | $2.06 \pm 0.25$ | $1.58 \pm 0.31$ |
| Dogster | $4.28 \pm 0.23$ | $8.83 \pm 1.16$ | $4.25 \pm 0.66$ | $2.78 \pm 0.44$ | $4.93 \pm 0.98$ | $1.89 \pm 0.66$ |
| Pokec | $6.26 \pm 0.03$ | $11.45 \pm 0.10$ | $5.67 \pm 0.11$ | $2.14 \pm 0.05$ | $4.93 \pm 0.12$ | $2.19 \pm 0.08$ |
| Livemocha | $4.46 \pm 0.10$ | $8.56 \pm 0.43$ | $4.40 \pm 0.83$ | $2.57 \pm 0.20$ | $4.40 \pm 1.08$ | $2.51 \pm 0.54$ |
| Flickr | $16.90 \pm 0.03$ | $35.49 \pm 0.05$ | $15.80 \pm 0.10$ | $3.57 \pm 0.08$ | $8.58 \pm 0.24$ | $2.24 \pm 0.11$ |
| Brightkite | $17.07 \pm 0.30$ | $32.27 \pm 0.37$ | $14.69 \pm 0.47$ | $1.37 \pm 0.11$ | $10.92 \pm 0.77$ | $3.37 \pm 0.25$ |
| Facebook | $16.22 \pm 0.16$ | $28.89 \pm 0.23$ | $13.03 \pm 0.07$ | $2.90 \pm 0.09$ | $13.57 \pm 0.37$ | $5.01 \pm 0.27$ |
| Hyves | $21.94 \pm 0.11$ | $39.94 \pm 0.50$ | $17.46 \pm 0.34$ | $5.07 \pm 0.24$ | $10.06 \pm 0.81$ | $2.62 \pm 0.44$ |

