# OpenReview forum: "Learning Mean Field Games on Sparse Graphs: A Hybrid Graphex Approach"
_ICLR.cc/2024/Conference — ICLR 2024 poster_

### Official Review · Reviewer_e4bh · 2023-11-01

**Soundness:** 3 good
**Presentation:** 3 good
**Contribution:** 3 good
**Rating:** 6
**Confidence:** 3

**Summary:**

This paper introduces Graphex Mean Field Games (GXMFGs) which build on the graph theoretical concept of graphexes to include sparse network structures between agents. This improves over prior work on Graphon Mean Field Games which only allows for modelling with dense graphs. The authors derive convergence properties for the finite game. In addition, a learning algorithm based on online mirror descent is provided for a particular class of GXMFGs that follow a core-periphery network structure. Finally, the theoretical claims are empirically validated over both synthetic and real-world networks.

**Strengths:**

- This paper has a clear motivation to extend Graphon Mean Field Games to deal with sparse graphs which are frequently seen in practice. The hybrid graphex approach proposed in this work looks like a natural and intuitive solution.
- The technical development is principled and the analysis is nontrivial.
- The overall presentation and clarity is good.

**Weaknesses:**

- Even though the authors explained in the paper, I didn't like the fact that the proposed GXMFGs have no baseline competitors to compare against. While I agree that one could argue on the contrary that the ability to work with sparse graphs is precisely the unique advantage of GXMGFs, I think that the authors should at least spend some efforts to discuss (if empirical comparison with LPGMFG is indeed unsuitable) how GXMFGs would compare with LPGMFG and GMFG in practice.

**Questions:**

In Figure 3a, it looks like the curves are diverging rather than converging as k increases? Are the curves coloured correctly?

---

> ### Author Response · Authors · 2023-11-17
> **Response to Reviewer e4bh**
>
> We thank the reviewer for the positive evaluation of our paper and the valuable suggestion to include an empirical comparison with existing LPGMFG and GMFG models. The feedback is answered in the same order that it was given in the review.
>
> ---
>
> *Even though the authors explained in the paper, I didn't like the fact that the proposed GXMFGs have no baseline competitors to compare against. While I agree that one could argue on the contrary that the ability to work with sparse graphs is precisely the unique advantage of GXMGFs, I think that the authors should at least spend some efforts to discuss (if empirical comparison with LPGMFG is indeed unsuitable) how GXMFGs would compare with LPGMFG and GMFG in practice.*
>
> A: Thank you for bringing up the important topic of an empirical comparison with existing approaches. As you mention, the ability of GXMFGs to work with sparse realistic graphs can be seen as a major conceptual advantage over existing approaches such as GMFGs and LPGMFGs. We agree that there should be an empirical comparison complementing the discussion on conceptual differences in the first version of the paper.
> Thus, we have added an empirical comparison of GXMFGs and LPGMFGs on the eight real world networks and three tasks from the first paper version. Due to space constraints, we mention this new comparison in the main text and provide detailed results in the appendix, see Table 3 and the corresponding discussion at the end of the Appendix of the updated paper. The empirical comparison does not include GMFGs because they can be seen as a subclass of LPGMFGs and therefore will not yield better results than the LPGMFG framework.
>
> Since LPGMFGs are not able to depict finite degree agents in the limiting model, all agents in the LPGMFG simulation follow the policy learned for infinite degree agents. The overall result of Table 3 is that our hybrid graphex learning approach clearly outperforms LPGMFGs across all networks and tasks. On some networks and tasks, the improvement is considerable but relatively moderate, for example on Prosper RS (error of 2.80 for LPGMFG vs. 1.58 in GXMFG). On other problems, our approach yields results that are many times better than those of the LPGMFG framework, such as Flickr SIS (16.90 vs. 3.57), Brightkite RS (14.69 vs. 3.37), and Hyves SIR (39.94 vs. 10.06). Thus, the empirical results provide strong evidence that the conceptual advantages of GXMFGs also yield a remarkably better empirical performance compared to previous methods such as LPGMFGs. For more details, please see the updated paper.
>
> ---
>
> *Q: In Figure 3a, it looks like the curves are diverging rather than converging as k increases? Are the curves coloured correctly?*
>
> A: The colors in Figure 3a are correct and all curves converge as the graph size $\nu$ increases. For higher $k$ the curves tend to converge slower than for low $k$ which might seem counter-intuitive. The reason for the different convergence speed is that if we sample finite graphs from the power law graphex, with high probability they will have far more nodes with degree $k=2$ than with degree $k=6$. The relatively low number of high degree nodes reflects the power law nature of the network, where many nodes have low degrees and relatively few nodes have high degrees. As a consequence, a larger graph size is required to obtain a sufficiently large, representative subset of nodes with $k=6$ that yields a good mean field estimate. In contrast, relatively small sampled graphs already have numerous vertices with $k=2$ such that the mean field convergence is observed earlier for this subset of nodes.

---

> > ### Comment · Reviewer_e4bh · 2023-11-22
> >
> > I thank the authors for their response and I appreciate the additional comparison with LPGMFGs. All my questions have been addressed. I increased my confidence. I will keep the score because I am not an expert in this particular field.

---

### Official Review · Reviewer_hgJx · 2023-11-01

**Soundness:** 3 good
**Presentation:** 3 good
**Contribution:** 3 good
**Rating:** 8
**Confidence:** 2

**Summary:**

This paper introduces Graphex Mean Field Games (GXMFGs), a framework for addressing the challenge of learning agent behavior in large populations. GXMFGs leverage graphon theory and graphexes, which represent limiting objects in sparse graph sequences. This approach suits real-world networks with both dense cores and sparse peripheries. The paper presents a specialized learning algorithm for GXMFGs.

Key contributions include:

1. Introduction of GXMFGs, extending the scope of Mean Field Games.
2. Provides theoretical guarantees to show that GXMFGs accurately approximates finite systems.
3. Development of a learning algorithm tailored to GXMFGs.
4. Empirical validation on synthetic and real-world networks, demonstrating GXMFGs' ability to model agent interactions and determine equilibria effectively.

**Strengths:**

- Well-Written and Organized: The paper demonstrates strong writing and organization, enhancing its overall readability and accessibility.

- Clear Motivation: The paper effectively conveys a clear and compelling motivation for addressing the problem it tackles.

- Thorough Discussion of Prior Works: The paper provides a comprehensive and well-structured overview of prior works related to the research area.

- The paper provides solid theoretical contributions complimented with supporting empirical studies strengthens the paper's arguments and findings.

**Weaknesses:**

As the current paper falls outside the scope of my research interests, I am unable to identify any significant weaknesses in the paper. Consequently, my confidence in assessing the paper is limited.

**Questions:**

- Providing an intuitive explanation for assumptions 1(b) and 1(c) would greatly enhance the paper's overall readability and accessibility.

- While the paper assumes finite state and action spaces, it may be beneficial to explore whether the proposed approach can be extended to scenarios with infinite action spaces.
- Including the code for the simulations, would enhance reproducibility.

---

> ### Author Response · Authors · 2023-11-17
> **Response to Reviewer hgJx**
>
> We thank the reviewer for the positive feedback and the suggestions for improving the accessibility of our work as well as outlining possible extensions of our current approach. The comments are answered in the order they were brought up in the review.
>
> ---
>
> *Q1: Providing an intuitive explanation for assumptions 1(b) and 1(c) would greatly enhance the paper's overall readability and accessibility.*
>
> A: Thank you for the valuable suggestion! To increase the accessibility of Assumptions 1 b) and c), we have added a more detailed explanation for the respective assumptions in the updated paper draft. The intuitive interpretation of Assumption 1 b) is that it describes the behavior of $\xi_W$ at infinity. More specifically, for $\sigma \in (0,1)$ Assumption 1 b) states that $\xi_W (\alpha)$ is approximately a power function $\alpha^{- \sigma}$ for large $\alpha$ which aligns with our goal to generate power law graphs. On the other hand, Assumption 1 c) is a technical assumption from the graph theory literature and has no obvious intuitive explanation to the best of our knowledge. Nevertheless, it is especially fulfilled by all separable graphexes which are characterized by the accessible property $W (\alpha, \beta) = \xi_W (\alpha) \xi_W (\beta) / \bar{\xi}_W $. Combining these two findings, one can see that Assumption 1 is especially satisfied by the separable power law graphex used in our paper. We hope that the added intuition and explanation increases both readability and accessibility.
>
> ---
>
> *Q2: While the paper assumes finite state and action spaces, it may be beneficial to explore whether the proposed approach can be extended to scenarios with infinite action spaces*
>
> A: In our opinion, it is worthwhile to extend the GXMFG approach to continuous state and action spaces (and also continuous time) to increase the generality of the learning method. Since the extension to a continuous setting will require different and adapted mathematical and algorithmic approaches, it is outside the scope of our paper. We have mentioned the promising research direction of defining a continuous version of GXMFGs in the conclusion of the updated draft. Thanks for the idea!
>
> ---
>
> *Q3: Including the code for the simulations, would enhance reproducibility.*
>
> A: We have uploaded the code and will add a link in the final, deanonymized version of the paper.

---

> > ### Comment · Reviewer_hgJx · 2023-11-23
> >
> > I appreciate the authors for addressing the concerns, and I am satisfied with the response. Thank you.

---

### Official Review · Reviewer_P6cQ · 2023-11-01

**Soundness:** 3 good
**Presentation:** 3 good
**Contribution:** 3 good
**Rating:** 6
**Confidence:** 4

**Summary:**

In this paper, the authors study a class of games with many players who are interacting through a sparse graph structure. More specifically, they are interested in the regime where the number of players tend to infinity. The main solution concept is an extension of the notion of Nash equilibrium. The authors propose a learning algorithm based on online mirror descent. They conclude the paper with examples and numerical simulations.

**Strengths:**

Overall, the paper studies an interesting problem and is relatively clearly written. As far as I know, this is a new extension of MFG to sparse graphs. The algorithm is very inspired from existing ones but there is an adaptation to the problem under consideration (core vs periphery).

**Weaknesses:**

The model is quite abstract at some places. For the theoretical results, they are mostly about the analysis of the game and I am not sure how relevant they are for this conference (although they are certainly interesting for a certain community). It might have been more interesting to focus more on the learning algorithm.

There are some typos which make it hard to check the correctness of some parts (see questions).

**Questions:**

1. I am wondering if some assumptions are missing. For example below Lemma 1, should $f$ be at least measurable (and perhaps more?) with respect to $\alpha$ for the integral to make sense?

2. Assumption 2 as used for instance in Lemma 1 does not seem to make much sense (unless I missed something): What is $\boldsymbol{\pi}$? We do not know in advance the equilibrium policy and even if we did, we would still need to define the set of admissible deviations for the Nash equilibrium. Could you please clarify?

3. Algorithm 1, line 14: Could you please explain or recall what is $Q^{k, \mu^{\tau_{\mathrm{max}}}}$?

Some typos: Should the state space be either $\mathcal{X}$ or $X$ (see section 3 for instance)? Does $\mathbb{G}^\infty_{\alpha,t}$ depend on $\boldsymbol{\mu}$ or not (see bottom of page 4)? Etc.

---

> ### Author Response · Authors · 2023-11-17
> **Response to Reviewer P6cQ**
>
> We thank the reviewer for the positive evaluation of our paper and the helpful comments on how to improve the paper. The feedback and questions are answered in the order they were brought up in the review.
>
> ---
>
> *The model is quite abstract at some places. For the theoretical results, they are mostly about the analysis of the game and I am not sure how relevant they are for this conference (although they are certainly interesting for a certain community). It might have been more interesting to focus more on the learning algorithm.*
>
> A: The analysis of the game provides the key insights into complex agent systems that are necessary to eventually provide the equilibrium learning algorithm. Only through a thorough understanding of the core periphery structure and its implications it is possible to state a principled equilibrium learning approach. Therefore, we do believe that the understanding of these complex agent networks and the resulting learning algorithm are relevant for this conference. Nevertheless, we agree that there are various open challenges and hope that our GXMFG learning approach provides a useful framework for future research.
>
> ---
>
> *Q1: I am wondering if some assumptions are missing. For example below Lemma 1, should $f$ be at least measurable (and perhaps more?) with respect to $\alpha$ for the integral to make sense?*
>
> A: We thank the reviewer for pointing out the potential issue with the assumptions on $f$. In the theoretical results such as Theorem 1 we state the assumptions on $f$, i.e. measurable and bounded. However, we did not mention these conditions on $f$ when defining the integral you referred to. To avoid the misleading impression that there are no assumptions on $f$, we also added them to the respective definition in the updated draft.
>
> ---
>
> *Q2: Assumption 2 as used for instance in Lemma 1 does not seem to make much sense (unless I missed something): What is $\boldsymbol \pi$? We do not know in advance the equilibrium policy and even if we did, we would still need to define the set of admissible deviations for the Nash equilibrium. Could you please clarify?*
>
> A: We completely agree with the reviewer: the policy $\boldsymbol \pi$ should not be part of Assumption 2 and (of course) we do not assume the equilibrium policy to be known in advance; the set of admissible deviation policies from the Nash equilibrium is not restricted. Instead, we have added the Lipschitz condition (up to a finite number of discontinuities) on $\boldsymbol \pi$ to the respective theoretical results, such as Theorems 1-4. Thank you for spotting the mistake in Assumption 2. We have corrected it in the updated paper version.
>
> ---
>
> *Q3: Algorithm 1, line 14: Could you please explain or recall what is $Q^{k, \mu^{\tau_{\max}}}$?*
>
> A: In Algorithm 1, $Q^{k, \mu^{\tau_{\max}}}$ is defined similar to $Q_{i,t}^{\pi, \mu}$, except that we substitute the reward function $r$ by $r'_k$ and use the transition kernel $P'_k$ instead of $P$. We have added the definition in the updated paper.
>
> ---
>
> *Some typos: Should the state space be either $\mathcal{X}$ or $X$ (see section 3 for instance)?*
>
> A: The state space should be $\mathcal{X}$. We used $X$ in the beginning of Section 3 to denote an arbitrary finite set. Thanks for pointing out the ambiguous notation; we have corrected it in the updated paper version.
>
> ---
>
> *Does $\mathbb{G}^\infty_{\alpha, t}$ depend on $\mu$ or not (see bottom of page 4)? Etc.*
>
> A: In our framework, the neighborhood distribution $\mathbb{G}^\infty_{\alpha, t}$ always depends on the mean field $\boldsymbol \mu$. For notational convenience, we sometimes drop the dependence on $\boldsymbol \mu$ in the notations. The lack of any comment on dropping the dependence in the notation led to understandable confusion. We have added an explanation to the updated draft, thanks.

---

### Author Response · Authors · 2023-11-17
**Response to all Reviewers**

We thank all reviewers for thoroughly reading our paper, the positive assessment of our work and the detailed and constructive feedback. Here is an overview of the most important changes:
- We have included an empirical comparison to LPGMFGs that supports the conceptional advantages of GXMFGs over existing approaches;
- We have fixed various typos;
- We have added more explanations to improve the readability and accessibility of the paper;
- We have extended the conclusion to include additional directions for future work;
- We uploaded the code to enhance reproducibility.

---

### Meta-Review · Area_Chair_8xCJ · 2023-12-05

**Metareview:**

This paper considers the problem of learning the behaviour of a large interacting number of agents. The approach builds on the concept of sparse graph limits, and in particular on the graphex framework, a generalisation of the (dense) graphon framework. The proposed approach can be seen as a generalisation of the existing work on mean-field graphon games to the sparse regime. A significant concern raised by a reviewer was the lack of comparison to alternative approaches. This has been addressed satisfactorily by the authors in their response and the revision. This is a nice contribution and I recommend acceptance.

As a side note, the paper is missing a key reference on graphex. Although the name "graphex" was coined by Veitch and Roy (2015), the graphex class is derived from the class of sparse (multi)graphs based on exchangeable random measures introduced earlier by Caron and Fox (2017) and the associated representation theorem by Kallenberg (1990). Caron and Fox (2017) should be cited alongside Veitch and Roy (2015) and Borgs et al. (2018b). A reference to Kallenberg's 1990 representation theorem, which underpins the graphex construction, also appears appropriate here.
F. Caron and E. Fox. Sparse graphs using exchangeable random measures. Journal of the Royal Statistical Society. Series B (Statistical Methodology), Vol. 79, No. 5, 2017. (first version 2014)
O. Kallenberg. Exchangeable random measures in the plane. J. Theoret. Probab., 3, 81-136, 1990.

**Justification For Why Not Higher Score:**

This is a nice paper, but the contribution may appear a bit niche for the ICLR community, and somewhat natural given the existing work on mean-field graphon games.

**Justification For Why Not Lower Score:**

This paper generalises the existing work on mean-field (dense) graphons to the sparse regime, via the use of graphex models. This is an interesting and useful extension.

---

### Decision · Program_Chairs · 2024-01-16

Accept (poster)